# Cohesin prevents cross-domain gene coactivation

Peng Dong [1,4] ✉, Shu Zhang [2], Valentina Gandin[1], Liangqi Xie[1,5], Lihua Wang[1], Andrew L. Lemire [1], Wenhong Li[1], Hideo Otsuna[1], Takashi Kawase[1], Arthur D. Lander [3], Howard Y. Chang [2] & Zhe J. Liu [1] ✉

The contrast between the disruption of genome topology after cohesin loss and the lack of downstream gene expression changes instigates intense debates regarding the structure–function relationship between genome and gene regulation. Here, by analyzing transcriptome and chromatin accessibility at the single-cell level, we discover that, instead of dictating population-wide gene expression levels, cohesin supplies a general function to neutralize stochastic coexpression tendencies of *cis*-linked genes in single cells. Notably, cohesin loss induces widespread gene coactivation and chromatin co-opening tens of million bases apart in *cis*. Spatial genome and protein imaging reveals that cohesin prevents gene co-bursting along the chromosome and blocks spatial mixing of transcriptional hubs. Single-molecule imaging shows that cohesin confines the exploration of diverse enhancer and core promoter binding transcriptional regulators. Together, these results support that cohesin arranges nuclear topology to control gene coexpression in single cells.

The mammalian genome within a three-dimensional (3D) nucleus folds into higher-order structures, including local chromatin loops, self-contacting topologically associating domains (TADs) and active or inactive compartments[1–4]. Extensive genomic and imaging studies revealed that cohesin is the primary molecular machinery driving the formation of loop domains and TADs[5–7]. The emerging picture is that the cohesin ring extrudes the chromatin fiber and generates high probability contacts along its path until the extrusion is blocked by convergent CTCF sites at the domain boundary[8–11]. Despite the well-characterized role of cohesin in genome organization, one converging and perplexing result is the lack of substantial gene expression changes at the cell population level after cohesin loss[6,7]. Recent live-cell studies further showed that cohesin-mediated loops are dynamic and rare[12,13], and direct stable key-and-lock physical interactions might not be required for gene activation[14]. Thus, the relationship between genome organization and transcriptional regulation remains under intense debate[15–19].

Genomic studies showed that cohesin loss tilts the balance of genome organization from TAD formation to compartmentalization, promoting cooperative contacts within active chromatin[6,7,20]. Consistent with these findings, super-resolution imaging by 3D assay for transposase-accessible chromatin-photoactivated localization microscopy (ATAC-PALM) revealed that accessible ATAC-rich regions within A compartments are organized into non-uniformly distributed clusters or accessible chromatin domains (ACDs), encompassing active chromatin and actively transcribed genes[21]. Loss of cohesin disrupts the spatial organization of these ACDs, resulting in their extensive spatial mixing[18]. In this study, we used single-cell genomic and imaging assays to investigate whether cohesin loss affects coexpression correlation of genes residing in ACD pairs on the same chromosome and between chromosomes. Given that cohesin is essential for mitosis and cell viability[22], assessing the long-term effects of its loss on gene regulation may be challenging while immediate impacts could

[1]Janelia Research Campus, Howard Hughes Medical Institute, Ashburn, VA, USA. [2]Center for Personal Dynamic Regulomes and Howard Hughes Medical Institute, Stanford University, Stanford, CA, USA. [3]Department of Developmental and Cell Biology, Center for Complex Biological Systems, University of California, Irvine, CA, USA. [4]Present address: Institute of Biomedical and Health Engineering, Shenzhen Institutes of Advanced Technology, Chinese Academy of Sciences, Shenzhen, China. [5]Present address: Cancer Biology and Infection Biology, Lerner Research Institute, Cleveland Clinic, Cleveland, OH, USA. ✉e-mail: p.dong@siat.ac.cn; liuz11@hhmi.org

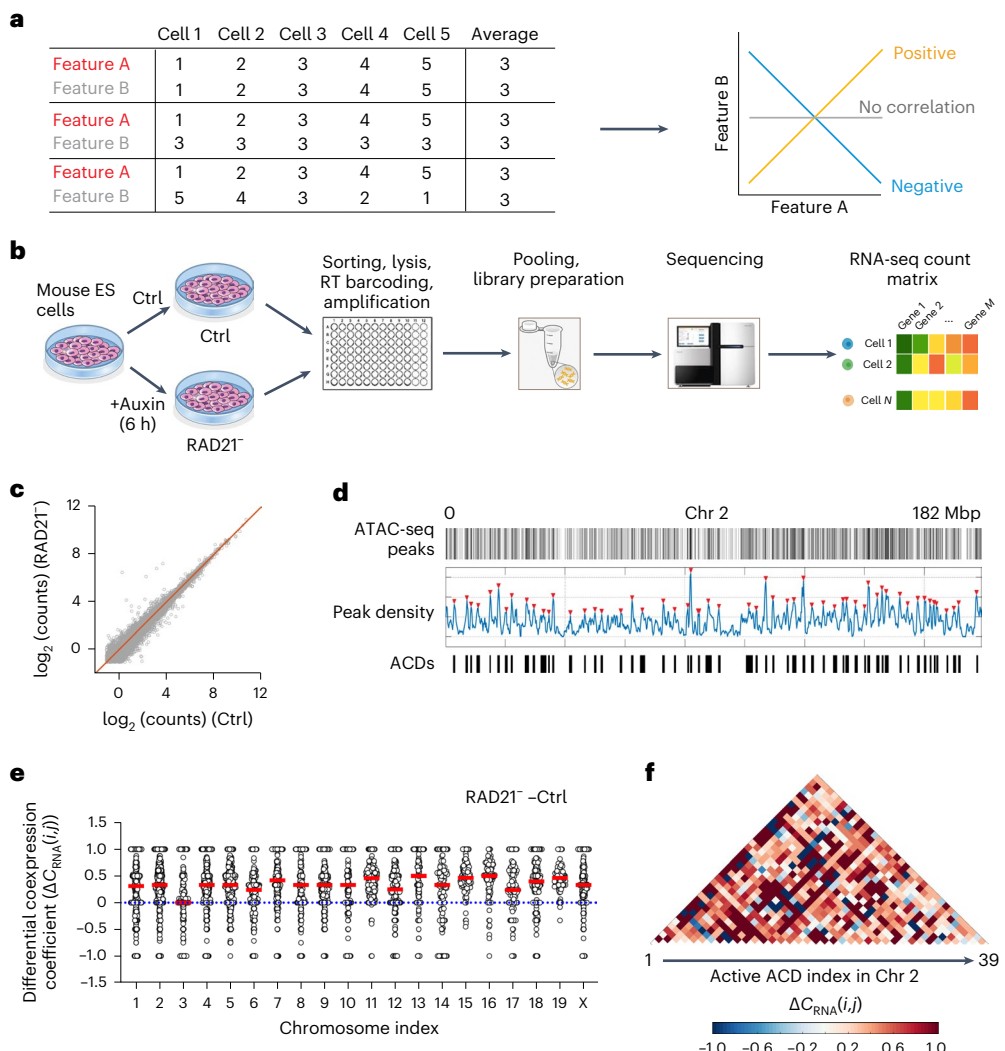

**Fig. 1 | Cohesin loss induces cross-domain gene coactivation. a**, Schematic diagram showing that the correlation of two genomic features A and B at the single-cell level may vary drastically even when their average levels remain the same in the cell population. **b**, The workflow for Smart-SCRB sequencing of control and RAD21-depleted mouse ES cells; Ctrl, control; RT, Reverse Transcription. **c**, Acute cohesin depletion results in only subtle changes in gene expression at the population level. Each circle represents a detected gene. Average gene expression was quantified by calculating the logarithmic value of averaged sequencing counts of ~400 individual cells. The red line marks the diagonal $y = x$. **d**, ACDs were identified based on ATAC peak densities by Gaussian peak fitting and thresholding for each chromosome. In total, 776 ACDs were identified across the genome. Red triangles indicate ATAC peaks. **e**, Statistics of $\Delta C_{RNA}(i,j)$ per ACD pair after cohesin depletion in chromosomes 1–19 and X. Each circle indicates the value of the differential coexpression coefficient for one ACD pair. The red line indicates the median value, and the dotted line indicates the zero-change line. Only active ACDs (those ACDs with detectable gene expression by Smart-SCRB) were included for analysis. The statistics were derived from 76,943 intrachromosomal ACD pairs with quantifiable values over 20 different chromosomes. **f**, Heat maps show elevated differential coexpression coefficients per active ACD pair in Chr 2 after cohesin depletion.

## Results

### Cohesin loss induces cross-domain gene coactivation

A simple example is provided here to illustrate that two genomic features can display distinct statistical relationships at the single-cell level even when their average levels remain the same in the population (Fig. 1a). To explore this possibility, we used Smart-Single-Cell RNA Barcoding (Smart-SCRB), a highly sensitive single-cell transcriptome assay[23,24]. Using a mouse embryonic stem (ES) cell line with cohesin subunit RAD21 fused to the auxin-induced degron (AID) system[18,25], we obtained Smart-SCRB data under control and acute cohesin loss conditions (6 h after auxin treatment; Fig. 1b and Extended Data Fig. 1a). In line with previous reports[6,7], aggregated sequencing data at the cell

population level showed only subtle gene expression changes (Fig. 1c and Extended Data Fig. 1b).

To evaluate cross-ACD gene coexpression, we next identified ACDs by calling local maxima of ATAC with sequencing (ATAC-seq) peak enrichment in the mouse ES cell genome (see details on ACD calling in Methods). As a result, each chromosome was segmented into a string of ACDs interspaced by ATAC-poor regions (Fig. 1d). In total, 776 ACDs with a size distribution from ~0.5 to ~2.5 megabase pairs (Mbps) were identified across the mouse genome (Extended Data Fig. 1c and Supplementary Table 1). When aligning ACDs with lamin-associated domains (LADs), compartments and ChromHMM-identified genomic regions[26–28], we found that ACDs overlap strongly with active compartments, enhancers and promoters but are more likely to be excluded from inactive compartments, intergenic regions, repressed chromatin, heterochromatin and LADs (Extended Data Fig. 1d).

be obscured by the averaging process inherent in cell population-based assays.

We then devised a statistical model based on Spearman's rank-order correlation to quantify gene coexpression coefficients per ACD pair (Extended Data Fig. 2a,b). We found that the average coexpression Spearman correlation ($\bar{S}_{RNA}$) per ACD pair decays as a function of the genomic distance, and cohesin loss increases $\bar{S}_{RNA}$ across all genomic length scales that we examined up to ~60 million bp apart (Extended Data Fig. 3a). It is worth noting that, using allele-specific single-cell RNA-sequencing (RNA-seq) data from mouse cells, a previous study showed that positive long-range gene coexpression correlation only occurs in *cis* but not between homologous chromosomes, and the hallmark of the *cis* dependence is the decay of the coexpression coefficient as a function of the genomic distance[29]. This result echoes with our observation, suggesting that gene coactivation that we measured occurs in *cis*. In addition, although cohesin loss altered the differential coexpression coefficient ($\Delta C_{RNA}(i,j)$) per ACD pair both positively and negatively in all chromosomes, the $\Delta C_{RNA}(i,j)$ averages showed uniform upward shifts toward the positive direction (Fig. 1e,f). As a control, we performed cross-validation by using two independent Smart-SCRB datasets under unperturbed conditions and revealed no significant changes (Extended Data Fig. 3b). To determine if this increased gene coexpression coefficient also occurs for ACDs from different chromosomes (*trans*), we calculated chromosome-wise differential coexpression coefficients by binning $\Delta C_{RNA}(i,j)$ per chromosome pair. We found that the pairwise correlations in *trans* increase broadly after cohesin depletion but are weaker and less significant than those in *cis*, suggesting a bona fide *cis* bias (Extended Data Fig. 3c,d). Next, we computationally parsed Smart-SCRB data to different cell cycle phases and found that acute cohesin removal led to a slight reduction of G1 (3.8%) and S (3.7%) phase cells and an increase in G2/M (7.5%) cells (Extended Data Fig. 3e). Notably, cohesin loss significantly increased cross-ACD gene coexpression coefficients in all cell cycle phases (Extended Data Fig. 3f), ruling out the cell cycle effect. Together, these findings suggest that cohesin loss selectively increases cross-domain gene coactivation probabilities in single cells without drastically altering average gene expression levels at the cell population level.

To probe possible links between gene coactivation and compartmentalization, we next parsed Hi-C interaction frequencies before and after cohesin loss into 776 ACDs. As expected, cohesin loss increased Hi-C contact frequencies per ACD pair in *cis* (Extended Data Fig. 3g,h), reflecting enhanced compartmentalization[6,7]. However, we did not detect a significant correlation between Hi-C contact frequency and the average coexpression Spearman correlation $\bar{S}_{RNA}$ per ACD pair (Extended Data Fig. 3i).

### Cohesin inhibits cross-domain chromatin co-opening

Gene expression is initiated by transcription factors binding to nucleosome-depleted regions such as enhancers and promoters. Thus, to dissect early regulatory changes after cohesin loss, we quantified chromatin accessibility by single-cell ATAC-seq (scATAC-seq)[30,31]. We obtained scATAC-seq data for two biological replicates before and after acute cohesin depletion (Fig. 2a) and binned scATAC-seq counts into 776 ACDs across the genome, yielding dense matrices for statistical analysis. Consistent with previous bulk ATAC-seq data showing that acute cohesin loss causes no significant changes in chromatin accessibility at enhancers, promoters and CTCF sites at the cell population level[18], aggregated scATAC-seq data revealed little changes of chromatin accessibility in ACDs before and after cohesin loss (Fig. 2b). We then used Spearman correlations to calculate chromatin coaccessibility per ACD pair in single cells (Extended Data Fig. 4a). In agreement with the Smart-SCRB data, we found that cross-ACD chromatin coaccessibility correlation decays as a function of genomic distance (Fig. 2c), and cohesin loss broadly increases the correlation across all genomic length scales and all chromosomes that we examined (Fig. 2c–g). As an internal reference, cohesin minimally affected the average coaccessibility correlation per ACD pair in *trans* (Extended Data Fig. 4b), consistent

with the notion that cohesin is a *cis*-acting factor in the interphase nucleus[9]. However, in contrast to Smart-SCRB data, we did observe a modest but positive correlation between Hi-C contact frequency and chromatin coaccessibility per ACD pair, and the correlation became much stronger after cohesin loss (Extended Data Fig. 4c). Together, these results support a role of cohesin in preventing megabase-scale, long-distance chromatin co-opening in *cis*.

### Cohesin prevents gene co-bursting along the chromosome

It was recently shown that early changes in gene activities and 3D genome organization can be investigated by genome-wide intron-sequential fluorescence in situ hybridization (seqFISH)[32]. To probe gene coactivation over large distances, we sought to use intron-seqFISH to examine co-bursting and spatial distribution of actively transcribed genes along the chromosome. Specifically, we designed seqFISH probes to simultaneously target intron regions of 208 randomly chosen active genes in ES cells across chromosome 2 (Chr2; Fig. 3a and Supplementary Table 2). To minimize the cell cycle effect, we performed seqFISH experiments 3 h after auxin treatment with 15 rounds of hybridization and imaging, followed by cell-level gene decoding (Fig. 3a,b, Extended Data Fig. 5a,b and Supplementary Video 1). The average gene bursting frequencies detected by seqFISH displayed a significant positive correlation with previously published nascent RNA-seq counts[19] (Extended Data Fig. 4d), cross-validating the detection reliability and sensitivity of the assay. In addition, intron-seqFISH imaging allowed us to spatially separate homologous chromosomes that occupy distinct territories in the nucleus (Fig. 3b). We used a distance cutoff of 1 μm to selectively quantify co-bursting events on the same chromosome (Extended Data Fig. 4e,f). Consistent with Smart-SCRB and scATAC-seq data, we found that cohesin loss elevates co-bursting of gene pairs across Chr 2 globally without substantially altering gene bursting frequencies over >1,000 cells analyzed for each condition (Fig. 3c–e). In addition, we found that cohesin loss leads to a reduction in physical distances between co-bursting gene pairs across Chr 2 (Fig. 3c,f). Consistent with this observation, 3D Airyscan imaging with a global FISH probe showed that actively transcribed genes in Chr 2 are organized to individual well-separated puncta before cohesin depletion (Fig. 4a,b). Cohesin loss induces extensive clustering of these puncta, leading to the formation of larger and more connected structures in the nucleus (Fig. 4c,d and Supplementary Video 2).

### Lineage-specific gene coactivation in *cis* after cohesin loss

To further validate these results, we performed intron-RNA-FISH to measure co-bursting of five representative pairs of lineage-specific genes far apart in *cis* (from 2.9 to 25.9 Mbps away from each other) in other chromosomes (Fig. 4e and Extended Data Fig. 6a–c). Single-cell intron-FISH analysis showed that cohesin loss significantly increases the co-bursting frequency for four of the five gene pairs tested (Fig. 4g), while bursting fractions for individual genes do not display a consistent trend of up- or downregulation in the population (Fig. 4f). Likewise, here we also used a cutoff of 1 μm to selectively analyze co-bursting events on the same chromosome. These results suggest that cohesin loss selectively increases co-bursting fractions of these gene pairs in single cells instead of altering bursting fractions of individual genes in the population. We also observed a reduction of average physical distances between these coactivated gene pairs (Fig. 4h,i and Extended Data Fig. 6c), echoing the finding that cohesin loss triggers clustering of actively transcribed genes in Chr 2. In comparison, single-cell intron-FISH analysis of six gene pairs from different chromosomes (*trans*) showed little change in either co-bursting frequencies or distances between bursting sites (Extended Data Fig. 6d,e).

### Cohesin blocks spatial mixing of Mediator hubs in live cells

Emerging studies indicate that a significant fraction of transcriptional regulators contain low-complexity domains that enable them to form

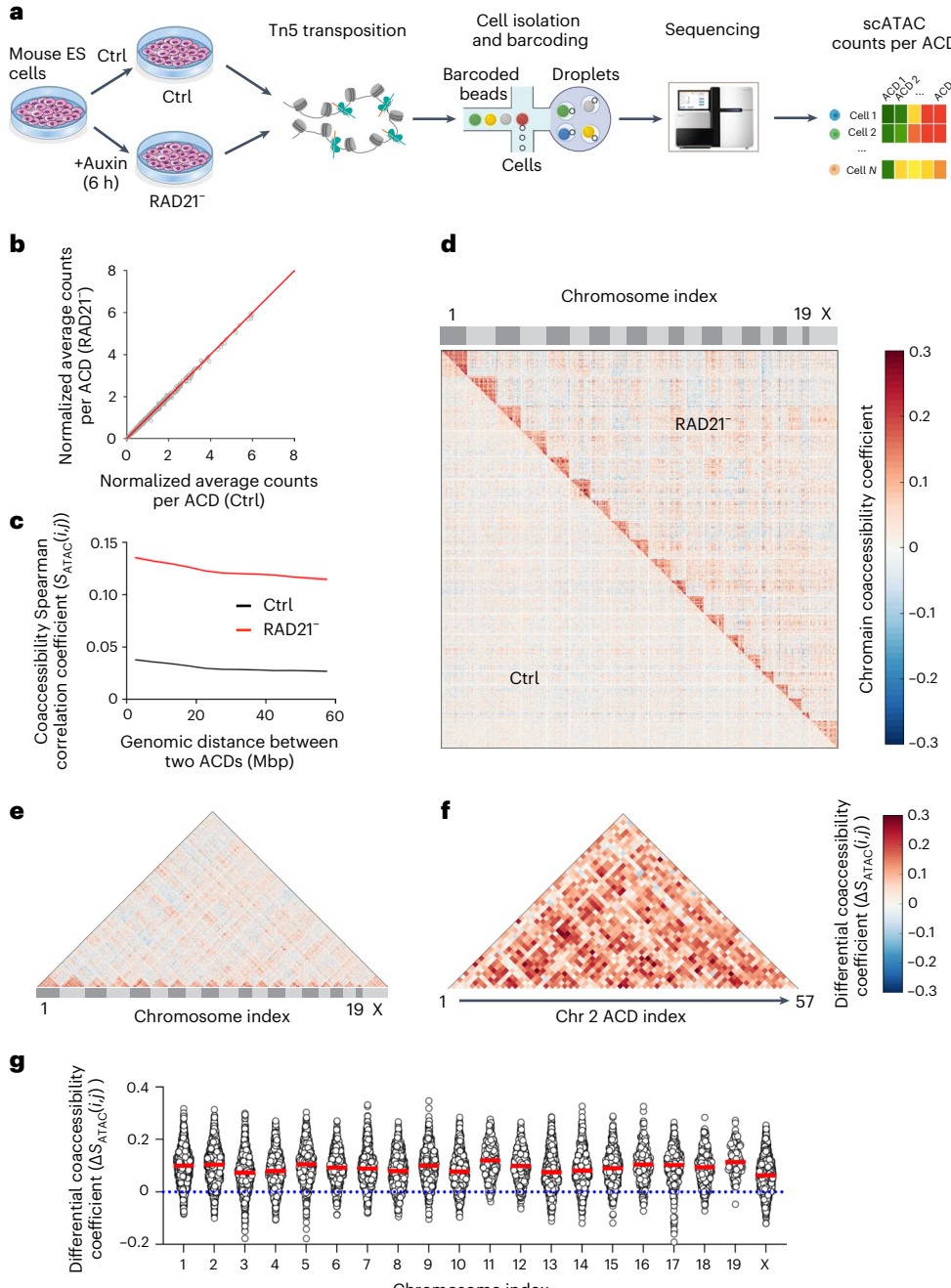

**Fig. 2 | Cohesin loss increases cross-domain chromatin co-opening.**
**a**, A schematic workflow for scATAC-seq (10x Genomics) of control and RAD21-depleted mouse ES cells. **b**, Acute cohesin loss results in no significant alteration of normalized average counts of ATAC peaks per ACD at the cell population level. Each circle represents one ACD, and empty ACDs without ATAC peaks were omitted from analysis. Normalized average counts of ATAC peaks per ACD were calculated by dividing the accumulated ATAC peak count per ACD by $M_{Average}$ (the average ATAC peak counts across all ACDs and cells). The red line marks the diagonal $y = x$. **c**, Acute cohesin loss increases chromatin coaccessibility per ACD pair globally. Coaccessibility Spearman correlation coefficients per ACD pair ($S_{ATAC}(i, j)$) under control (black) and cohesin-depleted (red) conditions were plotted as a function of the genomic distance after a five-point smoothing. Only the data from ACD pairs within the same chromosome were used to generate the plot. Data are presented as mean values ± s.e., and shadow regions indicate s.e.m. **d**, Heat map showing that acute cohesin loss

selectively increases intrachromosome chromatin coaccessibility per ACD pair across 20 chromosomes in the mouse genome. After filtering out cells with low read counts and batch normalization, around 3,000 cells were analyzed for each condition. **e**, The heat map shows elevated differential coaccessibility per ACD pair within each chromosome. Differential coaccessibility Spearman correlation coefficients per ACD pair were calculated by subtracting the coaccessibility Spearman correlation coefficient value under control conditions (**d**, top left) from that under cohesin-depleted conditions (**d**, top right). **f**, Heat map showing elevated differential coaccessibility per ACD pair in Chr 2. **g**, Dot plots of differential coaccessibility Spearman correlation coefficients ($\Delta S_{ATAC}(i, j)$) before and after cohesin depletion for chromosomes 1–19 and X. Every circle indicates the value of the differential coaccessibility Spearman correlation coefficient per ACD pair. The red line indicates the median value, and the dotted line indicates the zero-change line. The statistics were derived from 15,308 intrachromosomal ACD pairs with quantifiable values over 20 different chromosomes.

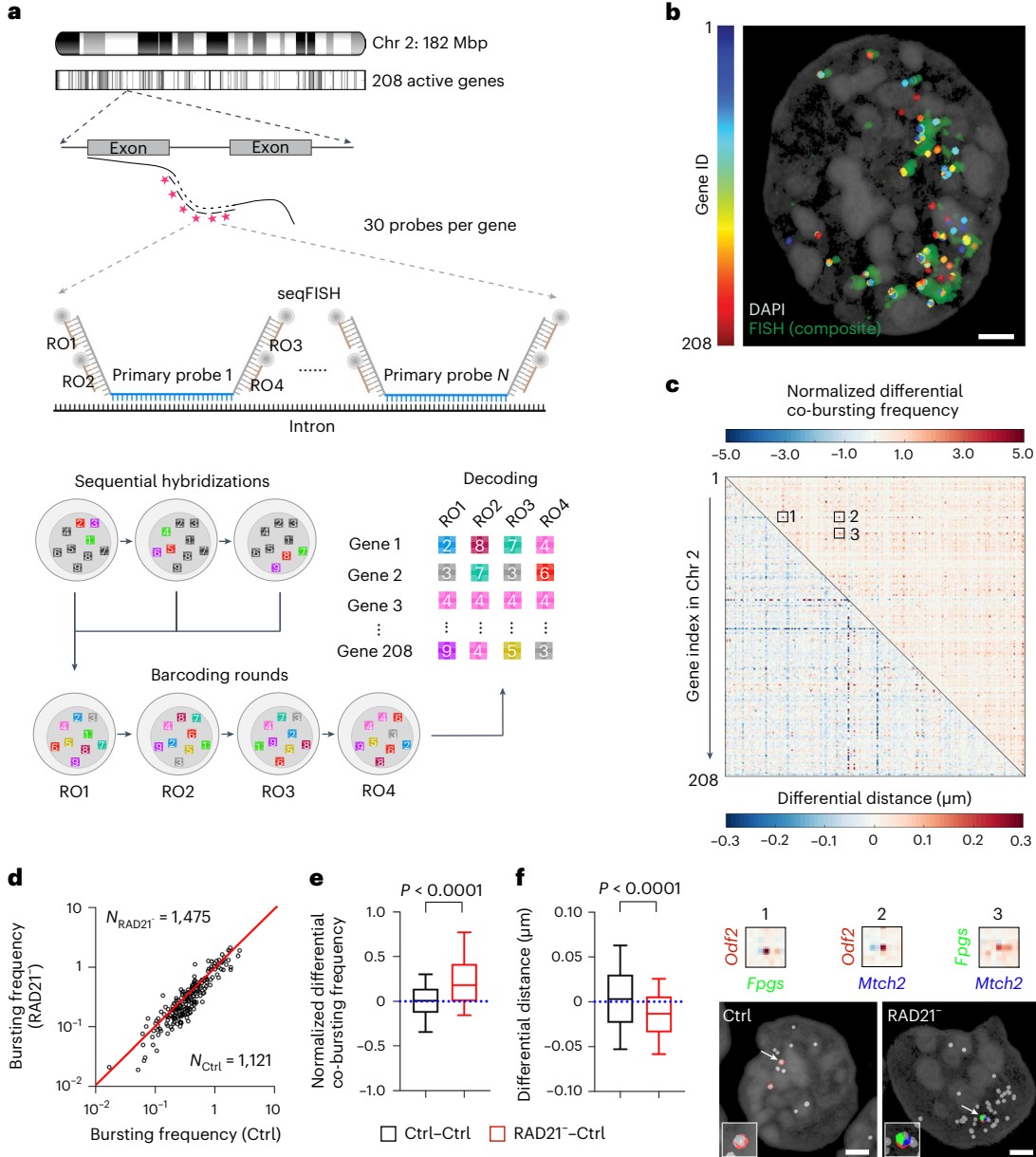

**Fig. 3 | Cohesin prevents gene co-bursting along the chromosome. a**, Active genes (208) across Chr 2 in mouse ES cells are targeted by 30 intron probes per gene. Primary probes for each gene contain four unique docking sites, and each docking site can be hybridized to a readout probe corresponding to nine pseudocolors in each readout round. The identity of each bursting gene can be determined according to the barcode after four readout (RO) rounds. The barcode is designed that the gene can be decoded even with one round of drop out. **b**, One representative cell with decoded genes (color coded). Composite FISH signals (green) were reconstructed by averaging images from all FISH channels from three hybridizations in RO1. The 3D image was rendered by using VVD-viewer[54]; scale bar, 2 μm. **c**, Heat map of pairwise differential co-bursting frequencies (top) and differential distances (bottom) for 208 genes in Chr 2. The matrix was calculated by subtracting values in the control condition (3 h) from those in the cohesin loss condition. Zoom-in views of the heat map and

representative images for three differentially coactivated genes (red, green and blue color coded) after cohesin loss are presented below; scale bar, 2 μm. **d**, Bursting frequencies for 208 genes (represented by circles) under control and cohesin loss (3 h) conditions. The bursting frequency per gene is calculated by dividing the total number of detected bursting sites (approximately zero to four in a cell) for the gene by the number of cells ($N$) analyzed under each condition (see Eq. (5)). The red line marks the diagonal $y = x$. **e,f**, Box plot of pairwise differential co-bursting frequencies (**e**) and differential distances (**f**) for the indicated conditions. A non-parametric two-sided Wilcoxon test was used for statistical testing. The bottom and top whiskers represent 10% and 90% values, respectively. The box represents the range from the 25th percentile to the 75th percentile, the center line represents the median, and the dotted line indicates the zero-change line. The statistics were derived from 21,528 gene pairs with quantifiable values.

protein hubs or condensates to modulate gene activities[33–36]. Thus, to investigate transcription mechanisms underlying cross-domain gene coactivation, we decided to image transcriptional hubs formed by Mediator because it has been well established that Mediator is structurally associated with RNA polymerase II[37] and is essential for transcription activation across the genome[38,39]. In addition, Mediator proteins

contain low-complexity domains (Extended Data Fig. 7a,b) and form protein hubs that are spatially correlated with transcriptional activities in living cells[33,34]. We used CRISPR–Cas9-based genome editing and fused HaloTag to endogenous Mediator subunits MED1/MED6 in the ES cell line harboring the RAD21 degron (Fig. 5a and Extended Data Fig. 7c–f). The biallelic fusion of MED1 or MED6 with HaloTag did

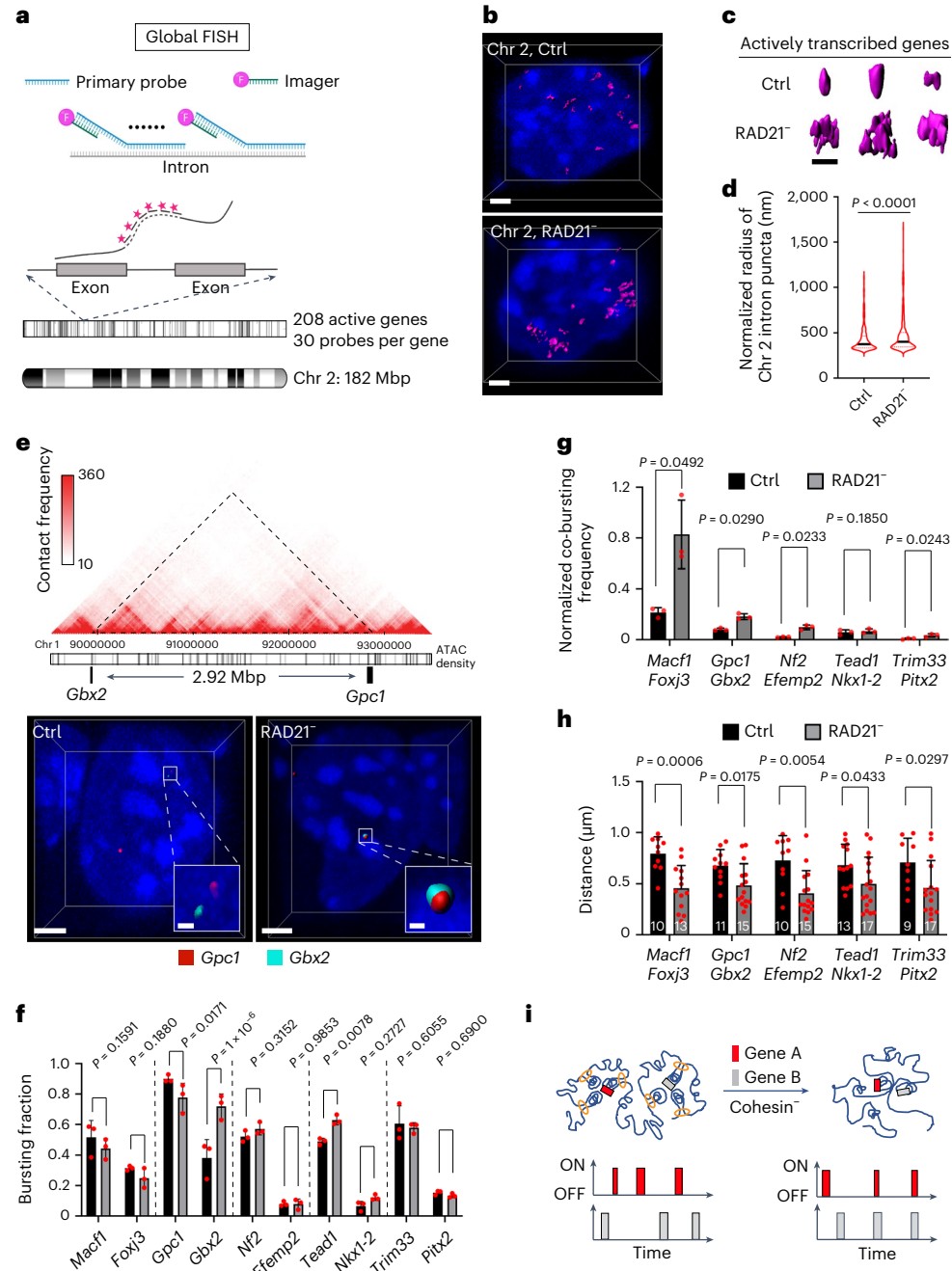

**Fig. 4 | Cohesin removal leads to spatial rearrangement of Chr 2 intron puncta and coactivation of lineage-specific genes. a**, Transcription bursting sites of 208 active genes from Chr 2 are collectively imaged by using intron primary probes (30 probes per gene) and a global imager probe conjugated to Alexa Fluor 647N. **b**, Representative 3D isosurface images show transcriptional bursting sites from 208 active genes across Chr 2 before (top) and after (bottom) cohesin loss. Nuclei were counterstained with DAPI (blue); scale bar, 2 μm. **c**, Isolated representative 3D isosurfaces for typical intron puncta formed by transcription bursting sites of 208 genes across Chr 2 before and after cohesin loss; scale bar, 1 μm. **d**, Violin plots show the normalized radii of detected Chr 2 intron puncta before and after acute cohesin depletion. Black lines are the median values, and dotted lines are the 25% and 75% quantiles. The measurement was repeated three times, and a non-parametric two-sided Wilcoxon test was used for statistical testing. **e**, Cohesin loss decreases the physical distance between coactivated lineage-specific genes *Gbx2* (ectoderm) and *Gpc1* (mesendoderm). Top, genomic

positions of *Gbx2* and *Gpc1* in Chr 1 with Hi-C and ATAC density information. Bottom, representative 3D isosurface images of *Gbx2* and *Gpc1* bursting sites before and after cohesin loss; scale bar, 2 μm; inset scale bar, 200 nm. **f**, Cohesin loss does not cause uniform up- or downregulation of bursting fractions for individual genes. **g**, Cohesin loss elevates normalized co-bursting frequencies of four of five pairs of lineage-specific genes. For **f** and **g**, data are presented as mean values ± s.d. The measurement was repeated three times independently, and two-sided Student's *t*-tests were used for statistical testing. **h**, Cohesin loss decreases average physical distances between coactivated lineage-specific genes in *cis* (five pairs). The number of data points (*n*) used for statistical analysis for each gene pair is marked at the bottom of the corresponding bar plot. A non-parametric two-sided Wilcoxon test was used for statistical testing. Data are presented as mean values ± s.d. **i**, Schematic showing chromatin reorganization (top) and gene coactivation (bottom) after cohesin depletion.

not significantly affect cell viability and cell cycle phasing (Extended Data Fig. 7k). Consistent with previous reports[33,34], we confirmed that MED1–HaloTag/MED6–HaloTag form concentrated hubs in both fixed and live cells (Fig. 5b and Supplementary Videos 3 and 4). Three-dimensional ATAC and Mediator PALM imaging showed that Mediator hubs overlap with ACDs extensively in fixed cells (Extended Data Fig. 8a–d and Supplementary Video 3).

Next, we confirmed that auxin treatment up to 6 h efficiently depletes RAD21 in MED1–HaloTag/MED6–HaloTag knock-in ES cell lines (Extended Data Fig. 7g,h). We then coupled acute cohesin loss with high-resolution 3D imaging of Mediator hubs in live cells by Airyscan microscopy and found that cohesin removal induces extensive structural changes in Mediator hubs, with a significant increase in average hub size (Fig. 5b–d and Supplementary Video 4). The average size of Mediator hubs was inversely correlated with residual RAD21 levels in single cells (Fig. 5e), suggesting a dose-dependent effect. We also observed an increase in the formation of 'super hubs' ($r > 500$ nm) after cohesin removal by dynamic mixing of smaller hubs observed under live imaging (Fig. 5f,g and Supplementary Video 5). It is worth noting that acute cohesin depletion does not affect nucleus size[18] nor MED1/MED6 protein levels (Extended Data Fig. 7e,f,i,j), excluding the possibility that spatial mixing of Mediator hubs is induced by altered protein concentrations.

## Localization of coactivated genes into shared Mediator hubs

Next, we analyzed the 3D volume overlap between Mediator hubs and bursting genes in Chr 2 by multiplexed protein and global intron-FISH imaging. We found that Mediator hub fusion spatially correlates with clustering of actively transcribed genes in Chr 2 after cohesin loss (Fig. 5h–j). In addition, we analyzed the colocalization between Mediator hubs and five pairs of lineage-specific genes that we examined previously (Extended Data Fig. 6a) and found significantly higher frequencies of co-bursting gene pairs connected by shared Mediator hubs after cohesin loss (Fig. 5k,l). As a control, we analyzed colocalization between Mediator hubs and co-bursting gene pairs localized within the same ACDs and found no significant changes after cohesin loss (Fig. 5l). Together, these results suggest that distant active genes in *cis* have higher probabilities of localizing in shared Mediator hubs after cohesin loss.

## Cohesin limits the exploration of transcriptional regulators

Gene activation is orchestrated by transcription factors and cofactors that are dynamically assembled at the enhancer and the core promoter to modulate transcription initiation and elongation. To further investigate transcriptional mechanisms underlying cross-domain gene coactivation, we decided to measure transcription factor binding and diffusion dynamics in live cells. We first selected a broad range of transcriptional regulators representing distinct processes in transcriptional

activation. These factors include site-specific transcription factors (OCT4), a histone mark reader (BRD4), Mediator subunits (MED1 and MED6) and a core promoter factor (TATA-binding protein (TBP)). Histones H2B and H2A.Z were also included as general markers for chromatin and active chromatin, respectively. We used bright, cell-permeable Janelia fluorophore dye JF549 (ref. 40) to label HaloTag fusion proteins that were either expressed endogenously (MED1, MED6 and BRD4) or were stably expressed at relatively low levels (OCT4, TBP, H2B and H2A.Z) in ES cells (Extended Data Figs. 7e,f and 9a,b). We performed fast live-cell, single-molecule imaging (100 Hz) followed by automated single-molecule tracking and diffusion analysis as previously described (Extended Data Fig. 9c)[41–44]. Surprisingly, a simple two- or three-state model[45] that assumes a bound state and one or two diffusive states failed to reveal significant differences in histone and transcription factor dynamics before and after cohesin loss (Extended Data Fig. 10a–g), suggesting that cohesin loss does not notably affect apparent diffusion and binding kinetics in the nucleus.

Next, we analyzed the local exploration of transcriptional regulators between clustered binding sites in the nucleus (Fig. 6a). Specifically, it was shown that transcription factors search for targets within protein hubs via a bound-state-dominant mode[41,43,46] likely guided by transient interactions with cofactors and non-coding RNAs[47,48]. Recently, a parameter called 'radius of confinement (RoC)' was specifically developed to analyze local exploration by quantifying the degree of confinement for transcription factors that are both tightly and loosely bound to chromatin[49]. By using a longer acquisition time (50 ms) and thus selectively imaging the dynamics of chromatin-bound molecules, we compared the RoC cumulative probability distributions before and after cohesin loss (Fig. 6b). We found that cohesin loss consistently increases the RoC for each representative transcriptional regulator tested (Fig. 6c). By contrast, the RoC for the active chromatin marker H2A.Z showed slight decreases (Fig. 6c). Interestingly, the RoC curve for the general chromatin marker H2B adopted a less sigmoid shape (Fig. 6c), implying more homogeneous chromatin movements after cohesin loss, consistent with previous reports[13,50]. Together, these results suggest that cohesin loss has distinct effects on transcriptional regulators and chromatin and that cohesin confines the exploration of a broad range of enhancer and core promoter binding transcriptional regulators.

## Discussion

Because of the limited space in the cell nucleus, it was proposed that higher-order organization of chromosomes partitions the genome into insulated structural and functional units for gene regulation[51]. Indeed, a fundamental question is whether a mechanism exists to prevent local biochemical fluxes from broadly influencing genes connected by the chromatin fiber. Recently, a series of elegant single-cell studies showed that long-distance stochastic gene coexpression in *cis* is significantly

**Fig. 5 | Cohesin separates transcriptional hubs. a**, Diagram showing biallelic integration of HaloTag and green fluorescent protein (GFP)–mini-AID (mAID) into endogenous *Med6* and *Rad21* gene loci (for auxin-induced protein degradation). **b**, Three-dimensional isosurface reconstruction of MED6 hubs (color coded by 3D volumes). Inlets are MED6 fluorescence images; scale bars, 2 μm. **c,d**, Violin plots show the size distribution of MED6 (**c**) and MED1 (**d**) hubs. Black lines are median values, and dotted lines are 25% and 75% quantiles. A non-parametric two-sided Wilcoxon test was used for statistical testing. **e**, MED6 hub sizes are inversely correlated with residual RAD21 protein levels in single cells. An *F*-test indicates that the slope is significantly non-zero with $P < 0.0001$; AU, arbitrary units. **f**, Timelapse imaging of MED6 hubs during cohesin depletion; scale bar, 2 μm. **g**, Quantification of MED6 hub sizes (red) and RAD21 residual levels (gray) in **f**. For MED6 hub size analysis, the statistics were derived from the top 50 hubs, and data are presented as mean values ± s.e. **h**, Three-dimensional isosurface rendering of overlaps between MED6 hubs (color coded by 3D volumes) and transcription bursting sites for 208 genes in Chr 2 (magenta);

scale bars, 2 μm. **i**, Representative 3D volume overlaps between MED6 hubs and Chr 2 intron clusters; scale bar, 1 μm. **j**, Violin plots show the statistics of 3D volume overlaps between MED6 hubs and transcription bursting sites for 208 genes in Chr 2. Black lines are the median values, and dotted lines are 25% and 75% quantiles. The measurements were obtained from 20 cells, and a non-parametric two-sided Wilcoxon test was used for statistical testing. **k**, Three-dimensional isosurface reconstruction of MED6 hubs and intron-FISH signals showing a representative case that co-bursting gene loci are connected by MED6 hubs after cohesin loss; scale bar, 500 nm; smRNA-FISH, single-molecule RNA-FISH. **l**, Bar plots for the fraction of co-bursting loci connected by MED6 hubs for paired genes within (shadowed) and across ACDs. The fraction was calculated by dividing the number of co-bursting loci that share a common MED6 hub by the total number of co-bursting loci. The measurement was repeated three times, and two-sided Student's *t*-tests were used for statistical testing. Data are presented as mean values ± s.d.

higher than that in *trans*, and genes encoding components of the same protein complex tend to be chromosomally linked[29,52,53]. Here, by single-cell sequencing and imaging, we found that, instead of affecting population-wide gene expression levels, acute cohesin removal induces widespread cofluctuation of gene activities and chromatin accessibility tens of million bases apart in *cis*. These results suggest that cohesin supplies a general function to neutralize stochastic coexpression tendencies of *cis*-linked genes in single cells. By suppressing long-range gene coactivation, the cohesin system would reduce gene–gene interference and make the placement of genes along chromosomes more flexible during genome evolution. Another physiological implication is that *cis*-linked developmental genes would have higher probabilities of

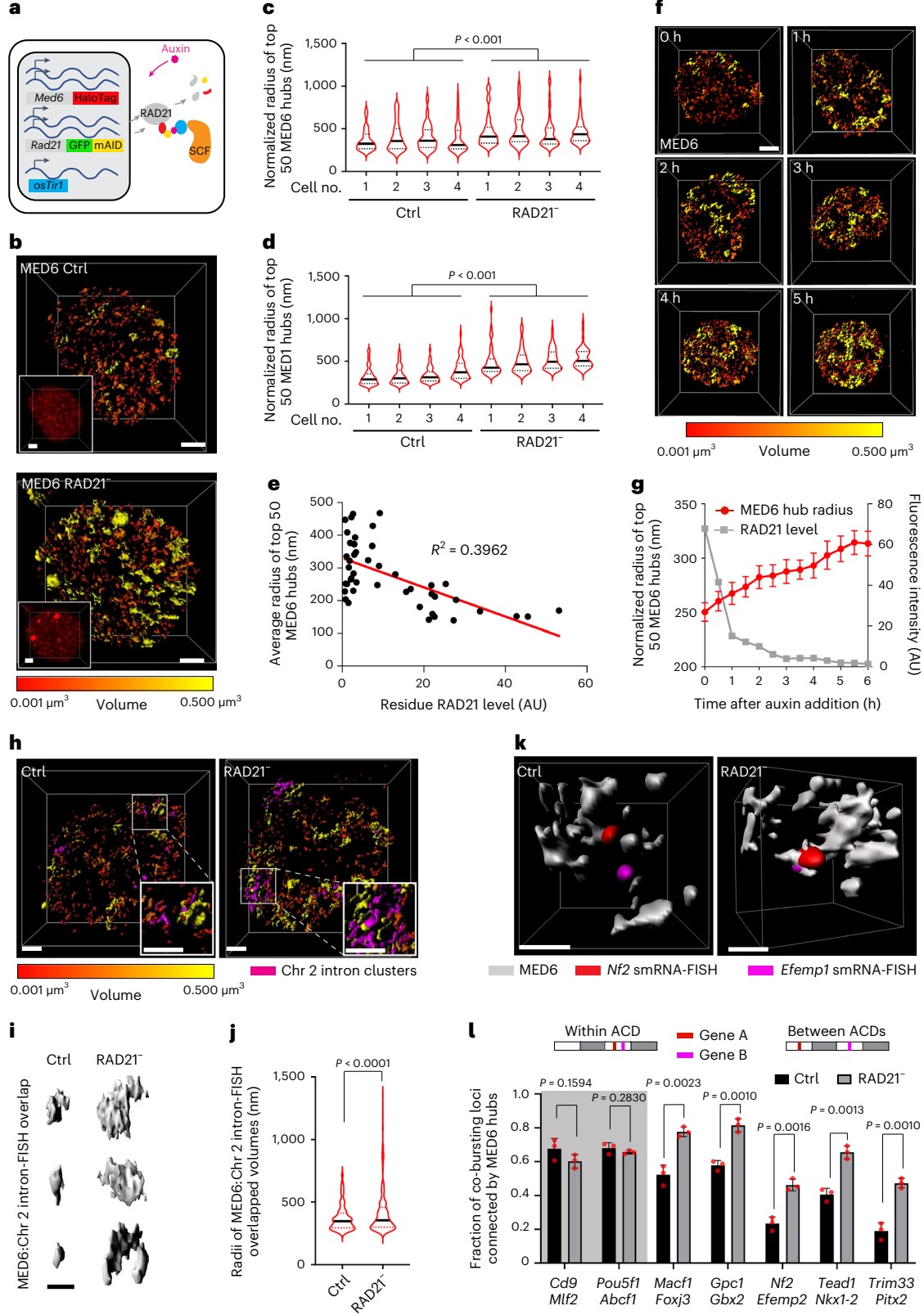

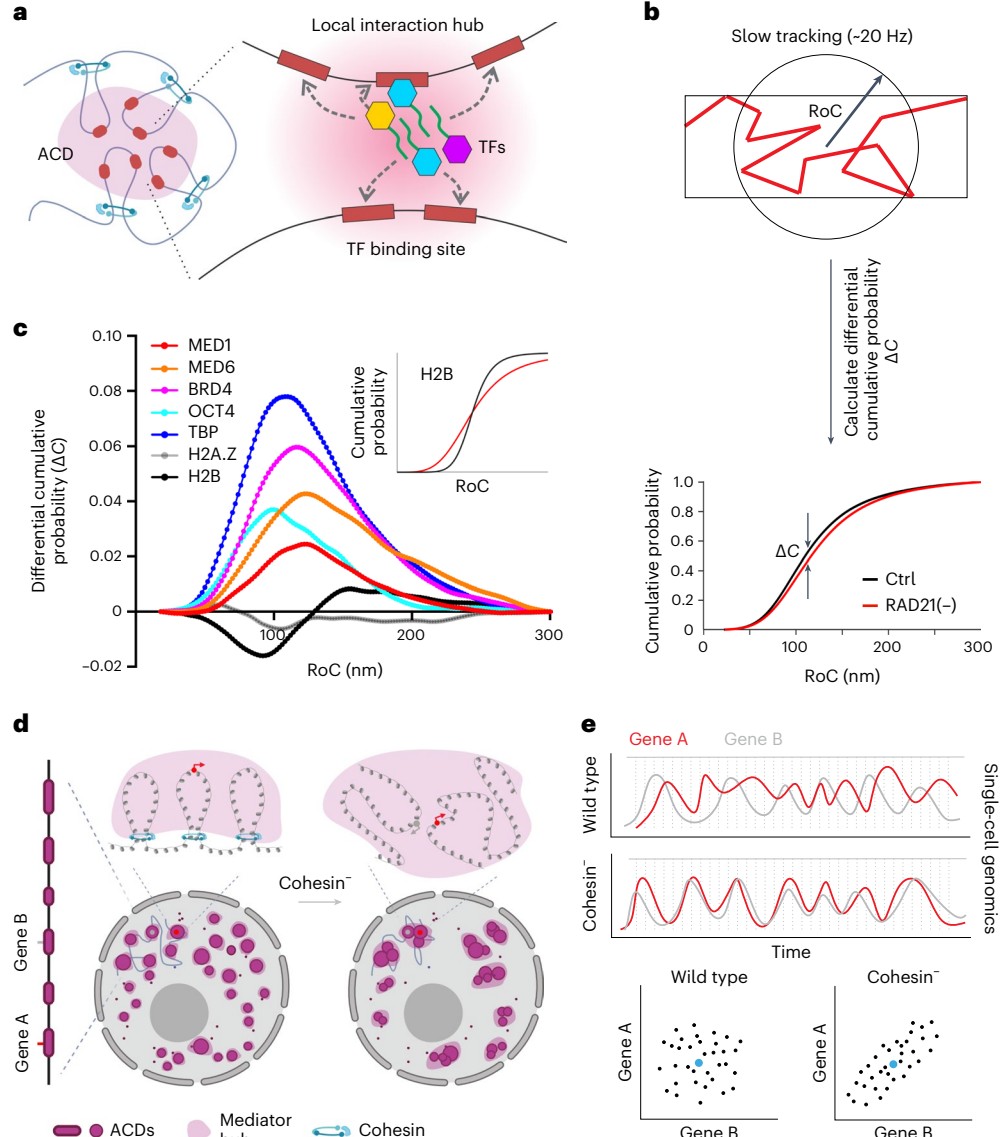

**Fig. 6 | A role of cohesin in regulating gene coexpression. a**, Transcription factors search for targets in a local interaction hub by a bound-state-dominant mode. Local exploration is proposed to be guided by transient interactions with other co-regulators (proteins or non-coding RNAs) in the hub; TF, transcription factor. **b**, The diagram illustrates the quantification of the RoC and the calculation of its differential cumulative probability $\Delta C$ ($\Delta C = \Delta C_{Ctrl} - \Delta C_{RAD21-}$) over analyzed tracks. For this analysis, both loosely and tightly bound fractions were obtained by using a slower frame rate (20 Hz). **c**, Differential cumulative probability ($\Delta C$) of the RoC for histone subunits (H2B and H2A.Z) and a broad range of transcriptional regulators (MED1, MED6, BRD4, OCT4 and TBP). Histone subunits H2B and H2A.Z were analyzed as controls. The inset shows curves

illustrating the cumulative probability of the RoC for H2B before (black) and after (red) cohesin loss. **d**, Cohesin loss leads to disruption of chromatin loops and spatial mixing of ACDs and Mediator hubs. As a result, the average distance between distant active genes (red and gray dots) in *cis* decreases, accompanied by their colocalization into a shared transcriptional hub and an elevated chance for their co-regulation. **e**, Schematic diagrams illustrating that an increase in cofluctuation of genes A (red) and B (gray; top) would alter gene coexpression correlation in single cells (bottom) without affecting their average levels (blue dots) at the cell population level. Gray dotted lines reflect sampling points of single-cell genomics for the correlation analysis in the bottom plots.

coactivation when the cohesin function is compromised, as we showed here for representative lineage-specific genes. However, due to the deleterious effect of cohesin loss on the cell cycle, it is difficult to examine the impact on lineage commitment at later time points. Analyzing cell fate determination in cohesin haploinsufficient disease models, such as Cornelia de Lange syndrome, could provide more insights into developmental consequences.

Gene activation is controlled by transcription factors and cofactors that are dynamically engaged at the enhancer and the core promoter. Recent studies indicated that a large fraction of transcriptional regulators contain low-complexity domains, which enable them to

form protein hubs to further modulate gene activities[33–36]. Thus, to understand the physical mechanism by which cohesin prevents cross-domain gene coactivation, it is essential to examine how cohesin regulates (1) the spatial organization of *cis*-regulatory elements (that is, enhancers and promoters), (2) the exchange and assembly dynamics of transcriptional regulators on these elements and (3) the ability of transcription factors to form protein hubs.

One informative clue comes from reports showing that acute cohesin removal promotes compartmentalization and spatial mixing of active chromatin[6,7,18]. Here we found that the spatial mixing of active chromatin after cohesin loss is accompanied by aberrant clustering of

genes along the chromosome and dynamic mixing of transcriptional hubs. As a result, genes located in distant active domains have higher probabilities of localizing to shared transcriptional hubs (Fig. 6d). These structural changes could potentially subject genes across architectural boundaries under the influence of similar local gene regulatory fluxes, elevating the chance of their co-regulation (Fig. 6e). Consistent with this model, scATAC-seq analysis showed that cohesin prevents cross-domain chromatin co-opening, suggesting that cohesin may restrict transcription factors and associated chromatin remodeling activities from acting across large genomic distances. Indeed, live-cell, single-molecule imaging shows that cohesin inhibits the exploration of diverse transcriptional regulators over long distances. Together, these results support a role of cohesin in arranging nuclear topology to control gene coexpression in single cells.

## Online content

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

## Methods

This research complies with all relevant ethical regulations by Environment, Health and Safety from Howard Hughes Medical Institute (HHMI)/Janelia Research Campus.

### Chemicals

The plant auxin analog indole-3-acetic acid sodium salt (Millipore Sigma, I5148) was dissolved in double-distilled water to a stock concentration of 500 mM, aliquoted and stored at −20 °C and used at a final concentration of 500 μM (acute depletion) or 5 μM for 6 h. α-Amanitin (Tocris, 4025) was dissolved at 1 mg ml$^{-1}$ in double-distilled water, aliquoted and stored at −20 °C and used at a final concentration of 100 μg ml$^{-1}$.

### Cell culture

JM8.N4 mouse ES cells from the C57BL/6N strain and their genome-edited derivatives were routinely cultured in 60-mm plates coated with 0.1% gelatin without feeders at 37 °C and 5% $CO_2$. The mouse ES cell culture medium was composed of optimized knockout DMEM for mouse ES cells (Thermo Fisher Scientific, 10829-018), 15% ES cell-qualified fetal bovine serum (ATCC, SCRR-30-2020), 1,000 U of leukemia inhibitory factor (self-purified), 1 mM GlutaMAX (Thermo Fisher Scientific, 35050-061), 0.1 mM MEM non-essential amino acids (Thermo Fisher Scientific, 11140-50), 0.1 mM β-mercaptoethanol (Thermo Fisher Scientific, 21985-023) and antibiotic–antimycotic (Thermo Fisher Scientific, 15240-062). 2i inhibitors were also added to the medium at final concentrations of 1 μM PD0325901 (Millipore Sigma, PZ0162) and 3 μM CHIR99021 (STEMCELL Technologies, 72052), respectively.

### Generation of stable cell lines and genetic knock-in cells

We constructed plasmids PiggyBac-EF1-HaloTag-OCT4, PiggyBac-EF1-H2A.Z-HaloTag and PiggyBac-EF1-HaloTag-mouse TBP (mTBP) based on an available PiggyBac-EF1-HaloTag backbone vector. We used them together with the PiggyBac supertransposase to generate stable cell lines that expressed HaloTag–OCT4, H2A.Z–HaloTag and HaloTag–mTBP fusion proteins for single-particle tracking experiments. For H2B single-particle tracking experiments, we adopted a previously constructed PiggyBac-EF1-H2B-HaloTag plasmid[55]. Maps of plasmids used in this study are available upon request.

To generate stable HaloTag–OCT4- and HaloTag–mTBP-expressing mouse ES cells, 8 μg of PiggyBac-EF1-HaloTag-OCT4 or PiggyBac-EF1-HaloTag-mTBP was nucleofected with 8 μg of PiggyBac supertransposase into ~1 × 10$^6$ mouse ES cells. The derived mouse ES cells were selected with 500 μg ml$^{-1}$ G418 (Thermo Fisher Scientific, 10131035) for ~5 days, stained with JF549 dye and sorted by fluorescence-activated cell sorting (FACS) for cell populations with intermediately expressed genes.

We have applied previously described methods to generate single guide RNA (sgRNA) constructs and to construct HaloTag–MED1, MED6–HaloTag and HaloTag–BRD4 donor plasmids for generating knock-in cells by CRISPR–Cas9 genome editing[18]. The corresponding Emerald versions of MED1/MED6 donor constructs were generated by replacing the HaloTag DNA coding sequence with Emerald coding sequence by Gibson assembly (New England Biolabs, E5520S). We generated HaloTag–MED1, MED6–HaloTag and HaloTag–BRD4 genetic knock-in cells based on a previously generated *Rad21* mAID degron ES cell line[18]. Briefly, 1.0 μg μl$^{-1}$ SpCas9-sgRNA-PGK-Venus construct and 1 μg μl$^{-1}$ donor construct were nucleofected into ~3 × 10$^6$ Tir1 mouse ES cells using Amaxa 4D-Nucleofector and P3 Primary Cell 4D-Nucleofector X kits (Lonza, V4XP-3024) following the manufacturer's protocols. Three days following nucleofection, cells were then stained with 50 nM JF549 HaloTag ligand (JF549) for 30 min, washed three times in 1× PBS and then in mouse ES cell medium for 15 min and subjected to sorting by FACS. JF549$^+$ cells were plated sparsely in 60-mm tissue culture plates and grown for another 5-7 days. Single colonies

were picked for genotyping by designing PCR primers outside of the homology arms. Biallelic knock-in mouse ES cell clones were verified by PCR genotyping, Sanger sequencing and western blotting.

### Western blotting

Mouse ES cells were lysed in 1× SDS sampling buffer (200 mM Tris-HCl (pH 7), 10% glycerol, 2% SDS, 4% β-mercaptoethanol, 400 mM DTT and 0.4% bromophenol blue). Lysates were sonicated and incubated on ice for 30 min, mixed with 2× loading buffer and denatured at 95 °C for 5 min. Lysed proteins were resolved by SDS–PAGE using Mini-PROTEAN TGX Precast Gels (Bio-Rad). The following primary antibodies were used: rabbit polyclonal anti-RAD21 (D213; Cell Signaling, 4321, 1:1,000 dilution), rabbit polyclonal anti-MED1 (CRSP1/TRAP220; Bethyl Laboratories, A300-793A, 1:1,000 dilution), rabbit polyclonal anti-MED6 (Abcam, ab220110, 1:1,000 dilution), rabbit monoclonal anti-BRD4 (E8V7I; Cell Signaling, 54615, 1:1,000 dilution), rabbit polyclonal anti-TBP (Cell Signaling, 8515, 1:1,000 dilution), rabbit polyclonal anti-OCT4 (Abcam, ab19857, 1:2,000 dilution), anti-histone H2A.Z (EPR18090; Abcam, ab188314, 1:2,000 dilution) and rabbit monoclonal anti-α-tubulin (11H10; Cell Signaling, 2125, 1:2,000 dilution). We used horseradish peroxidase-linked anti-rabbit IgG (Cell Signaling, 7074) secondary antibodies at a dilution of 1:1,000. Western blots were exposed by using the 20× LumiGLO chemiluminescent detection system (Cell Signaling, 7003) and imaged by using a Bio-Rad ChemiDoc MP detection system.

### Cell cycle analysis

Cell cycle analysis was performed by using propidium iodide staining following the protocols from the propidium iodide flow cytometry kit (Abcam, ab139418). Briefly, cells were trypsinized into single-cell suspensions and fixed with 67% ice-cold ethanol in PBS overnight at 4 °C. The next day, the cells were rehydrated with PBS, stained with propidium iodide (final concentration of 50 μg ml$^{-1}$) and treated with RNaseA (final concentration of 50 μg ml$^{-1}$) for 30 min at 37 °C before flow cytometry analysis. All samples were acquired on a Beckman Coulter CytoFLEX S with four lasers (405 nm, 488 nm, 561 nm and 638 nm) and operated by CytExpert Software v2.3 (Beckman Coulter) using 488-nm FSC-H (488-FSC-H) as the threshold parameter (threshold automatic setting). The detector's gain for fluorescence (CD4-FITC) and FSC/SSC detection were optimized by using control cells without treatment. The following gains were used for detectors with different spectral filters: 39, 15, 28 and 10 arbitrary units for 488-FSC, 488-SSC, 561-585/42 and 561-610/20, respectively. SSC-A versus FSC-W was used for initial gating of singlet cells, followed by SSC-A versus FSC-A to further define cellular events. An event count versus 561-610/20 histogram plot was used to determine the percentage of cells in G1, S and G2/M phases of the cell cycle. All samples were acquired for 5 min at a sampling rate of 30 μl min$^{-1}$ or up to 15,000 cells. FlowJo v.10.7.1 (Flowjo) was used for analysis of the flow cytometry data.

### Smart-SCRB data acquisition

Mouse ES cells with an engineered AID system treated with or without auxin were resuspended in culture medium without Phenol red and sorted by FACS into 96-well PCR plates containing 3 μl of mild lysis buffer (nuclease-free water with 0.2% Triton X-100 + 0.1 U μl$^{-1}$ RNase inhibitor (Lucigen, 30281-2)). A total of ~400 cells were sorted and collected for each sample. The PCR plates with collected cells were briefly centrifuged, immediately frozen and stored at −80 °C until cDNA synthesis.

One microliter of harsh lysis buffer (50 mM Tris (pH 8.0), 5 mM EDTA (pH 8.0), 10 mM DTT, 1% Tween 20, 1% Triton X-100, 0.1 g l$^{-1}$ proteinase K, 2.5 mM dNTPs and ERCC Mix (10$^7$-fold dilution)) and 1 μl of 10 mM barcoded RT primer was added to each well. Plates were incubated at 50 °C for 5 min to lyse cells, and proteinase K was heat inactivated by subsequently incubating at 80 °C for 20 min. To minimize

contamination across wells, heavy-duty plate seals and quantitative PCR (qPCR) compression pads (Thermo Fisher Scientific, 4312639) were used to seal the plates. The lysis reaction was mixed with 2 μl of reverse transcription master mix 5× buffer (Thermo Fisher Scientific, 11756500), 2 μl of 5 M betaine (Sigma-Aldrich, B0300-1VL), 0.2 μl of 50 mM E5V6NEXT template switch oligonucleotide (Integrated DNA Technologies), 0.1 μl of 200 U μl⁻¹ Maxima H-RT (Thermo Fisher Scientific, EP0751), 0.1 μl of 40 U μl⁻¹ NxGen RNase Inhibitor and 0.6 μl of nuclease-free water (Thermo Fisher Scientific, AM9932). The reaction system was then incubated at 42 °C for 1.5 h, followed by 10 min at 75 °C to inactivate reverse transcriptase. PCR was performed by adding 10 μl of 2× HiFi PCR mix (Kapa Biosystems, 7958927001) and 0.5 μl of 60 mM SINGV6 primer and running the following program: 98 °C for 3 min, 20 cycles of 98 °C for 20 s, 64 °C for 15 s and 72 °C for 4 min and a final extension step of 5 min at 72 °C.

Single-cell cDNA was pooled by plate to make libraries. cDNA (600 pg) from each sample plate was used in a modified Nextera XT (Illumina, FC-131-1024) library preparation but using the P5NEXTPT5 primer and a tagmentation time of 5 min. The resulting libraries were purified following the Nextera XT protocol (0.6× ratio) and quantified by qPCR using a Kapa Library Quantification kit (Kapa Biosystems, KK4824). Four plates were pooled together on a NextSeq 550 high-output flow cell or NextSeq 2000 P2-100 flow cell with 26 bp in read 1 and 50 bp in read 2. PhiX control library (Illumina) was spiked in at a final concentration of 7.5% to improve color balance in read 1. Read 1 contains the spacer, barcode and unique molecular identifier and read 2 represents a cDNA fragment from the 3′ end of the transcript. The entire experimental procedure was replicated two more times, and ~400 cells were analyzed for each condition (control versus auxin treated).

Alignment and count-based quantification of single-cell data were performed by removing adapters, tagging transcript reads to barcodes and UMIs and aligning the resulting data to the mouse genome mm10. After quantification, 2,600 detected genes (gene_det > 2,600) was set as a threshold to eliminate unreliable sequencing results, and the data matrix ($M_{RNA}$) for all remaining cells was used for analysis.

## ACD calling
To identify ACDs across the genome, we referred to available bulk ATAC-seq data[18] and used a genomic distance of 200 kb as the binning unit for analysis. The MATLAB function findpeaks() was then called in 'MinPeakProminence' mode, and the value of 'MinPeakProminence' was set to the average of overall ATAC signals. In total, 776 ACDs were identified by using these criteria. The reason for using a large bin is to make sure that we generate a dense matrix without too many zeros. This is essential for robust downstream coexpression and coaccessibility calculations, as normal statistics break down when dealing with a sparse matrix with a lot of zeros.

To align ACDs with genomic features including A/B compartments, LADs and ChromHMM regions, the mouse genome (mm10) was divided into 500-bp windows using the 'tileGenome' function within the GenomicRanges package. These windows were then overlapped with multiple genomic features, including mouse ES cell bulk ATAC-seq peaks[18], LAD regions[27], A/B compartments[26] and ChromHMM states[28], and annotated accordingly. To assess the enrichment of ATAC-seq peaks within each of these annotated groups, we calculated the fold change as the ratio of ATAC-seq peaks in each group compared with the ratio across the entire genome. Groups exhibiting a fold change of greater than 1 were considered to be enriched with ATAC-seq peaks.

## Calculation of normalized contact frequency per ACD pair
To calculate the normalized contact frequency for a specific ACD pair, we used published mouse ES cell Hi-C data and calculated the average value over all the contact frequencies within the boundaries of the ACD pair[18]. The average value was defined as the normalized contact frequency for that specific ACD pair.

## Gene coexpression analysis
To evaluate Spearman's rank-order correlation among gene pairs, we filtered out low-expressed genes from $M_{RNA}$ with a threshold of 1 after averaging the RNA-seq counts over the single cells analyzed. For each pair of expressed genes, the Spearman correlation coefficient $S_{x,y}$ (considering tied ranks) was calculated by

$$S_{x,y} = \frac{\sum_i (x_i - \bar{x})(y_i - \bar{y})}{\sqrt{\sum_i (x_i - \bar{x})^2 \sum_i (y_i - \bar{y})^2}}. \tag{1}$$

$x_i$ and $y_i$ are the sequencing counts for genes $x$ and $y$ in the $i$th individual cell, respectively. For each ACD pair (ACD1 and ACD2), the Spearman correlation matrix $S$ was determined by calculating the Spearman correlation coefficient for each gene pair from ACD1 and ACD2, and the average Spearman correlation coefficient $\bar{s}_{RNA}$ was calculated by taking the average value across the whole matrix $S$ for control and cohesin-depleted conditions.

The differential Spearman correlation matrix $\Delta S_{RNA}$ was calculated by

$$\Delta S_{RNA} = S_{RNA,RAD21-} - S_{RNA,Ctrl}, \tag{2}$$

where $S_{RNA,RAD21-}$ and $S_{RNA,Ctrl}$ are Spearman correlation matrices for cohesin-depleted and control conditions, respectively. The workflow for the above analyses is illustrated in Extended Data Fig. 2a.

The gene coexpression coefficient $\Delta C_{RNA}(i,j)$ between two ACDs within the same chromatin was estimated by using the following equation:

$$\Delta C_{RNA}(i,j) = \frac{n_{\Delta S+} - n_{\Delta S-}}{n_{\Delta S+} + n_{\Delta S-}}, \tag{3}$$

where $n_{\Delta S+}$ and $n_{\Delta S-}$ represent the number of positive and negative elements within the differential Spearman correlation matrix $\Delta S_{RNA}$. The derived differential gene coexpression matrix $\Delta C_{RNA}$ was plotted as a heat map. The workflow for evaluating $\Delta C_{RNA}$ is illustrated in Extended Data Fig. 2b.

To compare the *cis* (within chromosome) and *trans* (between chromosomes) effects of gene coexpression per ACD pair, we calculated chromosome-wise differential coexpression coefficients by binning differential coexpression coefficients ($\Delta C_{RNA}(i,j)$) in *cis* or within one chromosome pair in *trans*. The results were plotted in a heat map or bar plot (the bar for *trans* effects is an averaged value) in Extended Data Fig. 3c,d.

## Cell cycle phase classification based on Smart-SCRB data
To dissect the effect of cell cycle stages on gene coexpression analysis, we adopted a computational method described by Scialdone et al. for classifying cells into cell cycle phases based on Smart-SCRB data[56]. Using a reference dataset, the difference in expression between each pair of genes was computed. Pairs with significant changes across cell cycle phases were selected as markers for classification and applied to a test dataset. Cells were then classified into the appropriate phase based on whether the observed change for each marker pair was consistent with one phase or another. This approach was implemented in the cyclone() function from the 'scran' package, which contains a pretrained set of marker pairs (mouse_cycle_markers.rds) for mouse cells.

## scATAC-seq data acquisition
Mouse ES cells treated with or without auxin were washed and resuspended in 1× PBS with 0.04% bovine serum albumin. Nuclei isolation for scATAC-seq from cell suspensions was performed according to the manufacturer's demonstrated protocol (CG000169, Rev E, 10x Genomics). Nuclei were counted using a Luna II automated cell counter

(Logos Biosystems). Approximately 15,000 nuclei per sample were loaded and subjected to a Chromium NextGem scATAC-seq v2 assay (10x Genomics). The resulting libraries were sequenced on a NextSeq 2000 (Illumina; 50 bp read 1, 8 bp i7 index read, 16 bp i5 index read and 50 bp read 2).

FASTQ files of raw data were processed by using the Cell-Ranger ATAC (10x Genomics, v2.1.0) analysis pipeline. Reads were filtered and aligned to mouse genome mm10 (10x Genomics, refdata-cellranger-arc-mm10-2020-A-2.0.0) using the cellranger-atac count() function with default parameters. The barcoded and aligned fragment files were then loaded by ArchR (version 1.0.1). Low-quality cells with minimum transcription start site enrichment scores of less than 4 and minimum fragment numbers of less than 1,000 were filtered out. Doublets were inferred by the addDoubletScores() function and removed using the filterDoublets() function with default parameters.

## Cross-ACD chromatin coaccessibility analysis

To evaluate the chromatin coaccessibility per ACD pair, we binned ATAC counts within each ACD to derive an ACD counts matrix ($M_{ATAC}$) for each experimental condition. After adding ACD regions using the addPeak-Set() function, counts for each ACD per cell were aggregated together using addPeakMatrix() with a very high ceiling value (1,000). To correct batch effects, we downsampled (50%) the scATAC-seq data to match the distribution of reads per ACD count between experimental conditions, and ten randomly downsampled dataset pairs were used for the calculations below. The ACD counts matrix ($M_{ATAC}$) for each condition was normalized to the average value $\bar{M}_{ATAC}$ of the matrix to derive a normalized ACD counts matrix ($N_{ATAC}$). To mitigate the influence from poorly sequenced cells (have 0 count for many ACDs), each column $j$ of $N_{ATAC}$ with mean value $\bar{N}_{ATAC}(j) \geq \bar{M}_{ATAC}$ was selected and integrated to form a new matrix $N_{ATAC-Filtered}$ for Spearman's rank-order correlation evaluation. The final results from ten downsampled dataset pairs were pooled and averaged.

For each ACD pair, the Spearman correlation coefficient for chromatin coaccessibility was calculated by using Eq. (1). The Spearman correlation matrix $S_{ATAC}$ was determined by calculating the Spearman correlation coefficient for each ACD pair across the genome. The differential Spearman correlation matrix $\Delta S_{ATAC}$ was calculated by

$$\Delta S_{ATAC} = S_{ATAC,RAD21-} - S_{ATAC,Ctrl}, \qquad (4)$$

where $S_{ATAC,RAD21-}$ and $S_{ATAC,Ctrl}$ are Spearman correlation matrices for cohesin-depleted and control conditions, respectively. $S_{ATAC,RAD21-}$, $S_{ATAC,Ctrl}$ and $\Delta S_{ATAC}$ were plotted as heat maps. The workflow for evaluating the Spearman correlation coefficients for chromatin coaccessibility is illustrated in Extended Data Fig. 4a.

## smRNA-FISH

smRNA-FISH probe blends were designed through Stellaris RNA-FISH probe designer (Biosearch Technologies) and include 40–48 serial probes that target only the intron regions of selected genes. The probe pairs designed for gene pairs selected from neighbor TADs were labeled by Quasar 570 and Quasar 670, respectively, for two-color experiments. The commercially synthesized oligonucleotide probe blends were dissolved in 400 μl of TE buffer (10 mM Tris-HCl and 1 mM EDTA, pH 8.0) to create a probe stock of 12.5 μM.

smRNA-FISH experiments were performed according to the Stellaris RNA-FISH protocol for adherent cells provided by Biosearch Technologies. Specifically, cells were grown on 18-mm round number 1 cover glass (Warner Instruments, CS-18R) in a 12-well cell culture plate coated with human recombinant laminin 511 (BioLamina, LN511-0202), fixed in 1× PBS with 3.7% formaldehyde (Millipore Sigma, F8775-25ML) for 10 min and permeabilized in ice-cold 70% (vol/vol) ethanol for 2 h. The coverslips with cells were immersed in 100 μl of hybridization buffer (90 μl of Stellaris RNA-FISH Hybridization Buffer

(Biosearch Technologies, SMF-HB1-10) and 10 μl of deionized formamide (Millipore Sigma, S4117)) at a final probe concentration of 125 nM and placed into a humidified chamber. The assembled humidified chamber was incubated overnight at 37 °C before the sample coverslips were washed, co-stained with 5 ng ml$^{-1}$ DAPI (Sigma-Aldrich, D8417) and mounted onto slides for future imaging analysis.

## Chr 2 intron-FISH probe design, synthesis and amplification

For 208 active genes across mouse Chr 2, we selected non-overlapping 35-nucleotide (nt) probes with several constraints, including a maximum melting temperature of 100 °C, minimum melting temperature of 74 °C, secondary structure temperature of 76 °C, cross-hybridization temperature of 72 °C, 30–90% GC content, no more than six contiguous identical nucleotides and spaces of at least 2 nt between adjacent probes. Primary probes were screened for potential non-specific binding with Bowtie2 (−very-sensitive-local) against mm10 genome sequences. Probes with more than one binding site were filtered out. Thirty qualified probes per gene closest to the transcription start site were selected. Spacers of 3 nt (random sequence) were extended at the 5′ and 3′ ends of the 35-nt probe sets. Two readout sequences (15 nt) separated by a 2-nt spacer (random sequence) were added at both the 5′ and 3′ ends of the probe, respectively, for the potential of performing seqFISH experiments. Universal primer sequences were then attached at the 5′ and 3′ ends. The 5′ primer contains a T7 promoter. The total length of each probe is 147 nt. The oligonucleotide probe pool (6,180 probes) was purchased from Twist Bioscience.

For probe amplification, limited PCR cycles were used to amplify the designated probe sequences from the oligonucleotide complex pool with Kapa HiFi HotStart Polymerase (Roche, KK2502). The amplified PCR products were then purified using a Zymo DNA Clean and Concentrator kit (Zymo Research, D4014) according to the manufacturer's instructions. The PCR products were used as the template for in vitro transcription (New England Biolabs, E2040S) followed by reverse transcription (Thermo Fisher Scientific, EP7051) with the forward primer. After reverse transcription, the probes were subjected to uracil-specific excision reagent enzyme (New England Biolabs, N5505S) treatment for ~24 h at 37 °C. Probes were then alkaline hydrolyzed with 1 M NaOH at 65 °C for 15 min to degrade the RNA templates, followed by 1 M acetic acid neutralization. Next, to clean up the probes, we used ssDNA/RNA Clean & Concentrator (Zymo Research, D7011) before hybridization.

## Chr 2 intron-FISH

Chr 2 intron-FISH was performed following a revised seqFISH protocol as described by Shah et al.[32]. Mouse ES cells were plated on human recombinant laminin 511-coated coverslips (Electron Microscopy Sciences, 72196-25). Cells were then fixed using 4% formaldehyde (Thermo Fisher Scientific, 28908) in 1× PBS diluted in molecular biology-grade water (Corning, 46-000-CM) for 15 min at 20 °C, washed with 1× PBS a few times and incubated in 70% ethanol for about 3 h at room temperature.

The coverslips were then washed twice with 2× SSC. For primary probe hybridization, samples were incubated with primary Chr 2 intron probes for 30 h at 37 °C in 50% Hybridization Buffer (2× SSC, 50% (vol/vol) formamide (Thermo Fisher Scientific, AM9344) and 10% (wt/vol) dextran sulfate (Sigma-Aldrich, D8906) in molecular biology-grade water) and washed in 55% Wash Buffer (2× SSC, 55% (vol/vol) formamide and 0.1% Triton X-100 (Sigma-Aldrich, 93443)) for 30 min at room temperature, followed by washing in 2× SSC. Alexa Flour 647-coupled Imager probes (Integrated DNA Technologies) for the first round of hybridization were incubated for 20 min at 50 nM each at room temperature in 10% EC buffer (10% ethylene carbonate (Sigma-Aldrich, E26258), 2× SSC, 0.1 g ml$^{-1}$ dextran sulfate and 0.02 U ml$^{-1}$ SUPERase·In RNase Inhibitor (Thermo Fisher Scientific, AM2694)) and washed for 5 min at room temperature in 10% Wash Buffer (2× SSC, 10% (vol/vol) formamide and 0.1% Triton X-100), followed by a 1-min wash in 2× SSC.

Samples were then co-stained with 5 ng ml$^{-1}$ DAPI for 15 min and imaged in an antibleaching buffer (50 mM Tris-HCl (pH 8.0), 2× SSC, 3 mM Trolox (Sigma-Aldrich, 238813), 0.8% D-glucose (Sigma-Aldrich, G7528), 100-fold diluted catalase (Sigma-Aldrich, C3155), 0.5 mg ml$^{-1}$ glucose oxidase (Sigma-Aldrich, G2133) and 0.02 U ml$^{-1}$ SUPERase·In RNase Inhibitor).

### Intron-seqFISH data acquisition

Fifteen rounds of imaging were performed with transistor–transistor logic automated imaging and a fluidic system consisting of a Nikon CSU-W1 spinning disk microscope with a CFI Plan Apochromat ×60/1.42-NA oil objective lens and a spatial genomics fluidics pack from Elveflow. After primary probe hybridization, the 40-mm coverslip (number 1.5) with cells was mounted into a closed-top FCS2 chamber (Bioptechs) and subsequently loaded into a custom stage adaptor on the microscope. The flow rate was normally maintained at 100 µl min$^{-1}$. For each readout probe hybridization, cells were first washed with the Wash Buffer (2× SCC with DAPI) for 5 min, followed by 20 min of 10% EC buffer injection (10% ethylene carbonate (Sigma-Aldrich, E26258), 2× SSC, 0.1 g ml$^{-1}$ dextran sulfate (Sigma-Aldrich, D4911) and 0.02 U ml$^{-1}$ SUPERase In RNase Inhibitor (Invitrogen, AM2694); flow rate of 50 µl min$^{-1}$) and 10 min of hybridization (flow rate of 0 µl min$^{-1}$). Samples were then washed with 10% Wash Buffer (2× SCC, 10% formamide and 0.1% Triton X-100) for 10 min and then with Wash Buffer (2× SCC with DAPI) for 5 min. After these two washes, imaging buffer (50 mM Tris-HCl (pH 8.0), 300 mM NaCl, 2× SSC, 3 mM Trolox (Sigma, 238813), 0.8% D-glucose (Sigma, G7528), 100-fold diluted catalase (Sigma, C3155), 0.5 mg ml$^{-1}$ glucose oxidase (Sigma, G2133) and 0.02 U ml$^{-1}$ SUPERase·In RNase Inhibitor (Invitrogen, AM2694)) was flowed into the chamber for 10 min, followed by zero flow implementation with the T junction method recommended by Elveflow. Imaging acquisition was performed on a Nikon CSU-W1 equipped with the Uniformizer, five laser lines (405 nm/514 nm/561 nm/594 nm/640 nm) and a Hamamatsu BT fusion camera. The four image stacks were acquired sequentially from the 640-nm channel (Alexa Fluor 647N) to the 561-nm (Cy3), 488-nm (Alexa Fluor 488) and 405-nm (DAPI + blue beads) channels under ultraquiet mode with a fixed framerate (5.1 Hz). A total axial range of 20 µm with z steps of 300 nm were covered. To detect weaker single-molecule signals, four-frame averaging was used for 640-nm, 561-nm and 488-nm channels, and two-frame averaging was used for the DAPI channel. After imaging, cells were washed with probe stripping buffer (2× SCC, 55% formamide and 0.1% Triton X-100) for 10 min before starting the next round of hybridization. The z position of the objective between imaging rounds was maintained by Nikon's perfect focusing system.

### seqFISH cell-level gene decoding and data analysis

Three-dimensional single-molecule localizations were performed using FISH-Quant 2 (refs. 57,58) and fixed thresholds for each color channel. *xyz* drift corrections were performed based on localizations of blue beads in the 405-nm channel. Chromatic corrections were calculated based on 3D multicolor bead image stacks acquired using coverslips coated with tetraspeck beads. Localizations from different hybridization rounds and color channels were pooled after applying corresponding drift and chromatic corrections. Gene decoding was performed according to the code book (Supplementary Table 3) with a maximal *xyz* distance cutoff of 1 pixel between all readout rounds. Three-dimensional nucleus segmentation was performed by using Cellpose 2 (ref. 59) with a pretrained specialist's model optimized for our imaging condition. Decoded genes were parsed into single cells based on gene localizations and 3D masks in the field of view.

The Chr 2 intron-FISH data were screened for number (*N*) of cells. The number of bursting (*N*$_{\text{b}}$) events for each gene was counted over all the cells detected. The number of co-bursting (*N*$_{\text{co}}$) events over *N* cells for each gene pair was counted when a visible and approximated (Euclidean distance ≤ 1 µm) pair of intron-FISH localizations from two channels were identified. The bursting frequency (*F*) of each gene was computed by

$$F = N_{\text{b}}/N. \tag{5}$$

The normalized co-bursting frequency (*F*$_{\text{co}}$) for gene pairs (A and B) was computed by normalizing co-bursting frequency (*N*$_{\text{co}}$/*N*) to the product of the bursting frequencies of both genes

$$F_{\text{co}} = N_{\text{co}}/(F_{\text{A}} \times F_{\text{B}} \times N). \tag{6}$$

*F*$_{\text{A}}$ and *F*$_{\text{B}}$ are the bursting frequencies for A and B, respectively.

### Airyscan imaging and image analysis

Fluorescence images obtained from smRNA-FISH or Chr 2 intron-FISH were acquired on a Zeiss LSM880 inverted confocal microscope attached to an Airyscan 32 gallium arsenide phosphide-PMT area detector. Before imaging, the beam position was calibrated to center on the 32-detector array. Images were taken under Airyscan super-resolution mode by a Plan Apochromat ×63/1.40-NA oil objective in a lens immersion medium with a refractive index of 1.515. The Airyscan super-resolution technology used a very small pinhole (0.2 AU) at each of its 32 detector elements to increase the signal-to-noise ratio approximately 4- to 8-fold and enable ~1.7-fold improvement of resolution after linear deconvolution in both lateral (*x*, *y*) and axial (*z*) directions. DAPI, mEmerald, Quasar 570 (and JF549) and Quasar 670 signals were illuminated/detected at excitation/emission wavelengths of 405 nm/460 nm, 488 nm/510 nm, 561 nm/594 nm and 633 nm/654 nm, respectively. *z*-stacks were acquired with a step of 300 nm. After image acquisition, Airyscan images were processed and reconstructed using the provided algorithm from the Zeiss LSM880 platform.

Three-dimensional Airyscan image stacks were processed by using Imaris 7.2.3. As an initial step, we manually inspected the images and removed low-quality images based on the following criteria: (1) nonspecific signal outside of the DAPI-stained nuclei, (2) cropped signal at the edge of the images and (3) very faint signal.

To characterize and visualize MED6 and MED1 protein hubs, we used the Surfaces object in Imaris and ran the following algorithms: (1) apply 'Background Subtraction (Local Contrast)' mode and set 'Diameter of Largest Sphere' as 800 nm, (2) uniformly use a threshold value of 30 for MED6 and 10 for MED1 to segment protein hubs and (3) set the 'Minimal Number of Voxels' as 20 for filtering out noise signals. To characterize and visualize smRNA-FISH puncta, we used the Surfaces object from Imaris and and ran the following algorithms: (1) apply 'Background Subtraction (Local Contrast)' mode and set 'Diameter of Largest Sphere' as 2,000 nm, (2) uniformly use a threshold value of 35 for Quasar 570 and 70 for Quasar 670 to illustrate smRNA-FISH puncta and (3) set the 'Minimal Number of Voxels' as 50 for filtering out noise signals. To characterize the colocalization signal between Mediator hubs and smRNA-FISH puncta, we used the 'Colocalization' function in Imaris with a common threshold value of 100 to identify the overlapping volumes and reconstruct the 3D isosurfaces.

To measure the 3D distance between puncta of different smRNA-FISH signals, we localized the voxels corresponding to the local maximum of identified RNA-FISH signal using the Imaris 'Spots' function module and calculated the Euclidean distance by using the Measurement Points object with a voxel size of 43.6 nm × 43.6 nm × 300 nm. The potential drifts among different imaging channels were estimated by using 100-nm multispectral beads under the same acquisition settings and were considered in the calculation of Euclidean distance between smRNA-FISH foci.

The gene pair smRNA-FISH data were screened for number (*N*) of cells. The number of bursting (*N*$_{\text{b}}$) events for each gene was counted when a visible smRNA-FISH puncta was identified. The number of

co-bursting ($N_{co}$) events over $N$ cells for each gene pair was counted when a visible and approximated (Euclidean distance of <1 μm) pair of smRNA-FISH puncta from two channels were identified. The bursting frequency ($F$) of each gene and the normalized co-bursting frequency ($F_{co}$) were computed according to Eqs. (5) and (6), respectively.

## Three-dimensional ATAC-PALM, Mediator–HaloTag PALM and image analysis

We prepared the reagents for 3D ATAC-PALM experiments as described previously[21]. One day before the experiment, cells were plated on 5-mm coverslips (Warner Instruments, 64-0700) at a confluency of around 70–80% with proper coating. Cells were fixed with 3.7% formaldehyde (Millipore Sigma, F8775-25ML) for 10 min at room temperature. After fixation, cells were washed three times with 1× PBS for 5 min and permeabilized with ATAC lysis buffer (10 mM Tris-HCl (pH 7.4), 10 mM NaCl, 3 mM MgCl₂ and 0.1% Igepal CA-630) for 10 min at room temperature. After permeabilization, the sample coverslips were washed twice in 1× PBS, and the transposase mixture solution (1× Tagmentation buffer: 10 mM Tris-HCl (pH 7.6), 5 mM MgCl₂, 10% dimethylformamide and 100 nM Tn5-PA-JF549) was added to the sample. The coverslips were placed in a humidified chamber and incubated for 30 min at 37 °C. After the transposase reaction, the coverslips were washed three times in 1× PBS containing 0.01% SDS and 50 mM EDTA for 15 min at 55 °C before being mounted onto the lattice light-sheet microscope sample stage for imaging.

Three-dimensional ATAC-PALM data were acquired by lattice light-sheet microscopy at room temperature[60]. The light sheet was generated from the interference of highly parallel beams in a square lattice and dithered to create a uniform excitation sheet. The inner and outer numerical apertures of the excitation sheet were set to be 0.44 and 0.55, respectively. A variable-flow peristaltic pump (Fisher Scientific, 13-876-1) was used to connect a 2-l reservoir with the imaging chamber with 1× PBS circulating through at a constant flow rate. Labeled cells seeded on 5-mm coverslips were placed into the imaging chamber, and each imaging volume took 100-200 image frames, depending on the depth of the field of view. Specifically, spontaneously activated PA-JF549 dye was initially pushed into the fluorescent dark state through repeated photobleaching by scanning the whole imaging volume with a 2-W, 560-nm (or 640-nm) laser (MPB Communications). The samples were then imaged by iteratively photoactivating each plane with very-low-intensity 405-nm light (<0.05-mW power at the rear aperture of the excitation objective and 6 W cm⁻² power at the sample) for 8 ms and by alternatively exciting each plane with a 2-W, 560-nm laser and a 2-W, 640-nm laser at its full power (26-mW power at the rear aperture of the excitation objective and 3,466 W cm⁻² power at the sample) for an exposure time of 20 ms. The specimen was illuminated when laser light went through a custom 0.65-NA excitation objective (Special Optics), and the fluorescence generated within the specimen was collected by a detection objective (CFI Apo LWD water immersion ×25/1.1-NA, Nikon), filtered through a 440/521/607/700-nm BrightLine quad-band bandpass filter (Semrock) and N-BK7 Mounted Plano-Convex Round cylindrical lens ($f$ = 1,000 mm, Ø 1′, Thorlabs, LJ1516RM) and eventually recorded by an ORCA-Flash 4.0 sCMOS camera (Hamamatsu, C13440-20CU). The cells were imaged under sample scanning mode and the dithered light sheet at a step size of 500 nm, thereby capturing a volume of ~25 μm × 51 μm × (27-54) μm, considering a 32.8° angle between the excitation direction and the stage moving plane.

To precisely analyze the 3D ATAC-PALM data, we embedded nano-gold fiducials within the coverslips for drift correction as previously described[21]. ATAC-PALM images were taken to construct a 3D volume when the sample was moving along the '$s$' axis. Individual volumes per acquisition were automatically stored as TIFF stacks, which were then analyzed by in-house-developed scripts in MATLAB. The cylindrical lens introduced astigmatism in the detection path and recorded

each isolated single molecule with its ellipticity, thereby encoding the 3D position of each molecule relative to the microscope focal plane. All processing was performed by converting all dimensions to units of $x$–$y$ pixels, which were 100 nm × 100 nm after transformation due to the magnification of the detection objective and tube lens. We estimated the localization precision by calculating the standard deviation of all the localization coordinates ($x$, $y$ and $z$) after the nano-gold fiducial correction. The localization precision is 26 ± 3 nm and 53 ± 5 nm for $xy$ and $z$, respectively.

## Three-dimensional pair cross-correlation function

The 3D pair cross-correlation function $c_0(r)$ between localizations of molecule A and those of molecule B can be formulated as

$$c_0(r) = \frac{3V}{4\pi(3r^2 \times \Delta r + 3r \times \Delta r^2 + \Delta r^3)} \times \frac{1}{M \times N} \sum_{i=1}^{M} \sum_{j=1}^{N} \delta(r - r_{ij}). \quad (7)$$

$M$ is the total number of localizations for molecule A, and $N$ is the total number of localizations for molecule B. $\Delta r = 50$ nm is the binning width used in the analysis. The Dirac Delta function is defined by

$$\delta(r - r_{ij}) = \begin{cases} 1 & r - r_{ij} \leq \Delta r \\ 0 & r - r_{ij} > \Delta r \end{cases}, \quad (8)$$

where $r_{ij}$ represents the pairwise Euclidean distance between localization points $i$ and $j$. The normalized 3D pair autocorrelation function $C(r)$ was calculated by

$$C_{(r)} = \frac{c_0(r)}{c_r(r)}. \quad (9)$$

$c_r(r)$ refers to the pair cross-correlation function calculated from uniform distributions with the same localization density in the same volume as real data used.

## Single-molecule imaging

Single-molecule imaging experiments were performed as previously described[42,49] on a Nikon Eclipse TiE motorized inverted microscope equipped with a ×100/1.49-NA oil immersion objective lens (Nikon), four laser lines (405/488/561/642 nm), an automatic TIRF illuminator, a perfect focusing system, a tri-cam splitter, three EMCCDs (iXon Ultra 897, Andor) and Tokai Hit environmental control (humidity, 37 °C, 5% CO₂). Proper emission filters (Semrock) were switched in front of the cameras for GFP and JF549 emission, and a band mirror (405/488/561/633-nm BrightLine quad-band bandpass filter, Semrock) was used to reflect the laser into the objective.

To perform single-molecule imaging of transcription factors and cofactors, cells were seeded on 25-mm number 1.5 coverglass precleaned with potassium hydroxide and ethanol and coated with human recombinant laminin 511 according to manufacturer's instructions. Live-cell imaging experiments were conducted by culturing mouse ES cells in imaging medium composed of FluoroBrite DMEM (Thermo Fisher Scientific, A1896701), 10% fetal bovine serum, 1× GlutaMAX, 1× non-essential amino acids, 0.1 mM β-mercaptoethanol and 1,000 U ml⁻¹ leukemia inhibitory factor. The TIRF illuminator was adjusted to deliver a highly inclined laser beam to the cover glass with the incident angle smaller than the critical angle. Oblique illumination (HILO) has much less out-of-focus excitation than regular epi-illumination. Transcription factors and cofactors linked to HaloTag were labeled with 5 nM HaloTag ligand-JF549 for 15 min and imaged using a 561-nm laser with an excitation intensity of ~50 W cm⁻². To minimize drift, the imaging experiments were performed in an ultraclear room with a precise temperature control system. The environment control chamber for cell culturing was thermoequilibrated. The imaging system was calibrated with beads to confirm a minimal drift during imaging ($x$–$y$ drift < 100 nm h⁻¹).

For fast-molecule tracking and throughout the experiments, we used an imaging acquisition time of 10 ms and took 5,000 continuous frames per imaging view after photobleaching of saturatedly labeled single molecules at the beginning. About 10-15 views were imaged for each labeled transcription factor/cofactor under normal or cohesin-depleted conditions. By contrast, for evaluating the RoC, which captures the motility of stably bound molecules, we used a longer acquisition time of 50 ms and took 5,000 continuous frames per imaging view.

### Single-particle tracking analysis

Each imaging view was recorded as a TIFF stack, and single molecules were tracked using SLIMfast, a custom-written MATLAB implementation of the MTT algorithm[61]. Frame-to-frame motions are defined by the distance between consecutive positions of the particle and can be potentially related to (1) Brownian (random walk) or confined motions of the molecule or (2) potential artifactual effects such as imperceptible movements of the nucleus. To filter out the effects resulting from point 1, it is thus necessary to define a maximal expected diffusion coefficient ($D_{Max}$), which defines the maximal distance ($d_m$) between two consecutive frames for a particle to be considered as the same object. As in the previous publication[42], a cutoff was set to 3 $d_m$ to ensure a 99% confidence level, and $D_{Max}$ was set as 1 $\mu m^2 s^{-1}$.

For each imaging view, SLIMfast generated a .txt output file consisting of a series of successive $x/y$ coordinates and times of detection corresponding to the displacement of each individual molecule. The output SLIMfast.txt files included the following information: $x/y$ coordinates (two-dimensional coordinates of the molecule in micrometers), trajectory index (ID number of the trajectory) and frame number (the index of the frame on which each single molecule was detected). These track files were used as the inputs to perform two-state (for histone subunits) or three-state (for transcriptional regulators) kinetic fitting by using Spot-On[45] software to compute the biophysical parameter values of single particles.

The RoC represents the circle best encompassing the motion track rather than encompassing it strictly. Thus, the measurement of the RoC is largely independent of the track duration. Tracks with lengths of <5 frames were discarded in the preprocessing step. To quantify the RoC, the mean square displacement (MSD) curves of each track were fitted using the nonlinear least-squares approach in MATLAB with a circle confined diffusion model[62] as illustrated in the following equation:

$$\text{MSD}_{circle} = \text{RoC}^2 \times \left(1 - e^{\frac{-4 \times D \times t_{lag}}{\text{RoC}^2}}\right) + \text{offset}. \quad (10)$$

The fitting provided values for RoC, the diffusion coefficient at short timescales $D$ and a constant offset value due to the localization precision limit, which is inherent to all the localization-based microscopy methods. To discard fitting errors related to artifacts such as erroneously connected jumps, we have discarded the trajectories with squared norm of the residual higher than $10^{-5}$ and a RoC higher than 500 nm.

### Statistics and reproducibility

Unless specified, data are presented as mean ± s.d. We normally applied two-sided Student's $t$-tests for measurements of technical replicates among different conditions but applied a non-parametric two-sided Wilcoxon rank-sum test for data that clearly do not follow normal distribution. No statistical method was used to predetermine sample size, but our sample sizes are similar to those reported in previous publications[18,25]. No data were excluded from the analyses. The experiments were not randomized, and the investigators were not blinded to allocation during experiments and outcome assessment. For results shown for representative experiments, each measurement was repeated three times independently with similar results.

### Reporting summary

Further information on research design is available in the Nature Portfolio Reporting Summary linked to this article.

## Data availability

Raw single-cell sequencing data were processed by aligning with the mouse genome mm10. The raw and processed data are available through Gene Expression Omnibus accession codes GSE264649 (scRNA-seq) and GSE266089 (scATAC-seq). Merged counts tables generated from scRNA-seq experiments need to be downloaded and arranged according to their associated annotation files to replicate gene coexpression patterns, while ATAC peak files need to be downloaded and arranged to replicate chromatin coaccessibility patterns. Raw image data from Airyscan imaging, intron-seqFISH, 3D ATAC-PALM and single-particle tracking experiments, due to their massive volumes, were not deposited but will be available upon request. We have deposited associated image quantifications (processed seqFISH, 3D ATAC-PALM and single-particle tracking data) and flow cytometry data at Zenodo at https://doi.org/10.5281/zenodo.11406939 (ref. 63). For analysis of Extended Data Fig. 1d, our identified ACDs were aligned with LADs, compartments and ChromHMM-identified genomic regions of mouse ES cells downloaded from previous publications[26–28]. Source data are provided with this paper.

## Code availability

All custom MATLAB codes and relevant datasets have been deposited in Zenodo (https://doi.org/10.5281/zenodo.11406939) (ref. 63). The software (VVD-viewer) for spatial genome imaging data visualization is available via GitHub (https://github.com/takashi310/VVD_Viewer) and Zenodo (https://doi.org/10.5281/zenodo.11259043) (ref. 64).

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

## Acknowledgements

We thank the R. Tjian-X. Darzacq laboratory members for helpful discussions and constructive suggestions. We also thank C. Stringer and S. Preibisch for guidance on establishing Cellpose 2 nucleus segmentation and the single-molecule localization workflow, respectively. We also thank K. Schaefer from Cell and Molecular Biology Shared Resources in Janelia Research Campus for assistance in cell sorting and CytoFLEX flow cytometry experiments, M. DeSantis and D. Alcor from the Imaging Core Facility at Janelia

for microscopy training and M. Radcliff for administrative support. P.D., V.G., L.X., L.W., A.L.L., W.L. and Z.J.L. are funded by the HHMI. P.D. receives support from the National Key R&D Program of the Ministry of Science and Technology (2023YFF0715200), Chinese National Science Fund for Excellent Oversea Young Scholars and Shenzhen Medical Research Fund (B2302038). L.X. receives support from the Janelia Visitor Program. S.Z. and H.Y.C. are funded by HHMI and NIH grants (RM1-HG007735 and R35-CA209919). A.D.L. was supported by NIH grants (U54-CA217378 and R01-DE019638). This work was made possible, in part, by software funded by the NIH: Fluorender: An Imaging Tool for Visualization and Analysis of Confocal Data as Applied to Zebrafish Research (R01-GM098151-01). The funders had no role in study design, data collection and analysis, decision to publish or preparation of the manuscript. This article is subject to HHMI's Open Access to Publications policy. HHMI lab heads have previously granted a non-exclusive CC-BY-4.0 license to the public and a sublicensable license to HHMI in their research articles. Pursuant to those licenses, the author-accepted manuscript of this article can be made freely available under a CC-BY-4.0 license immediately upon publication.

## Author contributions

P.D. and Z.J.L. conceived and designed the study. P.D. constructed reagents/cell lines, performed imaging (3D ATAC-PALM, Airyscan and single-molecule tracking), biochemical experiments and data analysis. P.D. and S.Z. performed Smart-SCRB and scATAC-seq analyses. L.X. helped with cell line construction. L.W. and A.L.L. performed Smart-SCRB and scATAC-seq experiments of samples provided by P.D. A.D.L. conducted the gene–gene correlation analysis on Smart-SCRB data. P.D. and Z.J.L. conducted the intron-seqFISH experiments and data analysis. V.G. performed intron-FISH experiments for genes in *trans*. W.L. performed western blotting to characterize RAD21 levels before and after auxin treatment in different cell lines. The custom VVD-viewer for visualizing intron-seqFISH data was developed by H.O. and T.K. P.D. and Z.J.L. wrote the paper with input from the other authors. Z.J.L. and H.Y.C. supervised the study.

## Competing interests

H.Y.C. is a cofounder of Accent Therapeutics, Boundless Bio, Cartography Biosciences and Orbital Therapeutics and is an advisor for 10x Genomics, Arsenal Biosciences and Spring Discovery. The other authors declare no competing interests.

## Additional information

**Extended data** is available for this paper at https://doi.org/10.1038/s41588-024-01852-1.

**Correspondence and requests for materials** should be addressed to Peng Dong or Zhe J. Liu.

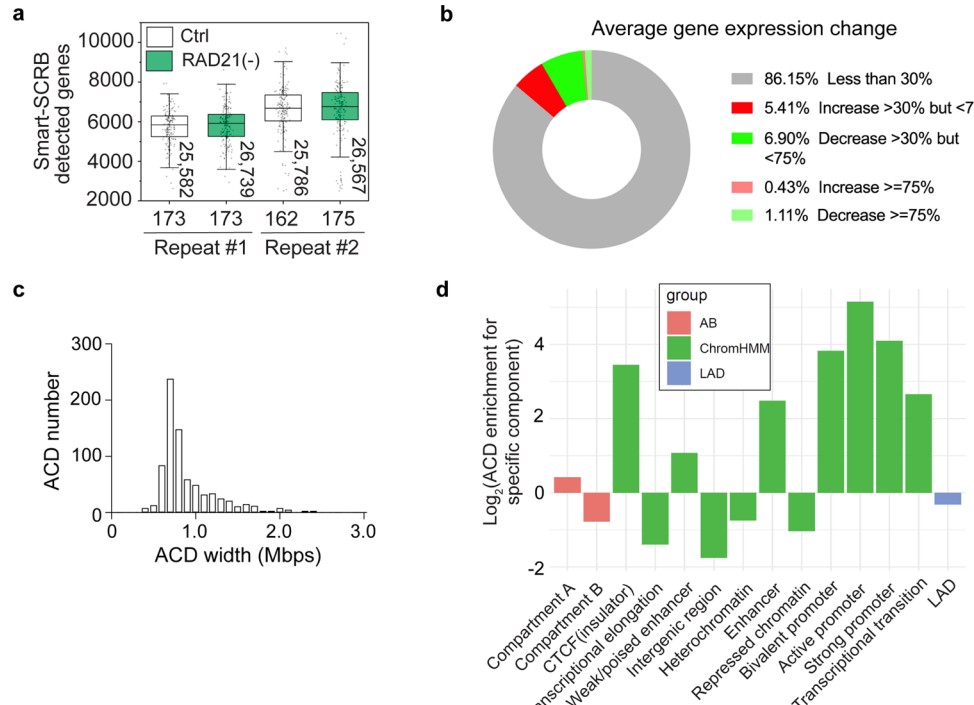

**Extended Data Fig. 1 | Single-cell transcriptome analysis upon cohesin loss.**
**a**. Box and dot plots of the number of detected genes per cell under control and cohesin-depletion conditions for two biological replicates by using Smart-SCRB technology. The total number of genes detected across the population is annotated vertically. Numbers at the bottom of the chart represent the numbers of single cells analyzed for each condition. For all box charts, upper and lower whiskers represent outlier cut-offs based on the 1.5 interquartile range rule; the box represents the range from 25% to 75% percentile; the center line represents the median. **b**. The pie plot shows the degrees of global gene expression changes from pooled Smart-SCRB data. **c**. The size distribution of 776 ACDs across the mouse genome. **d**. ACDs overlap with active compartments, enhancers and promoters, but are more likely to exclude with inactive compartments, intergenic regions, repressed chromatin, heterochromatin and LADs. The ACD enrichment for each specific element calculated by counting the ratio of ACDs overlapped with that element throughout the genome. Our identified ACDs were aligned with LADs, compartments and ChromHMM-identified genomic regions of mESCs downloaded from previously publications[26–28].

**a**

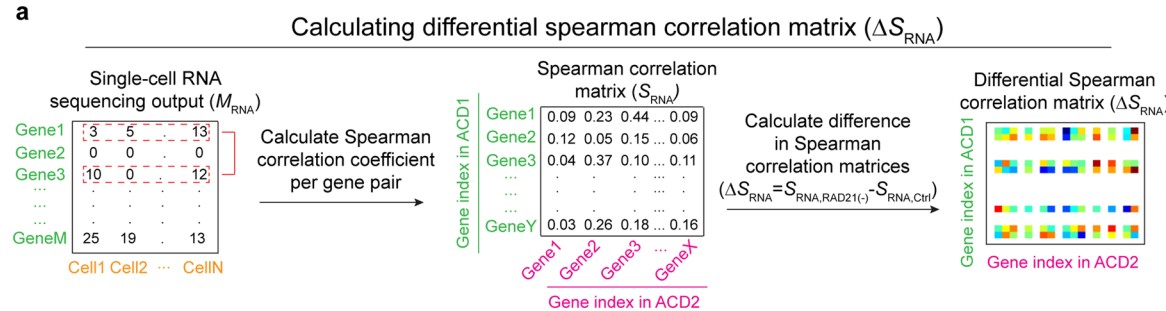

**b**

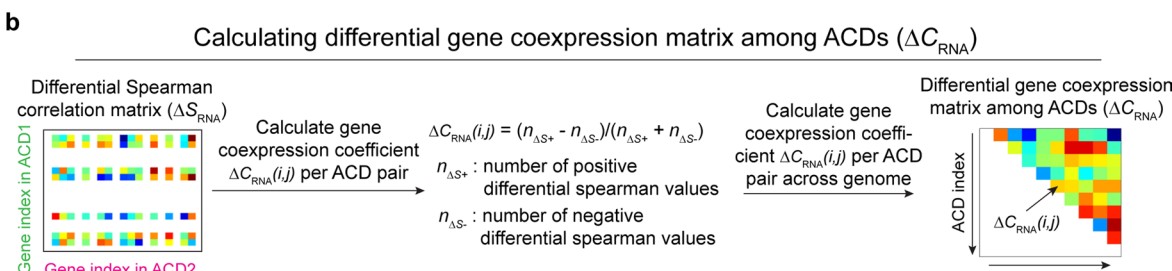

**Extended Data Fig. 2 | Quantification of cross-domain gene co-expression.**
**a**. The workflow for calculating differential Spearman correlation matrix ($\Delta S_{RNA}$) per gene pair (from two ACDs) before and after cohesin depletion from single-cell RNA-seq count matrix ($M_{RNA}$). The colormaps included are pure cartoon representations. **b**. The workflow for calculating differential gene co-expression matrix ($\Delta C_{RNA}$) per ACD pair before and after cohesin depletion from $\Delta S$ in (**a**).

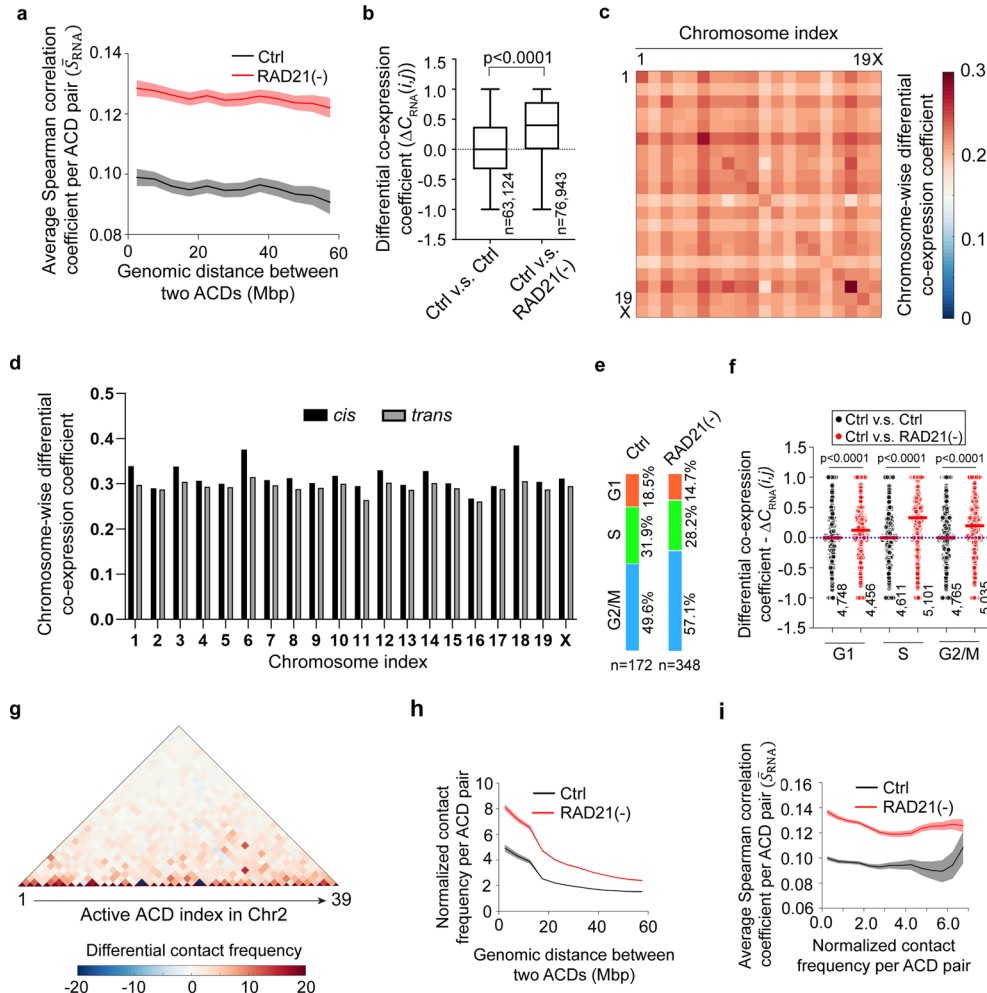

**Extended Data Fig. 3 | Cell cycle and Hi-C correlation analysis of cross-domain gene co-expression. a.** Acute cohesin removal increases the average Spearman correlation coefficient ($\bar{S}_{RNA}$) per ACD pair globally. $\bar{S}_{RNA}$ under control and cohesin-depletion conditions was plotted as a function of the genomic distance between ACD pair after a five-point smoothing. Only the data from ACD pairs within the same chromosome were used to generate the plot. **b.** Box plots show the pooled statistics of differential co-expression coefficient ($\Delta C_{RNA}(i,j)$) per ACD pair throughout the whole genome after cohesin depletion. Differential co-expression coefficient calculated from two repeats with control ES cells was included as a control. The upper and lower whiskers represent maximum and minimum values; the box represents the range from 25% to 75% percentile; the center line represents the median; the dotted line indicates the zero-change line. **c.** Chromosome-wise differential co-expression coefficient calculated by binning differential co-expression coefficient ($\Delta C_{RNA}(i,j)$) in *cis* or in *trans*. **d.** Comparison of chromosome-wise differential co-expression coefficients in *cis* and in *trans*

across all chromosomes. **e.** Computational assignment of single cells into cell cycle phases by analyzing Smart-SCRB data with *cyclone()* (R-programmed cell-cycle phase classifier). **f.** Dot plots show the distributions of differential gene co-expression coefficients before and after cohesin depletion (red dots) for cells in G1, S and G2M cell cycle phases, respectively. The calculated coefficients between two independent control groups (black dots) were used as the control. Red lines indicate the median values, and the blue dotted line indicates the zero-change line. For **b** and **f**, two-sided non-parametric Wilcoxon test was used for statistical testing. **g.** Heatmaps show increased normalized differential Hi-C contact frequencies in Chr 2 after cohesin loss. **h.** Normalized contact frequency per ACD pair (from Hi-C data) as a function of genomic distance between that ACD pair. **i.** Average Spearman correlation coefficient ($\bar{S}_{RNA}$) per ACD pair as a function of normalized Hi-C contact frequency per ACD pair. The plot was generated by five-point smoothing. For **a, h** and **j**, data are presented as mean values ± S.E. and shadow regions indicate S.E. of the mean.

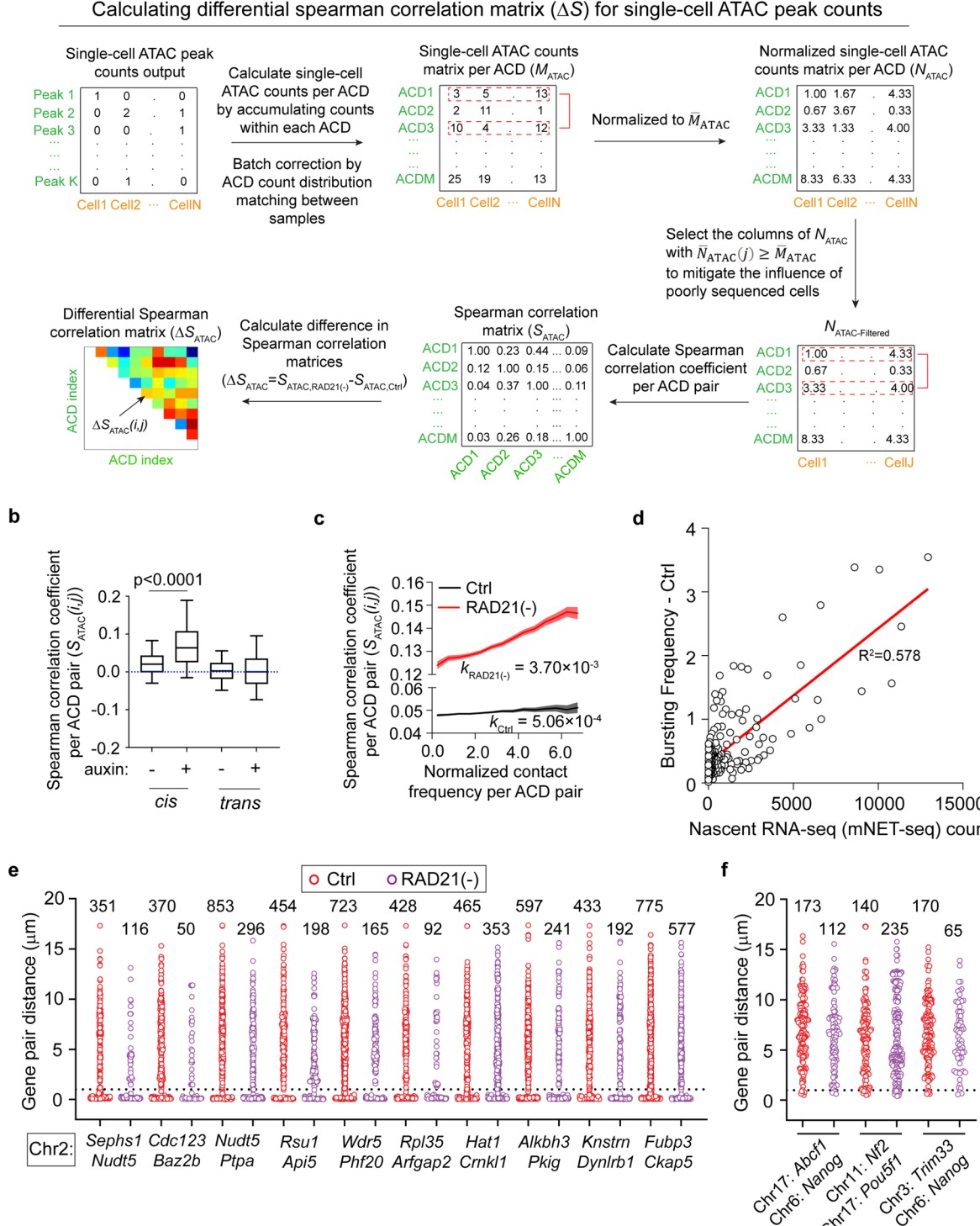

**Extended Data Fig. 4 | See next page for caption.**

**Extended Data Fig. 4 | Quantification of cross-domain chromatin co-accessibility and seqFISH validations. a**. The workflow for calculating differential Spearman correlation matrix ($\Delta S_{ATAC}$) per ACD pair before and after cohesin depletion from single-cell ATAC-seq count outputs. **b**. Box plots of co-accessibility Spearman correlation coefficients ($S_{ATAC}(i,j)$) per ACD pair in *cis* (within the same chromosome) or *trans* (from different chromosomes). In the box charts, lower and upper whiskers represent 5%-95% values; the box represents the range from 25% to 75% percentile; the center line represents the median. Dotted line indicates the zero-change line. Non-parametric two-sided Wilcoxon test was used for statistical testing. 15,481 pairs in *cis* and 279,815 pairs in *trans* were used for statistical analysis. **c**. Co-accessibility Spearman correlation coefficient ($S_{ATAC}(i,j)$) per ACD pair as a function of normalized Hi-C contact frequency per ACD pair. The plot was generated by five-point smoothing. Data are presented as mean values ± S.E. and shadow regions indicate S.E. of the mean. The slope derived from linear regression of the curve for each condition was labelled below. **d**. Validating bursting frequencies measured by seqFISH using published nascent RNA-seq data[4]. 199 of 208 genes in Chr2 (probed by seqFISH experiment) with detected nascent RNA-seq counts were used for scatter plot. Each circle represents one gene and the red line is the linear regression line with $R^2$ = 0.578. **e**. Distances between spots of co-bursting gene pairs (10 randomly selected pairs from the pool of 208 active genes in Chr 2) in *cis*. Each circle represents the distance between one pair of co-bursting spots within a single cell. The dotted line indicates 1 μm cut-off. As shown from the results, there is a specific enrichment of co-bursting spots on the same chromosome with distance < 1 μm. **f**. Distances between co-bursting gene pairs in *trans*. The results only showed the uniform distribution within the range between 0 μm and 20 μm, without the concentrated fraction < 1 μm. For **e** and **f**, the number of data points (n) used for statistical analysis for each gene pair is marked on top of the corresponding dot plot.

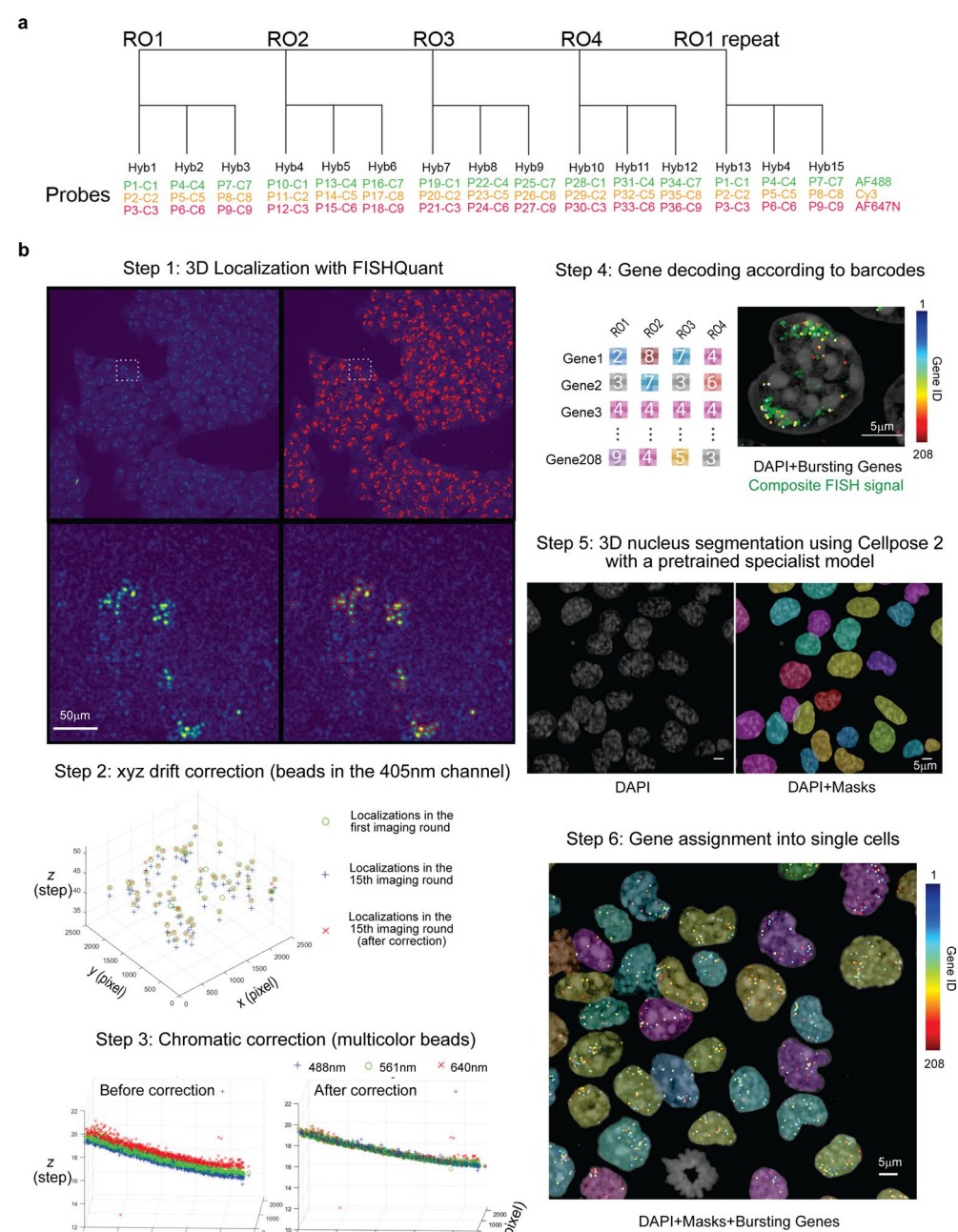

**Extended Data Fig. 5 | seqFISH workflow to spatially resolve actively transcribed genes in Chr 2. a.** Hybridization scheme: 4 readout rounds plus a repeat of the first readout round were performed. 3 hybridization per round with 9 distinct readout probes (P) and 9 pseudocolor (C) per round. In each hybridization, 4 sequential 3D acquisitions (640 nm, 561 nm, 488 nm, 405 nm) were performed to image intron seqFISH signals (AF488, Cy5 and AF647N) and nuclei (DAPI) with blue beads (405 nm). **b.** seqFISH data analysis and gene decoding pipeline: 1) 3D single-molecule localizations for 4 color channels were performed with FISHQUANT[5,63] for all color channels; 2) *xyz* drift correction based on localizations for beads in the 405 nm channel; 3) Chromatic correction based on multicolor Tetraspeck beads coated on the coverslip surface; 4) Pooling localizations and gene decoding according to predesigned barcodes; 5) 3D nucleus segmentation (right: random colored masks) based on DAPI stains (left) with a pretrained specialist model using Cellpose 2[7]; 6) Parsing genes into single cells based gene localizations and 3D masks for nuclei. The images were rendered by using VVD-viewer.

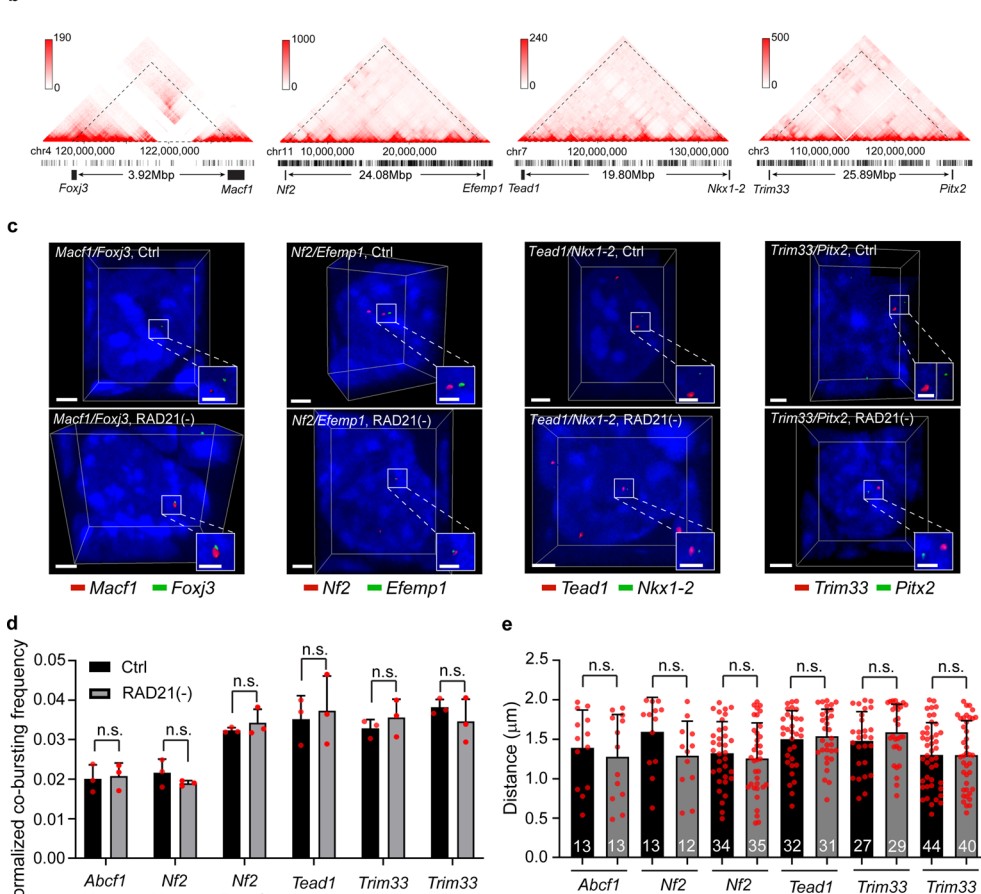

**a**

| Ectoderm lineage | Expression level (average counts) | | Mesendoderm lineage | Expression level (average counts) | | Chromosome | Distance (Mbps) |
|---|---|---|---|---|---|---|---|
| | Ctrl | RAD21(-) | | Ctrl | RAD21(-) | | |
| *Gbx2* | 2.50 | 2.74 | *Gpc1* | 0.36 | 0.32 | 1 | 2.92 |
| *Foxj3* | 1.88 | 1.96 | *Macf1* | 8.63 | 8.18 | 4 | 3.92 |
| *Nf2* | 8.77 | 7.82 | *Efemp1* | 0.36 | 0.34 | 11 | 24.08 |
| *Nkx1-2* | 1.94 | 2.02 | *Tead1* | 6.50 | 6.68 | 7 | 19.80 |
| *Trim33* | 1.91 | 1.89 | *Pitx2* | 1.67 | 2.89 | 3 | 25.89 |

**Extended Data Fig. 6 | Intron-FISH imaging of lineage-specific gene pairs in cis. a**. Information of lineage, genomic location and expression level for five pairs of developmental genes. The expression levels in both conditions were computed by averaging counts from Smart-SCRB measurements. **b**. Genomic positions of four pairs of representative lineage-specific genes with Hi-C and ATAC density information. The genomic distance between the two genes within each pair is labelled. **c**. Representative 3D *iso*-surface images of transcription bursting sites (intron-RNA-FISH) of gene pairs showed in (**a**) before and after cohesin loss. Scale bar, 2 μm. Inlet scale bar, 1 μm. **d**. The alteration of co-bursting

frequencies between co-bursting gene pairs in *trans* before and after cohesin depletion. Data are presented as mean values ± standard deviation (S.D.). The measurement was repeated three times and two-sided Student's t-test was used for statistical testing. n.s., not significant. **e**. The alteration of distances between co-bursting gene pairs in *trans* before and after cohesin depletion. Each dot represents one pair of co-bursting genes. Data are presented as mean values ± S.D. The number of data points (n) used for statistical analysis for each gene pair is marked at the bottom of the corresponding bar plot and non-parametric two-sided Wilcoxon test was used for statistical testing.

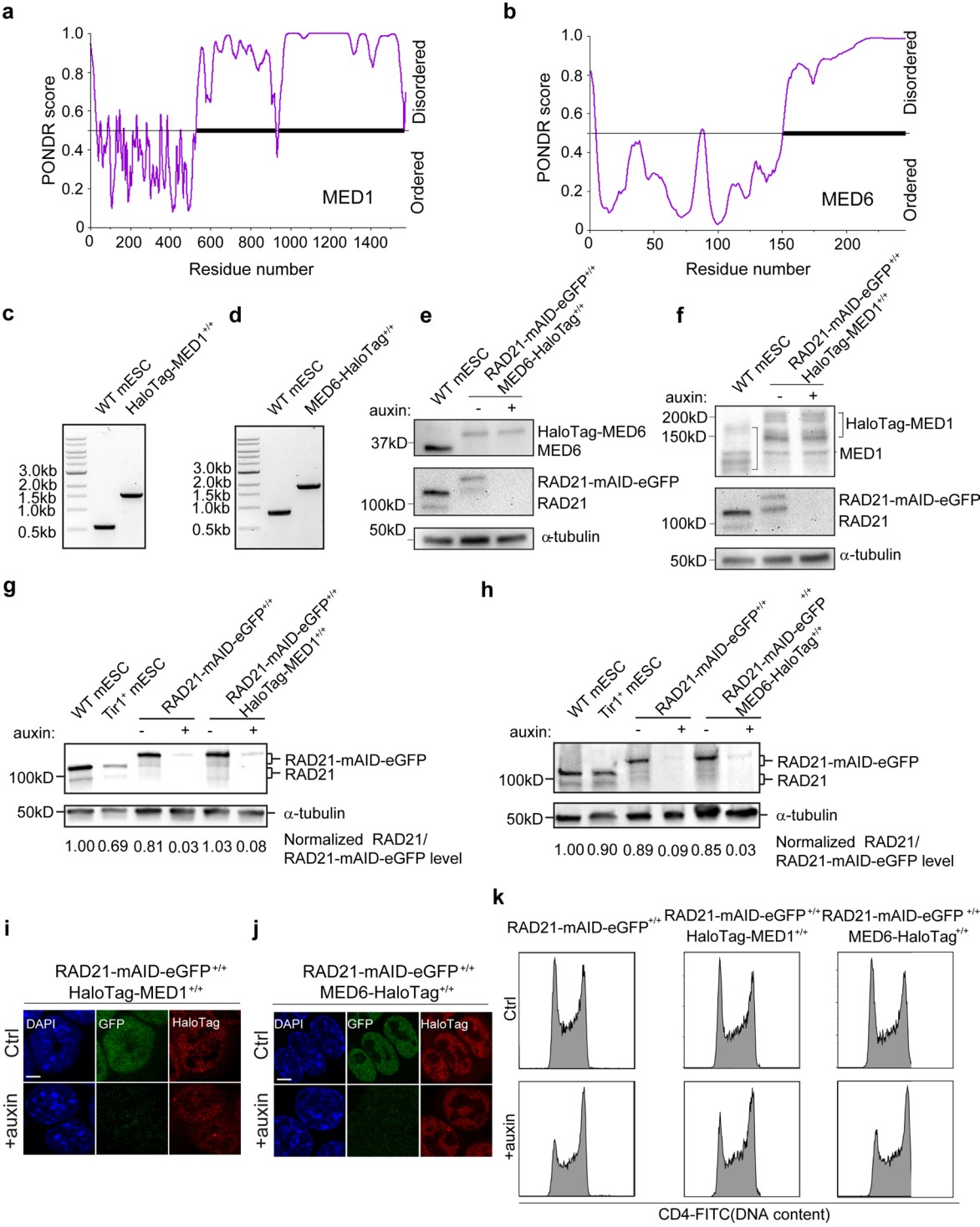

**Extended Data Fig. 7 | Tagging endogenous mediator subunits with HaloTag.**
**a-b.** PONDR score charts indicate predicted ordered and disordered regions within MED1 (**a**) and MED6 (**b**). **c-d.** PCR genotyping results showing bi-allelic fusion of HaloTag to MED1 c; N-terminus) and MED6 (**d**; C-terminus). Genomic DNA from wild-type mouse ES cells was used as the control. **e-f.** Western blots showing HaloTag-MED1 (**e**) and MED6-HaloTag (**f**) protein levels before and after RAD21 depletion by auxin-induced degron system. α-tubulin protein was blotted and used as a loading control. **g-h.** Western blots show the efficacy of RAD21 degron system in parental cell lines and established MED1 (**g**) and MED6 (**h**) knockin cell lines before and after auxin treatment for 6 hours. The normalized

RAD21 or RAD21-mAID-eGFP protein level to the loading α-tubulin level for each condition was quantified and shown below each lane. **i-j.** Fluorescence images showing RAD21-mAID-eGFP (Green) and HaloTag-MED1 (**i**; Red) or MED6-HaloTag (**j**; Red) levels without or with the auxin treatment (6 hrs). DNA was counter-stained with DAPI (Blue). Scale bar, 5 μm. For experiments from **c** to **j**, the measurement was repeated three times independently with similar results. **k.** Propidium iodide (PI) staining and flow cytometry analysis of DNA contents (CD4-FITC) from parental RAD21-mAID-eGFP cell line, and established MED1 and MED6 knockin cell lines before and after acute cohesin loss.

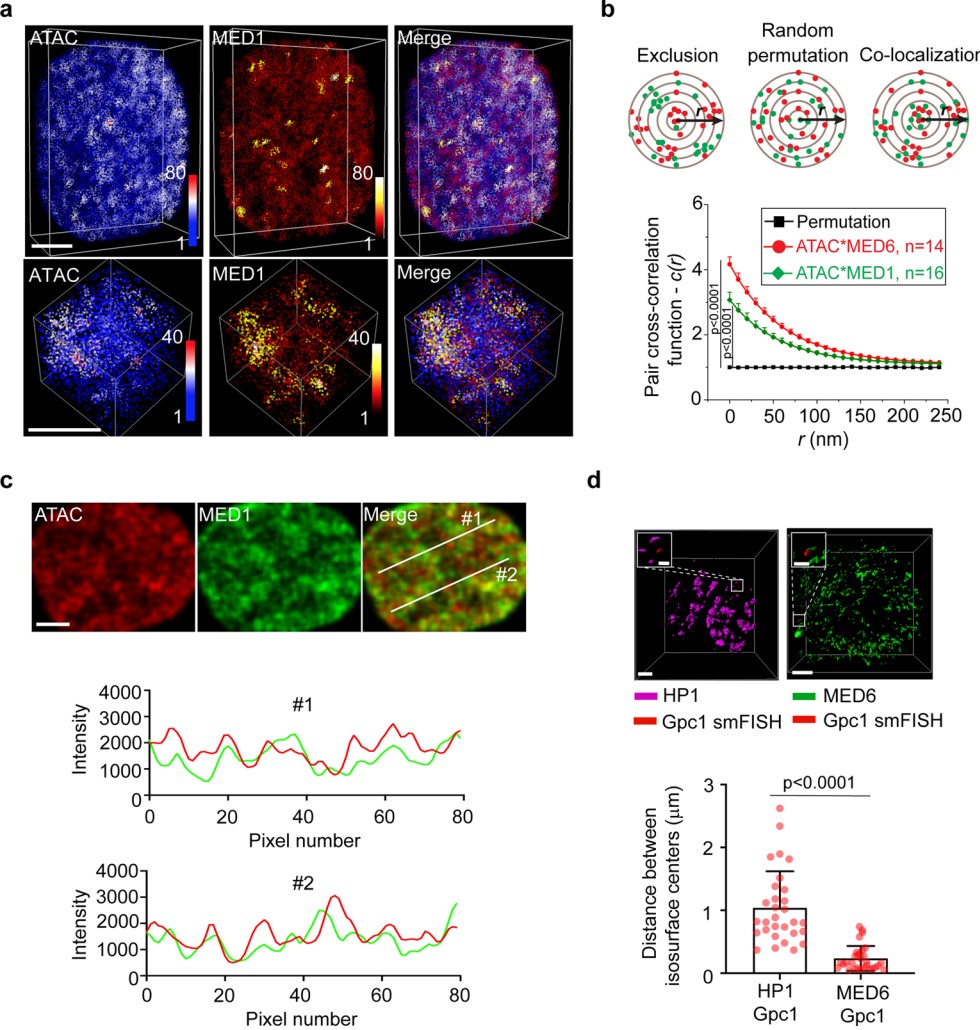

**Extended Data Fig. 8 | Mediator hubs colocalize with ACDs. a**. Two-color 3D PALM imaging captures spatial distribution of both accessible chromatin sites (ATAC) (left) and MED1-HaloTag (middle) localizations. Color bars indicate localization densities. Scale bar, 2 μm. See Movie S3 for 3D rotatory rendering. In the lower panel, the cropped localization map indicates that ACDs colocalize with MED1 hubs. Color bars indicate localization densities. Scale bar, 500 nm. **b**. Quantification of colocalization of accessible chromatin localizations and MED1 (or MED6) localizations by pair cross-correlation function c(r). In the upper panel, schematic shows three different spatial relationship – exclusion, uncorrelated (random permutation) and colocalization – between two localization maps. Data are presented as mean values ± S.E. The experiment was repeated for three times and non-parametric two-sided Wilcoxon test was used for statistical testing. **c**. One 2D section of two-color 3D ATAC and MED1 PALM images (**a**) was used for spatial intensity correlation analysis in **c**. The original 3D

localization maps were binned into 100 nm³ cubic to generate 3D image volumes for both channels and one slice was selected for colocalization analysis. Scale bar, 2 μm. One-dimensional intensity correlation analysis was performed for signals from two channels along selected line #1 and #2. The measurement was repeated three times independently with similar results. **d**. 3D reconstruction shows the overlap between Gpc1 intron-FISH *iso*-surface and MED6-HaloTag hub *iso*-surface and the separation between Gpc1 intron-FISH *iso*-surface and HP1-GFP *iso*-surface. Scale bar, 2 μm. Quantification of the physical distance between the centroid of Gpc1 intron-FISH signal and MED6 hub *iso*-surfaces and that between the centroid of Gpc1 intron-FISH intron-FISH singal and the nearest HP1 *iso*-surface. Non-parametric two-sided Wilcoxon test was used for statistical testing. The statistics were derived from 30 measured data points for each group. Data are presented as mean values ± S.D. in the bar plots.

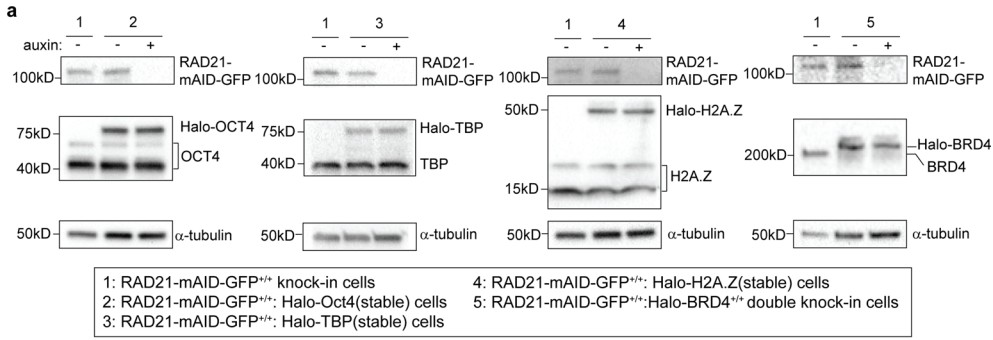

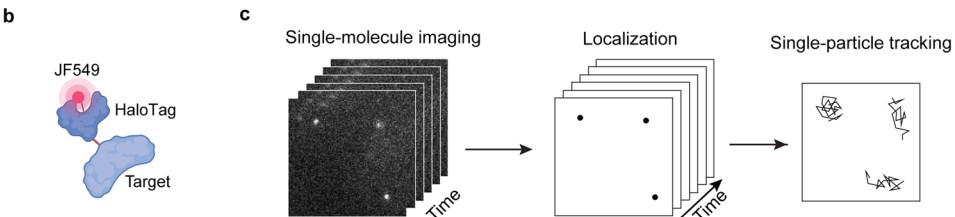

**Extended Data Fig. 9 | Single molecule tracking of transcription regulators and histone subunits with the HaloTag technology. a.** Western blots indicate HaloTag fusion protein levels for either stably expressed (OCT4, TBP1 and H2A.Z) or endogenously labelled (BRD4) transcriptional regulators before and after RAD21 depletion. α-tubulin was used as the loading control. The measurement was repeated three times independently with similar results. **b**. A diagram shows the labeling of HaloTag fusion proteins with JF549 dye. **c**. A schematic diagram illustrates the procedures for single-molecule imaging, localization, and tracking.

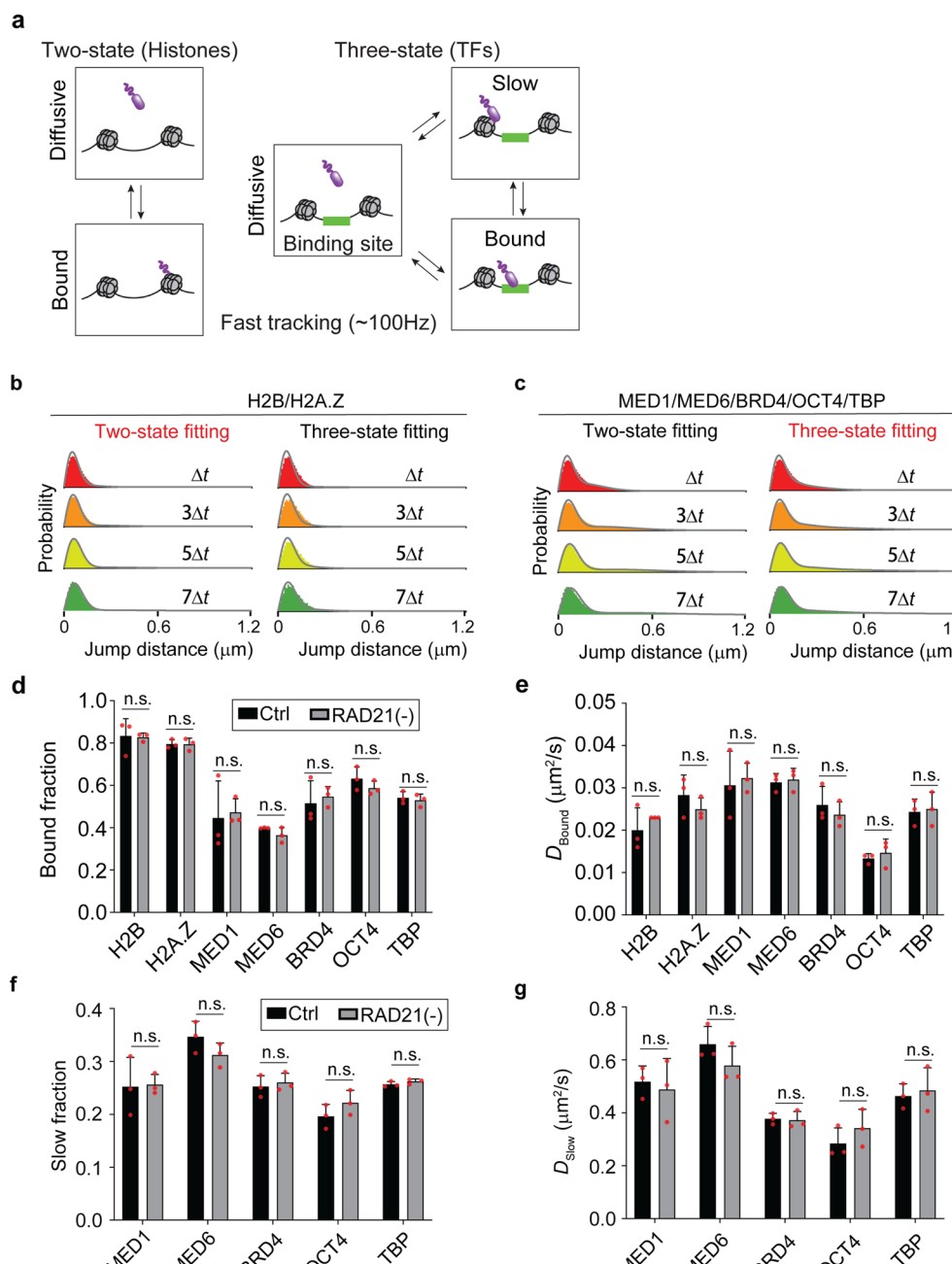

**Extended Data Fig. 10 | Fast tracking reveals no apparent changes in Histone and TF dynamics after cohesin loss. a.** The diagram shows two-state and three-state fitting of single-particle tracks for histone subunits and transcriptional regulators, respectively. For this analysis, fast single molecule imaging (100 Hz) was used to capture the movement of both diffusive and bound molecules. **b-c.** Representative fittings of 2 state and 3 state model to jump histograms of histone proteins (**b**) and diverse transcriptional regulators (**c**) with variable $\Delta t$. Two-state model assumes that TFs alternates between one bound and one diffusive state, whereas three-state model assumes that TFs alternates between one bound, one slow and one fast diffusive state. **d.** Fast tracking reveals that cohesin loss does not significantly alter apparent bound fractions for histone subunits (H2B and H2A.Z) and a broad range of transcriptional regulators (MED1, MED6, BRD4, OCT4 and TBP). Data are presented as mean values ± S.D.

The measurement was repeated for three times and two-sided Student's t-test was used for statistical testing. n.s., not significant. **e.** Fast tracking reveals cohesin loss has little effect on apparent diffusion coefficients for histone subunits (H2B and H2A.Z) and a broad range of transcriptional regulators (MED1, MED6, BRD4, OCT4 and TBP) in the bound state. Data are presented as mean values ± S.D. The measurement was repeated for three times and two-sided Student's t-test was used for statistical testing. **f.** Slow diffusive fractions for different transcriptional regulators before and after cohesin removal calculated by Spot-On. Data are presented as mean values ± S.D. The measurement was repeated for three times and two-sided Student's t-test was used for statistical testing. **g.** Diffusion coefficients for slow diffusive fractions as indicated in (**f**). Data are presented as mean values ± S.D. The measurement was repeated for three times and two-sided Student's t-test was used for statistical testing.

# Reporting Summary

## Statistics

For all statistical analyses, confirm that the following items are present in the figure legend, table legend, main text, or Methods section.

| n/a | Confirmed | |
|---|---|---|
| ☐ | ☒ | The exact sample size (*n*) for each experimental group/condition, given as a discrete number and unit of measurement |
| ☐ | ☒ | A statement on whether measurements were taken from distinct samples or whether the same sample was measured repeatedly |
| ☐ | ☒ | The statistical test(s) used AND whether they are one- or two-sided *Only common tests should be described solely by name; describe more complex techniques in the Methods section.* |
| ☐ | ☒ | A description of all covariates tested |
| ☐ | ☒ | A description of any assumptions or corrections, such as tests of normality and adjustment for multiple comparisons |
| ☐ | ☒ | A full description of the statistical parameters including central tendency (e.g. means) or other basic estimates (e.g. regression coefficient) AND variation (e.g. standard deviation) or associated estimates of uncertainty (e.g. confidence intervals) |
| ☐ | ☒ | For null hypothesis testing, the test statistic (e.g. *F*, *t*, *r*) with confidence intervals, effect sizes, degrees of freedom and *P* value noted *Give P values as exact values whenever suitable.* |
| ☒ | ☐ | For Bayesian analysis, information on the choice of priors and Markov chain Monte Carlo settings |
| ☒ | ☐ | For hierarchical and complex designs, identification of the appropriate level for tests and full reporting of outcomes |
| ☐ | ☒ | Estimates of effect sizes (e.g. Cohen's *d*, Pearson's *r*), indicating how they were calculated |

*Our web collection on statistics for biologists contains articles on many of the points above.*

## Software and code

Policy information about availability of computer code

| Data collection | All data collection software/codes are published |
|---|---|
| Data analysis | The visualization software (VVD-viewer) for spatial genome imaging data is available here: https://github.com/takashi310/VVD_Viewer |

For manuscripts utilizing custom algorithms or software that are central to the research but not yet described in published literature, software must be made available to editors and reviewers. We strongly encourage code deposition in a community repository (e.g. GitHub). See the Nature Portfolio guidelines for submitting code & software for further information.

## Data

Policy information about availability of data

All manuscripts must include a data availability statement. This statement should provide the following information, where applicable:

- Accession codes, unique identifiers, or web links for publicly available datasets
- A description of any restrictions on data availability
- For clinical datasets or third party data, please ensure that the statement adheres to our policy

Sequencing data that support the findings of this study will be deposited in NCBI-GEO database. Accession codes will be available before publication.

## Human research participants

Policy information about <u>studies involving human research participants and Sex and Gender in Research.</u>

| | |
|---|---|
| Reporting on sex and gender | N/A |
| Population characteristics | N/A |
| Recruitment | N/A |
| Ethics oversight | N/A |

Note that full information on the approval of the study protocol must also be provided in the manuscript.

## Field-specific reporting

Please select the one below that is the best fit for your research. If you are not sure, read the appropriate sections before making your selection.

☒ Life sciences    ☐ Behavioural & social sciences    ☐ Ecological, evolutionary & environmental sciences

For a reference copy of the document with all sections, see nature.com/documents/nr-reporting-summary-flat.pdf

## Life sciences study design

All studies must disclose on these points even when the disclosure is negative.

| | |
|---|---|
| Sample size | For genomic analysis, > 200,000 pairwise correlations of 776 ACDs were performed for each condition; for spatial genome imaging, >1000 cells were analyzed for each condition; for single molecule analysis, > 10,000 trajectories were analyzed for each condition. |
| Data exclusions | No data were excluded from this study. |
| Replication | High-hroughput sequencing data (scATAC-seq and Smart-SCRB) were repeated at least twice with biological replicates. All imaging data were repeated at least three times with biological replicates. |
| Randomization | Samples from different perturbation conditions were grouped and analyzed without a particular order. |
| Blinding | Data were processed with automated data analysis pipelines without human interference. No blinding is needed. |

## Reporting for specific materials, systems and methods

We require information from authors about some types of materials, experimental systems and methods used in many studies. Here, indicate whether each material, system or method listed is relevant to your study. If you are not sure if a list item applies to your research, read the appropriate section before selecting a response.

### Materials & experimental systems

| n/a | Involved in the study |
|---|---|
| ☐ | ☒ Antibodies |
| ☐ | ☒ Eukaryotic cell lines |
| ☒ | ☐ Palaeontology and archaeology |
| ☒ | ☐ Animals and other organisms |
| ☒ | ☐ Clinical data |
| ☒ | ☐ Dual use research of concern |

### Methods

| n/a | Involved in the study |
|---|---|
| ☒ | ☐ ChIP-seq |
| ☐ | ☒ Flow cytometry |
| ☒ | ☐ MRI-based neuroimaging |

## Antibodies

| | |
|---|---|
| Antibodies used | anti-RAD21(D213) rabbit polyclonal antibody (Cell Signaling, Cat. 4321);anti-MED1(CRSP1/TRAP220) rabbit polyclonal antibody (Bethyl Laboratories Inc, Cat. A300-793A), anti-MED6 Rabbit polyclonal antibody (Abcam, Cat. ab220110), anti-BRD4 (E8V7I) Rabbit monoclonal antibody (Cell Signaling, Cat. 54615), anti-TBP Rabbit polyclonal antibody (Cell Signaling, Cat. 8515), anti-OCT4 rabbit polyclonal antibody (Abcam, Cat. ab19857), Anti-Histone H2A.Z antibody [EPR18090] (Abcam, Cat. ab188314) and anti- -tubulin (11H10) rabbit monoclonal antibody (Cell Signaling, Cat. 2125). |

| Validation | anti-RAD21(D213) rabbit polyclonal antibody (Cell Signaling, Cat. 4321); application: Western blot; validation: knockout<br>anti-MED1(CRSP1/TRAP220) rabbit polyclonal antibody (Bethyl Laboratories Inc, Cat. A300-793A); application: Western blot; validation: HaloTag knock-in band shift<br>anti-MED6 Rabbit polyclonal antibody (Abcam, Cat. ab220110); application: Western blot; validation: HaloTag knock-in band shift<br>anti-BRD4 (E8V7I) Rabbit monoclonal antibody (Cell Signaling, Cat. 54615); application: Western blot; validation: HaloTag knock-in band shift<br>anti-TBP Rabbit polyclonal antibody (Cell Signaling, Cat. 8515); application: Western blot; validation: overexpression of HaloTag TBP band shift<br>anti-OCT4 rabbit polyclonal antibody (Abcam, Cat. ab19857); application: Western blot; validation: overexpression of HaloTag Oct4 band shift<br>anti-Histone H2A.Z antibody [EPR18090] (Abcam, Cat. ab188314); application: Western blot; validation: overexpression of HaloTag Oct4 band shift<br>anti-tubulin (11H10) rabbit monoclonal antibody (Cell Signaling, Cat. 2125). application: Western blot; validation; western blots |
| --- | --- |

## Eukaryotic cell lines

Policy information about cell lines and Sex and Gender in Research

| Cell line source(s) | JM8.N4 mouse embryonic stem cells (mESCs) |
| --- | --- |
| Authentication | 4DN Consortium Recommended Cell Lines; https://www.4dnucleome.org/cell-lines/ |
| Mycoplasma contamination | Negative for Mycoplasma contamination |
| Commonly misidentified lines<br>(See ICLAC register) | *Name any commonly misidentified cell lines used in the study and provide a rationale for their use.* |

## Flow Cytometry

### Plots

Confirm that:

☒ The axis labels state the marker and fluorochrome used (e.g. CD4-FITC).

☐ The axis scales are clearly visible. Include numbers along axes only for bottom left plot of group (a 'group' is an analysis of identical markers).

☐ All plots are contour plots with outliers or pseudocolor plots.

☐ A numerical value for number of cells or percentage (with statistics) is provided.

### Methodology

| Sample preparation | Cell cycle analysis was performed by propidium iodide (PI) staining following the protocols from the propidium iodide flow cytometry kit (Abcam, Cat. ab139418). |
| --- | --- |
| Instrument | Beckman Coulter CytoFLEX S with 4 lasers (405nm, 488nm, 561nm, 638nm) (Beckman Coulter) |
| Software | CytExpert Software v2.3 (Beckman Coulter) |
| Cell population abundance | All samples were acquired for 5 mins at a sampling rate of 30µl/min or up to 15,000 cells. |
| Gating strategy | SSC-A versus FSC-W was used for initial gating of singlet cells followed by SSC-A versus FSC-A to further define cellular events. An event count versus 561-610/20 histogram plot was used to determine the percentage of cells in G1, S or G2/M phases of the cell cycle. |

☐ Tick this box to confirm that a figure exemplifying the gating strategy is provided in the Supplementary Information.

