## [Peer Review File · Nature Genetics]

Peer Review Information

Manuscript Title: Cohesin prevents cross-domain gene co-activation

Corresponding author name(s): Dr Zhe (J) Liu, Dr Peng Dong

Editorial Notes:

Transferred manuscripts This manuscript has been previously reviewed at another journal. This document only contains reviewer comments, rebuttal and decision letters for versions considered at Nature Genetics.

Reviewer Comments & Decisions:

Decision Letter, initial version:

4th Apr 2023

Dear Zhe,

Your Article, entitled "Cohesin prevents cross-domain gene co-activation", has now been seen by 3 referees. You will see from their comments copied below that while they find your work of potential interest, they have raised quite substantial concerns that must be thoroughly addressed. In light of these comments, we cannot accept the manuscript for publication, but would be interested in considering a revised version that addresses these serious concerns.

Reviewer #1 thinks the findings are interesting, but would like you to perform additional analyses to further substantiate the conclusions.

Reviewer #2 also sees this work as a potentially valuable contribution to the field. They recommend a major revision since they have substantial technical concerns: e.g. important controls and methodological details are missing.

Reviewer #3 says the manuscript is impressive overall but they also raise several criticisms that warrant a major revision. Some of the concerns could potentially undermine the main conclusions of the paper, so they need to be carefully addressed.

In sum, the topic is important and this manuscript already contains a lot of data, but the reviewers (mainly #2 and #3) have major experimental/analytical requests.

We hope you will find the referees' comments useful as you decide how to proceed. If you wish to submit a substantially revised manuscript, please bear in mind that we will be reluctant to approach

the referees again in the absence of major revisions.

If you choose to revise your manuscript taking into account all reviewer comments, please highlight all changes in the manuscript text file. At this stage we will need you to upload a copy of the manuscript in MS Word .docx or similar editable format.

We are committed to providing a fair and constructive peer-review process. Do not hesitate to contact me if there are specific requests from the reviewers that you believe are technically impossible or unlikely to yield a meaningful outcome.

*2) If you have not done so already please begin to revise your manuscript so that it conforms to our Article format instructions, available here. Refer also to any guidelines provided in this letter.

Please be aware of our guidelines on digital image standards.

[redacted]

If you wish to submit a suitably revised manuscript we would hope to receive it within ~6 months. If you cannot send it within this time, please let us know. We will be happy to consider your revision so long as nothing similar has been accepted for publication at Nature Genetics or published elsewhere. Should your manuscript be substantially delayed without notifying us in advance and your article is eventually published, the received date would be that of the revised, not the original, version.

Nature Genetics is committed to improving transparency in authorship. As part of our efforts in this direction, we are now requesting that all authors identified as 'corresponding author' on published papers create and link their Open Researcher and Contributor Identifier (ORCID) with their account on

the Manuscript Tracking System (MTS), prior to acceptance. ORCID helps the scientific community achieve unambiguous attribution of all scholarly contributions. You can create and link your ORCID from the home page of the MTS by clicking on 'Modify my Springer Nature account'. For more information please visit www.springernature.com/orcid.

Thank you for the opportunity to review your work.

Sincerely,

Tiago

Tiago Faial, PhD
Chief Editor
Nature Genetics
<https://orcid.org/0000-0003-0864-1200>

Reviewers' Comments:

Reviewer #1:

Remarks to the Author:

Dong et al. present a manuscript in which they show that acute loss of cohesin leads to correlated expression of genes in active chromatin domains (ACDs). Using single-cell ATAC-seq data the authors show that this is associated with co-accessibility in ACDs as well. Using single-cell seq-FISH experiments measuring intron expression levels the authors determine bursting frequencies and show that co-bursting frequencies are increased while distances are decreased. Using fluorescence microscopy of tagged chromatin proteins the authors show that Mediator subunits MED1 and MED6 form hubs of increased size following RAD21 depletion, offering an explanation why increased co-bursting is observed.

The results are interesting and offer a provocative new role for cohesin and loop extrusion in the regulation of genes. Unfortunately, I felt that the results are very preliminary, which is caused mostly by the superficial analysis of the data. Also, there is a lack of clear examples where co-bursting occurs in response to cohesin depletion. Much of the claims are a consequence of the interpretation of abstract correlation values.

To take away some of the concerns that their observations are a direct consequence of a loss of loop extrusion and increased compartmentalization the authors should perform the following analyses:

1. They should perform the analysis on the correlation of RNA levels that they have performed in cis also with ACDs between chromosomes. This way they can show that this is indeed a cis effect (and hence a consequence of loss of loop extrusion), rather than an indirect effect caused by cohesin depletion. An even more powerful way of showing this is to perform single-cell RNA-seq in F1 hybrid cells (e.g. 129/Cast) in which cohesin is depleted. The genetic differences would allow the assignment

of mRNAs to specific parental alleles. The expectation is that alleles on different homologs should not co-burst, compared to genes on the same homolog.

2. The authors should show that the effects that they are observing on co-bursting do not occur in trans, that this is specifically a cis-effect. They should also perform the seq-FISH co-bursting analysis shown in Figure 3 with a probe set that covers two chromosomes, rather than one. This way they should be able to show that the effect of co-bursting occur exclusively in cis, which is their entire thesis.

For a unit so central to all their analyses, the authors should make more of an effort to explain how they calculate the ACDs. They also need to compare the ACDs to LADs, A/B compartmentalization, enrichment of chromHMM categories, etc.

I have a conceptual concern that the authors need to properly address. The authors claim that upon loss of RAD21 a correlation between genes in ACDs is detected in the single cell RNA-seq data. RNA-seq measures the stable transcript pool. Changes in this transcript pool is limited as the authors indicate themselves. Previous studies have shown that changes in transcription (i.e. newly produced transcripts) can be quite large following acute protein depletion, but it can take some time for these differences to transpire to the stable transcript pool. The authors should make it (more) plausible that RAD21 depletion can lead to changes in the stable transcript pool that can actually lead to correlated expression of gene pairs.

The authors claim that ACDs and spots in the ATAC-PALM overlap with each other (e.g. see title of Ext Figure 8). However, it is unclear how the author come to this conclusion. If they want to make this claim they should perform ATAC-PALM combined with FISH (or some other method) to show that these regions are actually overlapping.

Fig. 3f: the authors need to better explain what the normalized contact frequency means. It is unclear why there is still a decay in the contact frequency if it normalized. Normally, normalization should get rid of this decay.

Reviewer #2:

Remarks to the Author:

The manuscript by Dong et al. titled "Cohesin prevents cross-domain gene coactivation" aims to clarify the debate over the role chromatin structure on gene regulation. Recent work from different labs revealed limited effect on transcription upon acute short-term depletion of chromatin structural proteins casting questions about what the role of cohesin and genome structure is for gene expression (Hsieh et al. 2022). In this work, as the population averages obtained by bulk sequencing methods for gene expression and chromatin accessibility can over mask differences in a single cell level; the authors instead make use of single-cell assays of transcriptome and chromatin accessibility to suggest a role for cohesin to hinder the co-expression tendencies of cis-linked genes.

Using a number of cutting-edge single-cell RNA and ATAC-seq methods and microscopy techniques including 3D-PALM and seqFISH, the authors presented evidence to support a role for cohesin in limiting stochastic co-expression between genes located in different chromatin domains on a chromosome. The authors dive deeper into the previous observation that accessible chromatin domains (ACDs) mix and merge upon cohesin loss and correlates it to transcriptional outcomes in

single cells. They demonstrate a great role for cohesin to limit co-expression tendencies in single cells and demonstrates that single cell methods can provide valuable insights into the chromatin structure - function relationship that has been limited by use of bulk methods. On the other hand, in its current form the study still falls short on experimental controls, data analysis and interpretations, and method descriptions, as detailed below. Further explanations are required about the questions raised on cell line generation and FISH gene selections. If the authors are able to adequately address these technical issues, this study would make a valuable contribution to the chromatin & 3D genome field.

Comments:

1. Experimental design. The control condition for the Auxin degron is sub-optimal. Current description suggests that untreated mESCs were used as a control throughout the study. This control, however, is insufficient for the authors to make the conclusions stated due to the possibility that the perturbations of mESCs by the buffer used to dissolve auxin or the degron-independent effects of auxin, that caused the observed changes. It is therefore critical to include additional controls in the study, including using wild type mESCs treated with auxin, and mESCs treated with buffer only. Such controls should be included in the single-cell RNA-seq, single-cell ATAC-seq and single molecular imaging data acquisition, to provide confidence about the increased cross ACD gene co-expression and co-accessibility in cohesin-depleted mESCs.
2. The scRNA-seq dataset should be analyzed further to provide biological explanations to some of the phenomenon mentioned. One example is that they observe that cohesin loss alters differential co-expression coefficient per ACD pair both positively and negatively with the averages shifts toward the positive direction. It is important to elaborate on the ACD pairs that show negative diff co-expression coefficient upon cohesin loss and whether the location of the ACD pairs, the genomic distance or the genes that reside in them are relevant with this observation.
3. Are there regions transcribed that do not reside in the identified ACDs? If so, Is there an enrichment of co-expression only at the ACDs? A comparison using random or, non-ACD regions on the same chromosome would be beneficial to see (a) whether the co-expression effect is specific to ACDs, or the accuracy of ACD calling in the event of no-transcription elsewhere.
4. Please comment on whether the gene-co expression or expression changes in a single ACD upon cohesin loss. If so, does it have a correlation with the size of the ACD? What is the relationship between TADs and ACDs?
5. The definition of ACD is vague. Does it consider the uneven distribution of genes in the genome? If not, it is possible that the ACDs might not fully correspond to the ACD in the nuclear space.
6. Does the gene expression variance in between cells change upon cohesin loss in between single cells? Do you observe a variance change in a specific set of genes? Do you observe a co-variance change at ACDs after Rad21 depletion?
7. Previous work from the authors alluded to BRD2 mixing the accessible chromatin upon cohesin loss, and that acute loss of BRD2 reverses the spatial mixing of ACDs upon cohesin loss (Xie, Dong et al. 2022). Could the authors comment whether this system can reverse the co-activation of cis-linked genes that they observe with SMART-SCRB (Rad21, Brd2 co-depletion)?
8. Regarding FISH on Figure 3. Is there a correlation between differential distance and differential bursting frequency?
9. Figure 4e: Could you explain the reasoning behind the selection of gene couples? Why did the selection only include lineage specific genes? Is there any chance of seeing other sets of genes? Please provide gene expression levels of these lineage specific genes in both control and Rad21 depletion conditions with the scRNA-seq data.
10. Extended Figure 7: Concerns about Halo tagged Med1 and Med6 generation:

Is the WT mESC on first lane of western blots show no tagging of Rad21 with the degron, or is it the original Rad21 mAID line that is used to tag Med1? On top of Extended Figure 7e&f: it is written as Rad21-mAID-eGFP. Please explain /correct this. If it is WT cell in the first lane, please provide a lane with the Rad21 degron parental cell line (no depletion). Currently we do see a reduced Rad21 amount before depletion (lane 2 in e and f) and gives the impression that both Med6-Halo tag and Med1 halo tag cell lines demonstrate significantly lower levels of Rad21 even before the auxin loss. Not only that, but you also have lower levels of Med6 upon tagging with the Halo tag, is this an antibody issue? Does this reduced level of Med6 effect the cells in any way? Does it affect the size of ACDs? You could have confounding effects if both Rad21 and Med6 levels are reduced in this double tagged cell line specifically. Please also provide the cell cycle profile of the Rad21-degron parental line for extended figure 7i.

11. General notes: Please show Rad21 depletion by western blot for the time-points you have chosen for your experiments as they range from 3 hours (seqFISH) to 6 hours.

12. On Figure 5i, the authors show that 2 gene pairs that reside within the same ACD do not show any change in co-bursting fraction whereas gene pairs residing in different ACDs experience the increase upon Rad21 loss. I have concerns about the potential gene expression level differences effecting these results. Both Pou5f1(Oct4) and Abcf1 (Choi et al., Science Advances 2021) are crucial for ESC self-renewal and Cd9 is expressed at high levels in ESCs (Akutsu et al. 2009). Are the lineage specific genes expressed at similar levels to the other 4 (within ACD group) with similar variance? Could you please show whether one of the 4 genes (Cd9/ Mlf2, pou5f1/Abcf1) from the "within ACD" group shows increased co-bursting with another gene from another ACD on the same chromosome (and whether one of the lineage specific genes from different ACDs demonstrated to have a co-bursting fraction increase shows no change with a gene that is in the same ACD.

Minor comments:

1. Extended figure 1a. what are the numbers 173,173,171,169 correspond to?
2. The method for identifying ACD is insufficiently described, which makes it difficult to reproduce independently. Please provide more information, as other sections in the manuscript.
3. The description of the Rad21 degron mESC line is missing; how was this cell line constructed? If it was described previously, please provide a reference.
4. Line 244. "insulted" is a typo. Should be "insulated".

Reviewer #3:

Remarks to the Author:

Dong and colleagues present a tour de force manuscript with an impressive array of advanced imaging technologies applied to better understand how cohesin might contribute to control of gene expression -- beyond its limited acute effects on population average gene expression. They interpret the data to suggest that cohesin suppresses co-bursting of linearly distal genes by maintaining a more spatially segregated genome and counterbalancing aggregation of genes in cis and of TF clusters. Generally the manuscript is well written and the logic of each newly designed experiment and its interpretation are well explained. Several sections are a bit sparse on details in the main text, which seems to be a necessary but unfortunate consequence of packing so many complex and very diverse and bespoke experiment type into the same work. The corresponding figures (even with the ED figures) are also sparse on some instructive depictions of the data and some informative controls, as I describe below. I suspect several of these omissions were made in the interest of brevity and readability, but I believe

there are a few analyses worth adding to make the interpretation more convincing.

Primary Concern:

Mouse ESCs have rapid cell cycles and do not readily arrest in mitosis without cohesin, but proceed through mitosis with segregation defects. Segregation defects would skew the abundance of individual chromosomes, changing the number of templates and affecting RNA-seq levels, also changing the number of templates to provide ATAC signal from. As the cell loses or gains a whole chromosome, we would expect whole chromosome level correlated changes (i.e. cis-specific changes) in ATAC signal and in RNA expression. 6 hours of auxin depletion is quite long compared to the cell cycle time in mESCs (~8 hours, line-dependent), so a decent fraction of cells may well have been through an aberrant mitosis. At the population level, the relative gain or loss of a chromosome cell-to-cell would average out, but create variability at the single cell level. I am concerned that a number of the results in Figs 1 – 3 may be explained by this effect. In addition extranumerary foci in one of the intron FISH appear to suggest it. I discuss in my detailed comments below where I think this issue arises and what implications it has for the interpretation of the results, as well as some suggestions on how it may be best addressed.

Questions and suggestions pertaining to the main results (organized by figure):

Figure 1: It would be instructive for the authors to show more of the data, not just the few sparse quite downstream processing steps (as I describe in more detail in a moment). I appreciate this is a causality of staggering number of experiments that compress what could be a whole manuscript into a single figure. However, several key controls/comparisons would strengthen the claims of this section:

1) Fig 1e. What happens if you compute the C_RNA_Ctrl and C_RNA_Rad21 from S_RNA_Ctrl and S_RNA_RAD21- rather than from their difference? Is there generally a positive correlation in the gene expression from ACD to ACD in untreated cells? And then can you show this correlation gets weaker? The individual gene-to-gene correlations across ACDs seems to me to be a much noisier (sample size dependent) value to subtract. 400 cells is not a lot for looking at variation in gene expression genome wide.

2) Fig 1f. What do the trans correlations look like? I feel it would be instructive to create a map like Fig 2d, before jumping to the difference map, and it would be helpful to see the genome map (cis and trans) like Fig 2e, rather than just the single chromosome example. It would also improve the symmetry of the presentation.

3) Is there a cell cycle contribution driving the change in correlation? RAD21 depletion can have an effect on cell cycle. If one cluster is replicated, the cell must be in S/G2 and thus it is more likely other parts of the chromosome are replicated as well. If genes in multiple ACDs are replicated, there are 2 templates for each gene and on average one expects higher expression compared to if there is 1. Thus differences in cell cycle progression (primarily S phase for mESCs) in each sample may manifest as genome wide changes in correlated expression. This logic doesn't easily explain a cis/trans bias, but it is also not clear from the presentation of the RNA expression data that there is a cis bias, as only correlation within chromosomes is shown, unlike in Figure 2.

4) Are there chromosome abnormalities after this long depletion time in a rapidly cycling cell background? (We get defects if we do 6 hr auxin in mESCs with a cohesin degron in our hands). Might gaining an extra copy of the maternal or paternal for just 1 of the 23 chromosomes give the appearance of enhanced co-bursting purely due to the increased probability of observing expression for this 1 chromosome but not the others? The coburst would also appear to be concentrated in cis, as suggested by the data.

5) ED Fig 2 – I appreciated the explanation provided graphically in this figure. It might help if the colormaps had colorbars (though I appreciate the data appear to be purely cartoon representations to

understand the workflow rather than real data).

Figure 2, ATAC analysis

- 1) it would be helpful to state in the legend (and/or the text) how many cells were analyzed.
- 2) Might this data also be explained by the chromosomal abnormalities? An extra copy of just one chromosome would make it look like all the ATAC-accessible regions on that 1 chromosome were 2x more accessible (just due to increased template abundance). While the other chromosomes in that cell would all have WT levels of accessibility. This would look like a significant co-burst. Given the noise of the scATAC data at weak peaks, the 2x effect won't be sensitively detected at all sites, but the authors could test for it still by aggregating and plotting all the reads per chromosome per cell. It would be straightforward analysis (no new experiments required at least) to plot a matrix of cells x chromosomes with the total reads per chromosome represented in the matrix values (e.g. as color). This pattern could either help support or refute suspicion that chromosome abnormalities drive this apparent increase in co-accessibility reported.

Figure 3 – co-bursting by sequential intron FISH:

- 1) extra chromosome copies would increase the probability of detecting a co-burst. As most genes lack intron signals in most fraction of the cells, having more copies improves the probability one of them is co-bursting. I'm concerned the co-bursting is associated with these mitotic defects mentioned in my primary concern. Do the authors see these with shorter auxin treatment? Or in a cell type that doesn't have such rapid cell cycles? (Ideally the authors could differentiate to post-mitotic cells and repeat the tests there. Though care should be taken as mESCs also lose chromosomes spontaneously on differentiation, especially if the passage number is high, as is often necessarily the case when a large number of edits have been made in the line).
- 2) What is the distance between co-bursting genes? The figure plots change in distance, but not the actual distance distributions. The extended data have a very helpful illustration of the workflow and a nice presentation of the chromatic correction data, but I didn't find actual distance distributions there either.
- 3) What are the actual burst-frequencies per gene? While not the main point of the paper, it would be a helpful validation to include the data in the supplement.
- 4) It would be a helpful control to validate the iFISH data and the successful demultiplexing by comparing the aggregate levels to another data set. For example the authors might compare the burst frequencies to the bulk RNA-seq. There are also publically available nascent RNA-seq datasets and previously published genome wide intron RNA FISH data sets the authors could validate against. These data are compared between the Cntrl and degnon but it would be helpful to validate the frequency range in the ctrl.
- 5) I did appreciate the more substantial number of cells analyzed in this figure and the clear labeling (N=1475 and 1121 for degnon and control).

Fig 4

- 1) Clustering of transcribed genes looked good.
- 2) Fig 4e. There appear to be 3 Gbx2 nascent foci in the RAD21 case, and the ~1um scale distance between the upper two is considerably larger than expected for paired sisters. Thus, to me this image indicates a possibility of an extra chromosome copy in the cohesin depleted condition. It should be noted that depending on signal-to-noise of the intron FISH, this could be stray signal. It is hard to tell with the 3D rendering, and hard to appreciate the quality of the intron FISH data without the controls discussed above.

Figure 5 - Mediator clustering analysis

1) The changes in Mediator organization in the example images are striking, and the quantification and statistical analysis appear to be performed to a high standard.

2) The authors' images of Med hubs look qualitatively quite radically different from previous publications describing Mediator hubs (Sabari...Young 2018), which show much sparser, much more punctate staining, in both live and fixed cells (also mESCs), for both Mediator and BRD4. Given the Liu lab's expertise with this sort of imaging, this reviewer is more inclined to believe the results presented here, but I'm curious if the authors could comment on the striking discrepancy.

3) "We found that Mediator hub fusion spatially correlates with clustering of actively transcribed genes in Chr 2 upon Cohesin loss"

It wasn't clear to me from the analysis presented that this was hub *fusion* more than an increase in Mediator. Do mediator levels increase? (e.g. by Western?) Is the total aggregate volume occupied by Mediator increased? (it appears so in the images, it doesn't look like fewer, bigger clusters so much as just more mediator. The authors might clarify this by examining the number of clusters as well and see if this goes down, as expected for fusing a large number of small clusters into a smaller number of large clusters.

It also wasn't clear to me if the increased overlap with chr2 intron signal was due to a greater fraction of the nuclear volume being now occupied by mediator with more larger clusters, or if hub fusion could indeed be implicated. The only quantification for this claim shows that there is a larger radii of the overlapped regions, which might result from expanded Med signal, or expanded intron chr2 signal. The assertion to me implies it is specifically the fusion correlates with the active transcription, which implies that once separate MED clusters have joined, and this has resulted in gene expression, which is not what is shown in the figure. It's a reasonable inference from the data, especially if the amount of mediator total is unchanged and the number of clusters has decreased.

4) It's not clear to me how or why the radius "normalization" is done, or why it is just "average radius" but the rest

5) Fig 5 I shows the fraction of co-bursting loci that are connected by Med6 hubs. If there's more Med total (more clusters, higher nuclear volume fraction) more loci will be connected. Clarifying the change (question 3) will help clarify this. These two cases have different interpretation, and it would be good to be clear if the mechanism is fusion of existing mediator into fewer larger clusters driving more co-bursting, or just more mediator. The text seems to suggest the former rather strongly but the data to me look more like the latter (even though I don't know why the total mediator would change, and I suspect the western blots will show it doesn't). Some image quantification here could help.

Figure 6

No major objections. These are difficult experiments and they appear to be conducted with care. The results look interesting, though I found the conceptual connection to the other 5 figures a little less clear. It is interesting the activating TFs change in a different way the histones but I'm not sure what the implication is.

Minor textual comments:

1) "transcription activation does not require physical proximity between enhancer and promoter" 19 - I would say this claim is not widely accepted - even Alexander and colleagues propose that a degree of physical proximity is required, despite proposing that ~300 nm separation is sufficient proximity, a

distance much larger than the physical diameter of many TF complexes or of a single cohesin.
 - The bigger issue with the claim is that the enhancer and promoter are not TF scale (10 nm) objects (they are stretches of DNA that frequently span 300 nm) and that the authors didn't label either the enhancer or promoter, they put 10 kb stretches of probes 5-7 kb distal, and each 5-10 kb stretch is a long flexible polymer that frequently spans 300 nm. This can be seen by the centroid-to-centroid distance in FISH probes spaced 5 kb apart (e.g. Mateo...Boettiger 2019 supp mat for sox2 in mESC, or Huang...Ren 2021 for sox2 in mESC), or in live imaging (see the "del-TAD" data in Gabriele...Hansen 2022). See PMID 33310227 and 36854248 for more details.

2) Smart-SCRIB refs 23,24 I am bit confused at the referencing here, maybe the authors could double check. 23 looks like "smart-seq2", which I haven't seen go by "Smart-SCRIB". And Ref 24 looks like the AID degenon system.

3) Final section before discussion: "Next, we analyzed local hopping of transcriptional regulators between clustered 223 binding sites in the nucleus – a process known as intersegment transfer (Fig. 6a). 224 Intersegment transfer has been proposed as a mechanism underlying long-distance enhancer-promoter communication 38,43-47"

- I find this a bit too liberal a summary of the these works. While indeed these works speculate on a action at a moderate (~300 nm) distance, I think it goes a bit far to say they all argue for hopping of TFs. I think some of these references hypothesize rather that a large droplet of trans-activators is bound on one side to the enhancer and that the physical size of the condensate allows molecules on the other side to activate promoters 100s of nanometers away – including the possibility that a condensation reaction, templated on the enhancer, expands like a growing crystal or dew-drop until it reaches the promoter. Given that most of these references appear partial to postulating it is a liquid state aggregate, diffusion is probably happening from edge to edge, but it just seems a bit of an overstatement to summarize all these studies in one as hypothesizing "a process known as intersegment transfer." – what fraction actually use that term or are postulating something sufficient close?

Author Rebuttal to Initial comments

Point-by-point response

Nature Genetics submission: NG-A61918-T

Cohesin prevents cross-domain gene co-activation

Reviewer #1:

Remarks to the Author:

Dong *et al.* present a manuscript in which they show that acute loss of cohesin leads to

correlated expression of genes in active chromatin domains (ACDs). Using single-cell ATAC-seq data the authors show that this is associated with co-accessibility in ACDs as well. Using single-cell seq-FISH experiments measuring intron expression levels the authors determine bursting frequencies and show that co-bursting frequencies are increased while distances are decreased. Using fluorescence microscopy of tagged chromatin proteins, the authors show that Mediator subunits MED1 and MED6 form hubs of increased size following RAD21 depletion, offering an explanation why increased co-bursting is observed.

The results are interesting and offer a provocative new role for cohesin and loop extrusion in the regulation of genes. Unfortunately, I felt that the results are very preliminary, which is caused mostly by the superficial analysis of the data. Also, there is a lack of clear examples where co-bursting occurs in response to cohesin depletion. Much of the claims are a consequence of the interpretation of abstract correlation values.

Summary: We thanks this reviewer for valuable suggestions to improve the manuscript. The main concern from this reviewer is whether acute cohesin loss would affect genes in *trans* the same way it does in *cis*. In the revised manuscript, we calculated and compared chromosome-wise differential co-expression coefficients both in *cis* and in *trans* and indeed found a *cis* bias. We also performed intron-FISH experiments to examine the effect of cohesin loss on genes in *trans* (Fig. R1). We found that co-bursting frequency does not change for genes in *trans*. See details in the point-to-point response below.

To take away some of the concerns that their observations are a direct consequence of a loss of loop extrusion and increased compartmentalization the authors should perform the following analyses:

1. They should perform the analysis on the correlation of RNA levels that they have performed in *cis* also with ACDs between chromosomes. This way they can show that this is indeed a *cis* effect (and hence a consequence of loss of loop extrusion), rather than an indirect effect caused by cohesin depletion. An even more powerful way of showing this is to perform single-cell RNA-seq in F1 hybrid cells (e.g. 129/Cast) in which cohesin is depleted. The genetic differences would allow the

assignment of mRNAs to specific parental alleles. The expectation is that alleles on different homologs should not co-burst, compared to genes on the same homolog.

We thank the reviewer for his/her suggestions. We have calculated chromosome-wise differential co-expression coefficient by binning differential co-expression coefficients ($\Delta C_{\text{RNA}(i,j)}$) within the same chromosome (*cis*) or within one chromosome pair (*trans*). We found that the pairwise correlations in *trans* also increase generally after cohesin depletion, but are weaker and less significant than those in *cis* (Fig. R1, a and b). It is notable that gene expression measured by RNA-seq is an accumulative effect of gene co-regulation, and thus the difference for gene co-expression between *cis* and *trans* is not as direct and significant as the difference for chromatin co-accessibility presented in Figure 2 for reflecting early regulatory changes. We have included these results in the updated manuscript (Extended Data Fig. 3, c and d).

To rigorously test the effect of cohesin loss in *trans*, we also performed intron-FISH to examine co-bursting frequencies of genes localized in different chromosomes. Specifically, we focused on the same pool of genes showed elevated co-activation in *cis* upon cohesin loss (Fig. 5e and Extended Data Fig. 6a) and examined co-bursting frequencies and bursting-site distances when these genes are paired *in trans*. We found that both parameters remain largely unchanged before and after cohesin depletion (Fig. R1, c and d). These results suggest that the observed increase in the correlation of RNA expression per ACD pair in *cis* after cohesin depletion is a predominant and specific effect. These results are also consistent with our scATAC-seq result showing that cohesin loss specifically increases chromatin co-accessibility in *cis* but not in *trans* (Fig. 2, d and e) and our previous reported result that spatial mixing of accessible chromatin upon cohesin loss primarily occurs in *cis* but not in *trans*¹. We have included this new intron-FISH result in the updated manuscript (Extended Data Fig. 6, d and e).

Figure R1. Analysis of correlation of gene expression in *cis* and in *trans*. (a) Chromosome-wise differential co-expression coefficient calculated by binning differential co-expression coefficient ($\Delta C_{\text{RNA}}(i,j)$) within the same chromosome (*cis*) or within one chromosome pair (*trans*). (b) Comparison of chromosome-wise differential co-expression coefficients in *cis* and in *trans* across all chromosomes. For each chromosome, chromosome-wise pair differential co-expression coefficients in *trans* were averaged and plotted as a bar to compare with *cis* effect. (c) The alteration of co-bursting frequencies between co-bursting gene pairs in *trans* before and after cohesin depletion. Error bars indicate standard deviation (S.D.). The measurement was repeated three times and Student's t-test was used for statistical testing. n.s.; not significant. (d) The alteration of distances between co-bursting gene pairs in *trans*. Error bars indicate standard deviation (S.D.). Each circle represents one pair of co-bursting genes. The experiment was repeated for three times and Wilcoxon test was used for statistical testing.

2. The authors should show that the effects that they are observing on co-bursting do not occur in *trans*, that this is specifically a *cis*-effect. They should also perform the

seq-FISH co-bursting analysis shown in Figure 3 with a probe set that covers two chromosomes, rather than one. This way they should be able to show that the effect of co-bursting occur exclusively in cis, which is their entire thesis.

We thank the reviewer for his/her comments and understand his/her concern. To address this problem, we performed intron-FISH experiments to examine the effect of cohesin loss on co-bursting frequency of 6 pairs of genes *in trans* (different chromosomes). We found that co-bursting frequency and distances between co-bursting gene pairs remain largely unchanged before and after cohesin depletion for all examined gene pairs. The results are illustrated in Figure R1 and included in the updated manuscript (Extended Data Fig. 6, d and e).

For a unit so central to all their analyses, the authors should make more of an effort to explain how they calculate the ACDs. They also need to compare the ACDs to LADs, A/B compartmentalization, enrichment of chromHMM categories, etc.

We thank the reviewer for his/her suggestions. We described the procedures for calculating ACDs in Methods part in the manuscript. Specifically, to identify ACDs across the genome, we used available bulk ATAC-seq data ² and a 200kb genomic distance to bin ATAC reads along each chromosome. The Matlab function `findpeaks()` was then called in “MinPeakProminence” mode and the value of “MinPeakProminence” was set to the average of overall ATAC signals. A total of 776 ACDs were identified by using these criteria. The reason for using large ACD regions is to make sure that we generate dense matrices without too many zeros for our downstream gene co-expression and co-accessibility analysis. This is essential for robust statistical testing, as normal statistics breaks down when dealing with sparse matrix with a lot of zeros. We noted this reason in the revised manuscript as well (Page 61, Line 1037-1040).

Following this reviewer’s suggestion, we aligned identified ACDs with LADs, compartments and ChromHMM-identified genomic regions in mESCs published previously ³⁻⁵. As expected, we found that ACDs overlap strongly with active compartments, enhancers and promoters, but are more likely to be excluded from inactive compartments, intergenic regions, repressed chromatin, heterochromatin and LADs. We plotted the results from alignment analysis in Figure R2, included the results in the updated manuscript (Extended Data Fig. 1d) and updated the analysis procedures in the

Methods section (Methods: Identification and characterization of ACDs).

Figure R2. ACDs overlap with active compartments, enhancers and promoters, but are more likely to exclude with inactive compartments, intergenic regions, repressed chromatin, heterochromatin and LADs. To align ACDs with genomic features including A/B compartments, LADs and ChromHMM regions, the mouse genome (mm10) was divided into 500-base pair windows using the "tileGenome" function within the GenomicRanges package. These windows were then overlapped with multiple genomic features, including mESC bulk ATAC-seq peaks², LAD regions⁴, A/B compartments³ and ChromHMM states⁵, and annotated accordingly. To assess the enrichment of ATAC-seq

peaks within each of these annotated groups, we calculated the fold change as the ratio of ATAC-seq peaks in each group compared to the ratio across the entire genome. Groups exhibiting a fold change greater than 1 were considered to be enriched with ATAC-seq peaks.

I have a conceptual concern that the authors need to properly address. The authors claim that upon loss of RAD21 a correlation between genes in ACDs is detected in the single cell RNA-seq data. RNA-seq measures the stable transcript pool. Changes in this transcript pool is limited as the authors indicate themselves. Previous studies have shown that changes in transcription (i.e. newly produced transcripts) can be quite large following acute protein depletion, but it can take some time for these differences to transpire to the stable transcript pool. The authors should make it (more) plausible that RAD21 depletion can lead to changes in the stable transcript pool that can actually lead to correlated expression of gene pairs.

We agree with the reviewer that the effect from transient transcriptional output could be overwhelmed from the analysis of accumulated readout after RAD21 depletion. From our data, we have shown that acute depletion of RAD21 induces detectable changes in genome structure as early as 3 hours after auxin treatment, suggesting that there is a sufficient time window for the difference to accumulate and emerge. Moreover, a previous study based on the analysis of single-cell RNA-seq data showed that even when analyzing mRNA levels in wild type cells, stochastic gene co-expression is significantly higher in *cis* than in *trans*⁶. Again, it is important to note that the elevated gene co-expression correlation in *cis* based on SMART-SCRB was further corroborated by other independent imaging and genomic results such as increased gene co-bursting in *cis* detected by intron-FISH (Fig. 3f), spatial mixing of accessible chromatin in *cis*² and increased chromatin co-accessibility in *cis* (Fig. 2, d and e).

The authors claim that ACDs and spots in the ATAC-PALM overlap with each other (e.g. see title of Ext Figure 8). However, it is unclear how the author come to this conclusion. If they want to make this claim they should perform ATAC-PALM combined with FISH (or some other method) to show that these regions are actually overlapping.

We note here that, in our previously published 3D-ATAC-PALM method paper¹, we have

demonstrated by coupling 3D-ATAC-PALM with oligopaint DNA–fluorescence in situ hybridization (FISH), RNA–FISH and protein fluorescence that accessible chromatin clusters detected by 3D-ATAC-PALM enclose active chromatin and actively transcribed genes. In addition, ACDs are spatially excluded from heterochromatic regions labeled by HP1-GFP. Furthermore, in our current study, we combined 3D-ATAC-PALM with super-resolution protein PALM imaging and demonstrated that, within nucleus, ATAC-PALM spots globally colocalize with clusters of active chromatin marks (e.g. MED1) (Extended Data Fig. 8a-c). We have revised our manuscript (Page 4, Line 69-72; Page 7, Line 193-195) to emphasize results from these validation experiments.

Fig. 3f: the authors need to better explain what the normalized contact frequency means. It is unclear why there is still a decay in the contact frequency if it normalized. Normally, normalization should get rid of this decay.

We thank the reviewer for this suggestion. To calculate the normalized contact frequency for a specific ACD pair, we used published mESC HiC data ² and calculated the average value over all the contact frequencies within the boundaries of the ACD pair. The average value was defined as the normalized contact frequency for that specific ACD pair. As expected, there should be a decay for this normalized contact frequency per ACD pair along with increased genomic distance. We have updated the Methods section to elaborate the calculation method more extensively (Page 61, Line 1033-1049).

Reviewer #2:

Remarks to the Author:

The manuscript by Dong et al. titled “Cohesin prevents cross-domain gene coactivation” aims to clarify the debate over the role chromatin structure on gene regulation. Recent work from different labs revealed limited effect on transcription upon acute short-term depletion of chromatin structural proteins casting questions about what the role of cohesin and genome structure is for gene expression (Hsieh et al. 2022). In this work, as the population averages obtained by bulk sequencing methods for gene expression and chromatin accessibility can over mask differences in a single cell level; the authors

instead make use of single-cell assays of transcriptome and chromatin accessibility to suggest a role for cohesin to hinder the co-expression tendencies of cis-linked genes.

Using a number of cutting-edge single-cell RNA and ATAC-seq methods and microscopy techniques including 3D-PALM and seqFISH, the authors presented evidence to support a role for cohesin in limiting stochastic co-expression between genes located in different chromatin domains on a chromosome. The authors dive deeper into the previous observation that accessible chromatin domains (ACDs) mix and merge upon cohesin loss and correlates it to transcriptional outcomes in single cells. They demonstrate a great role for cohesin to limit co-expression tendencies in single cells and demonstrates that single cell methods can provide valuable insights into the chromatin structure - function relationship that has been limited by use of bulk methods. On the other hand, in its current form the study still falls short on experimental controls, data analysis and interpretations, and method descriptions, as detailed below. Further explanations are required about the questions raised on cell line generation and FISH gene selections. If the authors are able to adequately address these technical issues, this study would make a valuable contribution to the chromatin & 3D genome field.

Comments:

1. Experimental design. The control condition for the Auxin degron is sub-optimal. Current description suggests that untreated mESCs were used as a control throughout the study. This control, however, is insufficient for the authors to make the conclusions stated due to the possibility that the perturbations of mESCs by the buffer used to dissolve auxin or the degron-independent effects of auxin, that caused the observed changes. It is therefore critical to include additional controls in the study, including using wild type mESCs treated with auxin, and mESCs treated with buffer only. Such controls should be

included in the single-cell RNA-seq, single-cell ATAC-seq and single molecular imaging data acquisition, to provide confidence about the increased cross ACD gene co-expression and co-accessibility in cohesin-depleted mESCs.

We thank the reviewer for his/her comments. We would like to clarify that, throughout our study, we used indole-3-acetic acid sodium salt dissolved in water to treat cells and to

deplete RAD21. Moreover, we have demonstrated that auxin treatment of OsTir1 stable mESCs has little effect on either RAD21 level or cell viability in our previous publication². Therefore, the results from both the control group and the experimental group are comparable and the differences could only reflect the effect from cohesin loss. We further clarified this point in the Methods part (Page 56, Line 903-905).

2. The scRNA-seq dataset should be analyzed further to provide biological explanations to some of the phenomenon mentioned. One example is that they observe that cohesin loss alters differential co-expression coefficient per ACD pair both positively and negatively with the averages shifts toward the positive direction. It is important to elaborate on the ACD pairs that show negative diff co-expression coefficient upon cohesin loss and whether the location of the ACD pairs, the genomic distance or the genes that reside in them are relevant with this observation.

We thank the reviewer for his/her suggestions. We have performed more analyses to compare the distance and localization and chromatin environment of positively and negatively affected ACD pairs to test differential effects. As shown in Figure R3A, we did not observe significant differences in the pair-wise distance between positively and negatively affected ACD pairs upon cohesin loss (Fig. R3a).

To further dissect the relationship between positively and negatively affected ACD pairs and other genomic features, we compare ACDs with ChromHMM-classified regions (e.g. promoter, enhancer, repressed chromatin, heterochromatin and etc.) as shown in Fig. R2 and plotted the results in Fig. R3b. After evaluating ChromHMM enrichment score for each of 11 chromHMM features covered by ACD pairs with positive and negative differential co-expression coefficients, we found that positive and negative ACD pairs show similar proportion of enrichment for 11 chromHMM features. It is notable that positive ACD pairs have higher enrichment score than negative ones for each ChromHMM feature, which is consistent with the fact that the number of positively affected ACD pairs is larger than that of negatively affected ACD pairs.

Figure R3. Analysis of the features of ACD pairs with positive and negative differential co-expression coefficients. (a) Violin plot of genomic distance between ACD pairs with positive and negative differential co-expression coefficients. Solid line indicates median value while dotted lines are 25% and 75% quartiles. (b) Bar plot of ChromHMM enrichment for ACD pairs with positive and negative differential co-expression coefficients, respectively. The count for each chromHMM region (see Fig. R2) covered by ACD pairs with positive and negative differential co-expression coefficients was evaluated independently. n.s., not significant.

3. Are there regions transcribed that do not reside in the identified ACDs? If so, Is there an enrichment of co-expression only at the ACDs? A comparison using random or, non- ACD regions on the same chromosome would be beneficial to see (a)whether the co-expression effect is specific to ACDs, or the accuracy of ACD calling in the event of no-transcription elsewhere.

We identified ACDs for later use with the density of ATAC peaks above a certain threshold and therefore filtered out many sparsely distributed transcribed regions.

According to the reviewer's suggestion, we performed additional analysis of SMART-SCRB and scATAC-seq data by using non-ACD regions generated by random window shifting and compared the results with those by using identified ACDs. We found that the co-expression effect is much stronger in ACDs than that in non-ACDs. However, it is important to note that there are also increased co-expression coefficients in non-ACD regions after cohesin depletion (Fig.R4).

Figure R4. Comparison of co-expression effect in ACD regions (a) versus non-ACD regions (b). The statistics of $\Delta C_{RNA}(i,j)$ per ACD pair before and after Cohesin depletion in chromosome 1-19 and X. Each circle indicates the value of differential co-expression coefficient for one ACD pair. The red line indicates the median value; the dotted line indicates the zero-change line.

4. Please comment on whether the gene-co expression or expression changes in a single ACD upon cohesin loss. If so, does it have a correlation with the size of the ACD? What is the relationship between TADs and ACDs?

Gene-to-gene correlation within single ACD is challenging to perform because normal statistic method broken down for sparse matrices with lots of zeros (when dealing with expression levels of single genes in the single-cell data). The statistical testing at the ACD

level (with multiple genes) is more robust. So, we could not comment on gene co-expression changes within ACDs.

To assess the relationship between gene co-expression and the size of ACDs, we made a scatter plot of differential co-expression coefficient and ACD pair size (ACD1 length * ACD2 length) for all data points and performed the linear regression analysis. There is no relationship between these two parameters. The results are shown in Fig. R5.

We think that ACDs are primarily active compartments or active TADs.

Figure R5. Scatter plot analysis of differential co-expression coefficient and ACD pair size over all data points. The black line is the linear regression line.

5. The definition of ACD is vague. Does it consider the uneven distribution of genes in the genome? If not, it is possible that the ACDs might not fully correspond to the ACD in the nuclear space.

Regarding the heterogenous and stochastic nature of genome organization, we provided a detailed algorithm to compute and identify ACDs with pre-set threshold values. The algorithm identifies ACDs based on accessible chromatin density rather than the uneven distribution of genes in the genome.

We believe that our identified ACDs correspond to accessible chromatin in the physical space in the nucleus for two reasons: 1) in our previous study, we have demonstrated by coupling 3D-ATAC-PALM with oligopaint DNA–fluorescence in situ hybridization that ACDs defined in the linear genome colocalize with ATAC-PALM clusters in the nuclear space ¹; 2) We have performed 2 color PALM experiments and demonstrated that, within nucleus, ATAC-PALM spots globally colocalized with clusters of active chromatin marks (e.g. MED1) (Extended Data Fig. 8a-c), which have been shown to colocalize with active chromatin across 1D genome ⁷. We have revised our manuscript (Page 4, Line 69-72; Page 7, Line 193-195) to emphasize results from these validation experiments.

6. Does the gene expression variance in between cells change upon cohesin loss in between single cells? Do you observe a variance change in a specific set of genes? Do you observe a co-variance change at ACDs after Rad21 depletion?

We thank the reviewer for his/her comments. We analyzed the standard deviation (square root of variance) of gene expression for each gene detected in single cells in different conditions. For the majority of the genes, the expression variance showed no significantly change after cohesin depletion. However globally, more genes show decreased standard deviation than those showing increased standard deviation (Fig. R6a). We observed 6 genes with significant increase (>2 fold) and 27 genes with significant decrease (>2 fold) in standard deviation of gene expression (Fig. R6a).

To determine the co-variance change of gene expression per ACD pair after cohesin depletion, we quantified differential co-variance score per ACD pair and plotted the results chromosome-wise. Our analysis showed that ACD pairs in 12 of 20 chromosomes show significant co-variance increase after cohesin depletion (Fig. R6b). Regarding the fact that gene expression values measured by SMART-SCRB do not necessarily follow the normal distribution, this increase in co-variance is not equal to but consistent with the detected increase in co-expression coefficient across individual chromosomes shown in the manuscript.

Figure R6. Analysis of gene expression standard deviation (a) and chromosome-wise co-variance score per ACD pair (b) before and after cohesin depletion. (a) 7,700 genes with average RNA-seq count above 2 were analyzed and each circle represents one gene. The solid line is the bisector line and dotted lines illustrated the cut-off values for genes with significant change (>2 fold) in expression standard deviation. Gene names with significant changes are shown in boxes. (b) The differential co-variance score (use procedures similar as computing differential co-expression coefficient, see Extended Data Fig. 1G) per ACD pair was computed and plotted chromosome-wise. Red lines are

the median values.

7. Previous work from the authors alluded to BRD2 mixing the accessible chromatin upon cohesin loss, and that acute loss of BRD2 reverses the spatial mixing of ACDs upon

cohesin loss (Xie, Dong et al. 2022). Could the authors comment whether this system can reverse the co-activation of cis-linked genes that they observe with SMART-SCRB (Rad21, Brd2 co-depletion)?

We thank the reviewer for his/her comments. Currently we have one ongoing project that is investigating the role of BRD2/BRD4 proteins in gene co-expression and co-accessibility. We speculate that BRD2 depletion could reverse increased gene co-activation after cohesin depletion. We will update our findings in future publications.

8. Regarding FISH on Figure 3. Is there a correlation between differential distance and differential bursting frequency?

We thank the reviewer for his/her suggestion and performed the scatter plot of differential distance versus normalized differential bursting frequency in Fig. R7. Our results show little correlation between differential distance and normalized differential bursting frequency.

Figure R7. Scatter plot analysis of differential distance versus normalized differential bursting frequency. The black line is the linear regression line.

9. Figure 4e: Could you explain the reasoning behind the selection of gene couples? Why did the selection only include lineage specific genes? Is there any chance of seeing other sets of genes? Please provide gene expression levels of these lineage specific genes in both control and Rad21 depletion conditions with the scRNA-seq data.

We selected these gene pairs because each pair includes genes marking separate cell lineages, highlighting potential developmental consequences of unregulated gene co-expression. However, based on our genomic results and chromosomal wide intron-seqFISH results, we believe that cohesin supplies a general function to neutralize gene co-expression in *cis*.

We have taken the reviewer's suggestion and modified Extended Data Fig. 6a to incorporate the gene expression levels of these lineage specific genes in both control and RAD21 depletion conditions.

10. Extended Figure 7: Concerns about Halo tagged Med1 and Med6 generation: Is the WT mESC on first lane of western blots show no tagging of Rad21 with the degron, or is it the original Rad21 mAID line that is used to tag Med1? On top of Extended Figure 7e&f: it is written as Rad21-mAID-eGFP. Please explain /correct this. If it is WT cell in the first lane, please provide a lane with the Rad21 degron parental cell line (no depletion).

The first lane of western blots in Extended Data Fig. 7 represents the sample from Wild-type JM8.N4 ES cell line. To clarify this question, we performed western blots on WT mESCs, WT mESCs with stably expressed Tir1 (used to build up the RAD21 degron parental cell line), RAD21-mAID-eGFP^{+/+} degron parental cell line (no auxin v.s. 6h auxin), RAD21-mAID-eGFP^{+/+}/Halo-MED1^{+/+} cells (no auxin v.s. 6h auxin) and RAD21-mAID-eGFP^{+/+}/MED6-Halo^{+/+} cells (no auxin v.s. 6h auxin). The results are shown in Fig. R8 below and also included in the updated manuscript (Extended Data Fig. 7, g and h).

We have corrected “Rad21-mAID-GFP” as “Rad21-mAID-eGFP” in the figures.

Figure R8. Western blots show the efficacy of RAD21 degron system in the parental cell line and established MED1(a) and MED6(b) knockin cell lines before and after auxin treatment for 6 hours. The normalized RAD21 or RAD21-mAID-eGFP protein level to the loading α -tubulin level for each condition was quantified and shown below each lane.

Currently we do see a reduced Rad21 amount before depletion (lane 2 in e and f) and

gives the impression that both Med6-Halotag and Med1 halo tag cell lines demonstrate significantly lower levels of Rad21 even before the auxin loss. Not only that, but you also have lower levels of Med6 upon tagging with the Halo tag, is this an antibody issue? Does this reduced level of Med6 effect the cells in any way?

We think that this observed discrepancy in band intensity between RAD21 and RAD21-mAID-eGFP is primarily due to the different transferring efficacies between proteins of different molecular weight. The larger molecular weight the protein has, the slower it can be transferred from the gel to the membrane.

Does it affect the size of ACDs? You could have confounding effects if both Rad21 and Med6 levels are reduced in this double tagged cell line specifically. Please also provide the cell cycle profile of the Rad21-degron parental line for extended figure 7i.

We think that this observed discrepancy in band intensity between RAD21 and RAD21-mAID-eGFP is primarily due to the different transferring efficacies between proteins of different molecular weight. The larger molecular weight the protein has, the slower it can be transferred from the gel to the membrane.

According to the reviewer's suggestion, we provided the cell cycle profile of the RAD21-degron parental line before and after cohesin depletion in Extended Data figure 7k.

11. General notes: Please show Rad21 depletion by western blot for the time-points you have chosen for your experiments as they range from 3 hours (seqFISH) to 6 hours.

According to the reviewer's suggestion, we have performed the time-course WB experiments to examine RAD21 depletion at 1h, 3h, 6h and 9h, as shown in Fig. R9 below.

Figure R9. Western blots show the time-course depletion of RAD21 by the degron system after auxin treatment. The normalized RAD21 or RAD21-mAID-eGFP protein level to the loading α -tubulin level for each condition was quantified and shown below each lane.

12. On Figure 5i, the authors show that 2 gene pairs that reside within the same ACD do not show any change in co-bursting fraction whereas gene pairs residing in different ACDs experience the increase upon Rad21 loss. I have concerns about the potential gene expression level differences effecting these results. Both Pou5f1(Oct4) and Abcf1 (Choi et al., Science Advances 2021) are crucial for ESC self-renewal and Cd9 is expressed at high levels in ESCs (Akutsu et al. 2009). Are the lineage specific genes expressed at similar levels to the other 4 (within ACD group) with similar variance? Could you please show whether one of the 4 genes (Cd9/ Mlf2, pou5f1/Abcf1) from the “within ACD” group shows increased co-bursting with another gene from another ACD on the same chromosome (and whether one of the lineage specific genes from different ACDs demonstrated to have a co-bursting fraction increase shows no change with a gene that is in the same ACD.

We thank the reviewer for his/her comments. We clarify that we investigated the effect of cohesin depletion on the fraction of co-bursting loci connected by MED6 hubs on Figure 5i. We found that gene pairs that reside within the same ACD and within the same MED6 hub do not show significant change in the fraction, whereas gene pairs residing in different ACDs experience the increased probability of co-localizing into the shared MED6 hub.

To address the reviewer’s concern, we have performed Intron-FISH experiments of six selected genes (Cd9, Nanog, Mlf2, Ehmt2, Pou5f1 and Abcf1) to evaluate the co-bursting frequencies and distances for gene pairs that both reside between different

ACDs (Cd9/Nanog and Ehmt2/Pou5f1) and locate within the same ACD (Cd9/Mlf2 and Abcf1/Pou5f1). The results are shown in Fig. R10.

We plotted the bursting fraction (corresponding to expression level) of selected lineage-specific genes and additional six selected genes. Results show that lineage specific genes are expressed at different levels (Fig. R10a). Among them, Gpc1, Tead1 and Trim33 are expressed at similar levels to those six selected genes while the others have lower expression levels. All the genes examined have different but comparable expression variance.

We did observe that, for six selected genes, pairs from different ACDs on the same chromosome show increased co-bursting and decreased spatial distance after cohesin depletion, which is consistent with our observations on lineage specific genes. In contrast, pairs from the same ACD show decreased co-bursting and increased spatial distance after cohesin depletion (Fig. R10, b and c).

Figure R10. Bursting frequencies (a), co-bursting frequencies (b) and distances (c) of lineage-specific genes and 6 additionally selected genes analyzed before and after cohesin depletion.

Minor comments:

1. Extended figure 1a. what are the numbers 173,173,171,169 correspond to?

These numbers represent the numbers of single cells analyzed for each condition using SMART-SCRIB. We have revised figure legends to clarify this in the updated manuscript.

2. The method for identifying ACD is insufficiently described, which makes it difficult to reproduce independently. Please provide more information, as other sections in the manuscript.

We have described the procedures for identifying ACDs in Methods part in the manuscript and have uploaded the list of ACDs with their genomic information as the supplementary table 2.

3. The description of the Rad21 degron mESC line is missing; how was this cell line constructed? If it was described previously, please provide a reference.

We thank this reviewer for the comment. The RAD21 degron mESC line used in this study was generated in our published research ². We have clarified this in the method section (Page 57, line 942-943) of the updated manuscript.

4. Line 244. “insulted” is a typo. Should be “insulated”.

We have corrected the typo in the updated manuscript.

Reviewer #3:

Remarks to the Author:

Dong and colleagues present a tour de force manuscript with an impressive array of advanced imaging technologies applied to better understand how cohesin might contribute to control of gene expression -- beyond its limited acute effects on population average gene expression. They interpret the data to suggest that cohesin suppresses co-bursting of linearly distal genes by maintaining a more spatially segregated genome and counterbalancing aggregation of genes in cis and of TF clusters. Generally the manuscript is well written and the logic of each newly designed experiment and its interpretation are well explained. Several sections are a bit sparse on details in the main text, which seems to be a necessary but unfortunate consequence of packing so many complex and very

diverse and bespoke experiment type into the same work. The corresponding figures (even with the ED figures) are also sparse on some instructive depictions of the data and some informative controls, as I describe below. I suspect several of these omissions were made in the interest of brevity and readability, but I believe there are a few analyses worth adding to make the interpretation more convincing.

Primary Concern:

Mouse ESCs have rapid cell cycles and do not readily arrest in mitosis without cohesin, but proceed through mitosis with segregation defects. Segregation defects would skew the abundance of individual chromosomes, changing the number of templates and affecting RNA-seq levels, also changing the number of templates to provide ATAC signal from. As the cell loses or gains a whole chromosome, we would expect whole chromosome level correlated changes (i.e. cis-specific changes) in ATAC signal and in RNA expression. 6 hours of auxin depletion is quite long compared to the cell cycle time in mESCs (~8 hours, line-dependent), so a decent fraction of cells may well have been through an aberrant mitosis. At the population level, the relative gain or loss of a chromosome cell-to-cell would average out, but create variability at the single cell level. I am concerned that a number of the results in Figs 1 – 3 may be explained by this effect. In addition extranumerary foci in one of the intron FISH appear to suggest it. I discuss in my detailed comments below where I think this issue arises and what implications it has for the interpretation of the results, as well as some suggestions on how it may be best addressed.

Questions and suggestions pertaining to the main results (organized by figure):
Figure 1: It would be instructive for the authors to show more of the data, not just the few sparse quite downstream processing steps (as I describe in more detail in a moment). I appreciate this is a causality of staggering number of experiments that compress what could be a whole manuscript into a single figure. However, several key controls/comparisons would strengthen the claims of this section:

- 1) Fig 1e. What happens if you compute the C_RNA_Ctrl and C_RNA_Rad21 from S_RNA_Ctrl and S_RNA_RAD21- rather than from their difference? Is there

generally a positive correlation in the gene expression from ACD to ACD in untreated cells? And then can you show this correlation gets weaker? The individual gene-to-gene correlations across ACDs seems to me to be a much noisier (sample size dependent) value to subtract. 400 cells is not a lot for looking at variation in gene expression genome wide.

We have computed C_{RNA_Ctrl} and C_{RNA_RAD21} directly from S_{RNA_Ctrl} and S_{RNA_RAD21} by averaging all the Spearman coefficients within the region of one ACD pair. Our results show a general positive correlation in the gene co-expression between ACD pairs in untreated cells (Fig. R11a). We also calculated differential average co-expression coefficients (ΔC_{RNA}) per ACD pair across all the chromosomes and the results indicate a unified increase of gene co-expression after cohesin depletion (Fig. R11b), which is consistent with our gene co-expression analysis shown in the manuscript (Fig. 1e).

We agree that single gene-gene correlation analysis is not robust because normal statistic method broken down for sparse matrices with lots of zeros. This is the primary reason that we analyzed gene co-expression correlation based on ACDs. In addition to the number of cells analyzed, we think that the sequencing depth and reliability are more important to make a robust conclusion. Therefore, we chose to use Smart- Single Cell RNA Barcoding (Smart-SCRb) method instead of droplet-based approaches.

a

b

Figure R11. Analyzing gene co-expression effect by computing C_{RNA} directly from S_{RNA} . (a) Heatmaps show that C_{RNA} values per active ACD pair in Chr2 in untreated cells are largely positive. (b) The statistics of differential average co-expression coefficient (ΔC_{RNA}) per ACD pair after cohesin depletion in chromosome 1-19 and X. Each circle indicates the value of differential co-expression coefficient for one ACD pair. The red line indicates the median value; the dotted line indicates the zero-change line.

2) Fig 1f. What do the trans correlations look like? I feel it would be instructive to

create a map like Fig 2d, before jumping to the difference map, and it would be helpful to see the genome map (cis and trans) like Fig 2e, rather than just the single chromosome example. It would also improve the symmetry of the presentation.

We thank the reviewer for his/her suggestions. To address the question of *cis/trans* bias, we have performed the analysis on the correlation of RNA expression in both *cis* and *trans* as shown in Figure 2d (Fig. R1, a and b). The results indicate that there is indeed a *cis-over-trans* bias. Moreover, we also performed intron-FISH to examine co-bursting frequencies of genes localized in *trans*. Specifically, we focused on the same pool of genes showed elevated co-activation in *cis* upon cohesin loss and examined co-bursting frequencies and bursting-site distances when these genes are paired in *trans*. Our results suggest that the observed increase in the correlation of RNA expression per ACD pair in *cis* after cohesin depletion is a predominant and specific effect (Fig. R1, c and d).

3) Is there a cell cycle contribution driving the change in correlation? RAD21 depletion can have an effect on cell cycle. If one cluster is replicated, the cell must be in S/G2 and thus it is more likely other parts of the chromosome are replicated as well. If genes in multiple ACDs are replicated, there are 2 templates for each gene and on average one expects higher expression compared to if there is 1. Thus differences in cell cycle progression (primarily S phase for mESCs) in each sample may manifest as genome wide changes in correlated expression. This logic doesn't easily explain a *cis/trans* bias, but it is also not clear from the presentation of the RNA expression data that there is a *cis* bias, as only correlation within chromosomes is shown, unlike in Figure 2.

We understand the reviewer's concern. We have categorized cells into different cell cycle stages according to their cell-cycle-related gene expression and performed the correlation analysis for individual phases. Our data showed that the increase in gene co-expression takes place in all different cell cycle stages upon cohesin depletion (Extended Data Fig. 3, c and d). However, it is worth noting that cohesin loss does lead to larger gene co-expression increases for S/G2 cells than for G1 cells (Extended Data Fig. 3d), suggesting preferential cell cycle effects to some extent.

4) Are there chromosome abnormalities after this long depletion time in a rapidly

cycling cell background? (We get defects if we do 6 hr auxin in mESCs with a cohesin degnon in our hands). Might gaining an extra copy of the maternal or paternal for just 1 of the 23 chromosomes give the appearance of enhanced co-bursting purely due to the increased probability of observing expression for this 1 chromosome but not the others? The coburst would also appear to be concentrated in cis, as suggested by the data.

We thank this reviewer for bringing up this possibility. To assess chromosome segregation defects after cohesin loss, we took this reviewer suggestion and plotted ATAC reads per chromosome in control and RAD21(-) conditions (Fig. R12). Indeed, we observed slightly larger read spreads cross all chromosomes upon cohesin loss, suggesting the probability of chromosome mis-segregation is higher as this reviewer suspected.

Figure R12. Box plots of scATAC-seq reads from 6490 cells (batch normalized) parsed into each chromosome under ctrl and Rad21 loss conditions. The lower and upper whiskers represent 10% and 90% values; the box represents the range from 25% to 75% percentile; the center line and the dot represents the median and the mean respectively.

To investigate whether losing or gaining a chromosome alone would affect co-bursting

(and co-accessibility) in *cis*, we performed numerical simulations (Fig. R13). Specifically, in our seqFISH / smRNA-FISH experiment, we measured co-bursting events that occur within 1 μm distance. This cut-off would allow us to selectively measure co-bursting events *on the same chromosome*. In Smart-SCBR and scATAC-seq, we do not have the resolving power to separate co-activation / co-accessibility *from the same chromosome* and that *from sister chromosomes*. In our simulation experiments, we took both scenarios into consideration. We found that, under both conditions, only the variance of gene co-expression / co-accessibility but NOT the frequency of gene co-expression / co-accessibility increases as a function of chromosome mis-segregation. This observation makes sense as the uneven distribution of chromosomes in single cells increases the heterogeneity but not the average probability of gene co-expression / co-accessibility. These results suggest that slight increases in chromosome mis-segregation upon Cohesin loss could not account for higher cross-domain gene co-activation / co-accessibility observed in our single-cell imaging and genomic experiments.

Figure R13. a. schematics showing chromosome mis-segregation (left) and parameters

used in the simulation experiment. The simulation is repeated 1000 times on 1000 cells. The number of co-bursting / co-opening sites and its variance is calculated under cis-linked (b, d) and unlinked (c,d) conditions. For representation purpose, we only showed results where $p_A = p_B = 0.5$. The observed trends are consistent across a wide range of p_A and p_B values tested.

5) ED Fig 2 – I appreciated the explanation provided graphically in this figure. It might help if the colormaps had colorbars (though I appreciate the data appear to be purely cartoon representations to understand the workflow rather than real data).

We thank this reviewer's understanding that, in this figure, the colormaps are pure cartoon representations so that we did not include color bars. We have revised figure legends to clarify this further.

Figure 2, ATAC analysis

1) it would be helpful to state in the legend (and/or the text) how many cells were analyzed.

We thank the reviewer for his/her suggestions and have included the number of cells analyzed in the legends of Fig. 2d in the updated manuscript.

2) Might this data also be explained by the chromosomal abnormalities? An extra copy of just one chromosome would make it look like all the ATAC-accessible regions on that 1 chromosome were 2x more accessible (just due to increased template abundance). While the other chromosomes in that cell would all have WT levels of accessibility. This would look like a significant co-burst. Given the noise of the scATAC data at weak peaks, the 2x effect won't be sensitively detected at all sites, but the authors could test for it still by aggregating and plotting all the reads per chromosome per cell. It would be straightforward analysis (no new experiments required at least) to plot a matrix of cells x chromosomes with the total reads per chromosome represented in the matrix values (e.g. as color). This pattern could either

help support or refute suspicion that chromosome abnormalities drive this apparent increase in co-accessibility reported.

We thank the reviewer for his/her insightful suggestions and have plotted ATAC reads per chromosome in control and RAD21(-) conditions (Fig. R12). Indeed, we observed slightly larger read spreads cross all chromosomes upon cohesin loss, suggesting the probability of chromosome mis-segregation is higher as this reviewer suspected. We performed numerical simulation showing that only the variance of gene co-expression / co-accessibility but NOT the frequency of gene co-expression / co-accessibility increases as a function of chromosome mis-segregation. This observation makes sense as the uneven distribution of chromosomes in single cells increases the heterogeneity but not the average probability of gene co-expression / co-accessibility. So, increases in gene co-expression and co-accessibility could not be explained by chromosomal abnormalities.

Figure 3 – co-bursting by sequential intron FISH:

1) extra chromosome copies would increase the probability of detecting a co-burst. As most genes lack intron signals in most fraction of the cells, having more copies improves the probability one of them is co-bursting. I'm concerned the co-bursting is associated with these mitotic defects mentioned in my primary concern. Do the authors see these with shorter auxin treatment? Or in a cell type that doesn't have such rapid cell cycles?

(Ideally the authors could differentiate to post-mitotic cells and repeat the tests there. Though care should be taken as mESCs also lose chromosomes spontaneously on differentiation, especially if the passage number is high, as is often necessarily the case when a large number of edits have been made in the line).

We thank this reviewer's comments and the insightful suggestion. As shown in Fig.R13, chromosomal mis-segregation could not affect the average frequency of gene co-expression and co-opening. Currently, we have a following-up project evaluating the effect of cohesin loss in post-mitotic neurons using Nipbl Flox mouse model (in collaboration with Drs. Anne Calof and Arthur Lander). Preliminary imaging results

verified increased clustering of Chr 2 intron puncta upon the knock-out of Nibpl in post-mitotic neurons, consistent with what we showed in Fig. 4a-d. We plan to perform extensive genomic and imaging analysis and report these findings in a separate paper.

2) What is the distance between co-bursting genes? The figure plots change in distance, but not the actual distance distributions. The extended data have a very helpful illustration of the workflow and a nice presentation of the chromatic correction data, but I didn't find actual distance distributions there either.

We thank the reviewer for his/her appreciation and suggestion. We have included the actual distance plots of co-bursting genes before and after cohesin depletion (Fig. R14, also included as extended data Fig. 4d-e).

Figure R14. Heatmap (a) and violin plot (b) of actual distances between co-bursting genes before and after cohesin depletion. Red solid line indicates mean value while dotted lines are 25% and 75% quartiles.

3) What are the actual burst-frequencies per gene? While not the main point of the paper, it would be a helpful validation to include the data in the supplement.

We have included the list of actual bursting frequencies per gene before and after cohesin

depletion in the Supplementary Table 2 of the updated manuscript.

4) It would be a helpful control to validate the iFISH data and the successful demultiplexing by comparing the aggregate levels to another data set. For example the authors might compare the burst frequencies to the bulk RNA-seq. There are also publically available nascent RNA-seq datasets and previously published genome wide intron RNA FISH data sets the authors could validate against. These data are compared between the Cntrl and degran but it would be helpful to validate the frequency range in the ctrl.

We thank the reviewer for his/her suggestions. We have compared the bursting frequencies of selected 208 genes in Chr2 for seqFISH experiments in control cells with the nascent RNA-seq count downloaded from published mNET-seq dataset ⁸. We performed scatter plot and linear regression and found a significant positive correlation between these two parameters (Fig. R15, also included as extended data Fig. 4f). This cross-modality comparison validates our seqFISH experimental platform and suggests that the frequency range we have adopted for our analysis is appropriate.

Figure R15. Validating bursting frequencies measured by seqFISH using published nascent RNA-seq data. 199 of 208 genes in Chr2 (probed by seqFISH experiment) with detected nascent RNA-seq counts were used for scatter plot. Each circle represents one gene and the red line is the linear regression line with $R^2=0.578$.

5) I did appreciate the more substantial number of cells analyzed in this figure and the clear labeling (N=1475 and 1121 for degron and control).

We appreciate this reviewer's positive feedback.

Fig 4

1) Clustering of transcribed genes looked good.

We thank this reviewer for his/her positive comment.

2) Fig 4e. There appear to be 3 Gbx2 nascent foci in the RAD21 case, and the ~1um scale distance between the upper two is considerably larger than expected for paired sisters. Thus, to me this image indicates a possibility of an extra chromosome copy in the cohesin depleted condition. It should be noted that depending on signal-to-noise of the intron FISH, this could be stray signal. It is hard to tell with the 3D rendering, and hard to appreciate the quality of the intron FISH data without the controls discussed above.

We thank the reviewer for his/her insightful comments. We agreed that there are 3 Gbx2 nascent foci from 3D rendering in the RAD21 case and they could be in the reviewer's conjecture. However, since we have measured over 300 individual cells for each condition and have used an optimized uniform parameter setting for 3D rendering and analysis, we are confident about the results and the conclusion we have made. In the updated manuscript, we modified Fig. 4e with an unambiguous example.

Figure 5 - Mediator clustering analysis

1) The changes in Mediator organization in the example images are striking, and the quantification and statistical analysis appear to be performed to a high standard.

We thank this reviewer for his/her positive feedback.

2) The authors images of Med hubs look qualitatively quite radically different from previous publications describing Mediator hubs (Sabari...Young 2018), which show much sparser, much more punctile staining, in both live and fixed cells (also mESCs), for both Mediator and BRD4. Given the Liu lab's expertise with this sort of imaging, this reviewer is more inclined to believe the results presented here, but I'm curious if the authors could comment on the striking discrepancy.

For the imaging analysis, we used Imaris software to process the grayscale images and converted intensity signals to iso-surface by setting a unified threshold value. The discrepancy between our visualization and previously published results from Sabari *et al.* may be due to the different threshold values chosen. The higher the threshold is set, the sparser Mediator hubs are. However, as long as the threshold values are consistent between the control condition and cohesin-depleted condition, the conclusion should be solid.

3) "We found that Mediator hub fusion spatially correlates with clustering of actively transcribed genes in Chr 2 upon cohesin loss"

It wasn't clear to me from the analysis presented that this was hub *fusion* more than an increase in Mediator. Do mediator levels increase? (e.g. by Western?) Is the total aggregate volume occupied by Mediator increase? (it appears so in the images, it doesn't look like fewer, bigger clusters so much as just more mediator. The authors might clarify this by examining the number of clusters as well and see if this goes down, as expected for fusing a large number of small clusters into a smaller number of large clusters.

We thank the reviewer for his/her comments. We would like to clarify two points here.

First, despite forming high-concentration protein hubs, Mediator proteins are also diffusive throughout the nucleus at lower concentrations. The diffusive Mediator signals are not shown in our 3D image because of the rendering threshold we have set up. We suspected that the total volume of all Mediator condensates is much smaller than the

entire nucleus volume, and thus the total number of Mediator molecules within condensates could be even less than that in diffusive space. Moreover, Mediator proteins are dynamically exchanged between these two fractions. Therefore, the total volume of Mediator condensates does not necessarily reflect the total Mediator level in the nucleus. Second, we used Airyscan imaging and thresholding to detect Mediator condensates in live cells. Due to thresholding, our assay will inevitably neglect small Mediator hubs with weaker fluorescence signals and thus is incapable of measuring and analyzing all Mediator condensates. To ensure the robustness of our assay, we only focused on measuring the top 100 Mediator hubs ranked in size (Fig.5 c-e).

In conclusion, we think that our data based on currently available technological approaches cannot adequately address the reviewer's specific questions about total Mediator condensate volume and number. However, our results clearly show that, for large mediator hubs that we measured, there is an increased possibility of spatial mixing of open chromatin sites and concomitant enlarged condensate formation.

It also wasn't clear to me if the increased overlap with chr2 intron signal was due to a greater fraction of the nuclear volume being now occupied by mediator with more larger clusters, or if hub fusion could indeed be implicated. The only quantification for this claim shows that there is a larger radii of the overlapped regions, which might result from expanded Med signal, or expanded intron chr2 signal. The assertion to me implies it is specifically the fusion correlates with the active transcription, which implies that once separate MED clusters have joined, and this has resulted in gene expression, which is not what is shown in the figure. It's a reasonable inference from the data, especially if the amount of mediator total is unchanged and the number of clusters has decreased.

We clarify here that the entire nuclear volume for each cell has Mediator proteins, whereas regions other than detected hubs have low concentrations. Therefore, if we set a specific threshold to identify iso-surface, those low-concentration regions will display as dark regions without signals.

4) It's not clear to me how or why the radius "normalization" is done, or why e is just "average radius" but the rest

To evaluate the size of Mediator hubs, we calculated the iso-surface volume of each hub and computed the radius of a sphere with equal volume. We defined this radius as “normalized radius” of the mediator hub.

5) Fig 5 I shows the fraction of co-bursting loci that are connected by Med6 hubs. If there's more Med total (more clusters, higher nuclear volume fraction) more loci will be connected. Clarifying the change (question 3) will help clarify this. These two cases have

different interpretation, and it would be good to be clear if the mechanism is fusion of existing mediator into fewer larger clusters driving more co-bursting, or just more mediator. The text seems to suggest the former rather strongly but the data to me look more like the latter (even though I don't know why the total mediator would change, and I suspect the western blots will show it doesn't). Some image quantification here could help.

We thank the reviewer for his/her comments. As we explained in the reply to point # 3, we think that our results support the conclusion that the fusion of existing mediators into fewer larger condensates drives more co-bursting.

Figure

6

No major objections. These are difficult experiments and they appear to be conducted with care. The results look interesting, though I found the conceptual connection to the other 5 figures a little less clear. It is interesting the activating TFs change in a different way the histones but I'm not sure what the implication is.

We appreciate this reviewer's positive feedback.

Minor textual comments:

1) “transcription activation does not require physical proximity between enhancer and promoter” 1

9

- I would say this claim is not widely accepted – even Alexander and colleagues propose that a degree of physical proximity is required, despite proposing that ~300 nm separation is sufficient proximity, a distance much larger than the physical diameter of many TF complexes or of a single cohesin.

- The bigger issue with the claim is that the enhancer and promoter are not TF scale (10 nm) objects (they are stretches of DNA that frequently span 300 nm) and that the authors didn’t label either the enhancer or promoter, they put 10 kb stretches of probes 5-7 kb distal, and each 5-10 kb stretch is a long flexible polymer that frequently spans 300 nm. This can be seen by the centroid-to-centroid distance in FISH probes spaced 5 kb apart (e.g. Mateo...Boettiger 2019 supp mat for sox2 in mESC, or Huang...Ren 2021 for sox2 in mESC), or in live imaging (see the “del-TAD” data in Gabriele...Hansen 2022). See PMID 33310227 and 36854248 for more details.

We agree with the reviewer and have modified the text as “*direct stable key-and-lock physical interactions might be not required for gene activation*”.

2) Smart-SCRIB refs 23,24 I am bit confused at the referencing here, maybe the authors could double check. 23 looks like “smart-seq2”, which I haven’t seen go by “Smart- SCRIB”. And Ref 24 looks like the AID degron system.

Thanks for this review catching this citation error. We have corrected it in the updated manuscript. In addition, SMART-SCRIB described in Ref. 22 is a modified scRNA-seq technique based on SMART-Seq2 (Ref. 23).

3) Final section before discussion: “Next, we analyzed local hopping of transcriptional regulators between clustered 223 binding sites in the nucleus – a process known as intersegment transfer (Fig. 6a). 224 Intersegment transfer has been proposed as a mechanism underlying long-distance enhancer-promoter communication 38,43-

47”

- I find this a bit too liberal a summary of the these works. While indeed these works speculate on a action at a moderate (~300 nm) distance, I think it goes a bit far to say they all argue for hopping of TFs. I think some of these references hypothesize rather that a large droplet of trans-activators is bound on one side to the enhancer and that the physical size of the condensate allows molecules on the other side to activate promoters 100s of nanometers away – including the possibility that a condensation reaction, templated on the enhancer, expands like a growing crystal or dew-drop until it reaches the promoter. Given that most of these references appear partial to postulating it is a liquid state aggregate, diffusion is probably happening from edge to edge, but it just seems a bit of an overstatement to summarize all these studies in one as hypothesizing “a process known as intersegment transfer.” – what fraction actually use that term or are postulating something sufficient close?

We agree with the reviewer. To avoid a model bias (LLPS versus protein hubs), we have changed all wordings associated with “*intersegment transfer*” to “*cohesin confines the exploration of a broad range of enhancer and core promoter binding transcriptional regulators.*”

References:

- 1 Xie, L. *et al.* 3D ATAC-PALM: super-resolution imaging of the accessible genome. *Nat Methods* **17**, 430-436, doi:10.1038/s41592-020-0775-2 (2020).
- 2 Xie, L. *et al.* BRD2 compartmentalizes the accessible genome. *Nat Genet* **54**, 481-491, doi:10.1038/s41588-022-01044-9 (2022).
- 3 Bonev, B. *et al.* Multiscale 3D Genome Rewiring during Mouse Neural Development. *Cell* **171**, 557- 572 e524, doi:10.1016/j.cell.2017.09.043 (2017).
- 4 Peric-Hupkes, D. *et al.* Molecular maps of the reorganization of genome-nuclear lamina interactions during differentiation. *Mol Cell* **38**, 603-613, doi:10.1016/j.molcel.2010.03.016 (2010).
- 5 Pintacuda, G. *et al.* hnRNP K Recruits PCGF3/5-PRC1 to the Xist RNA B-Repeat to Establish Polycomb-Mediated Chromosomal Silencing. *Mol Cell* **68**, 955-969 e910, doi:10.1016/j.molcel.2017.11.013 (2017).
- 6 Sun, M. & Zhang, J. Chromosome-wide co-fluctuation of stochastic gene expression in mammalian cells. *PLoS Genet* **15**, e1008389, doi:10.1371/journal.pgen.1008389 (2019).
- 7 Murphy, K. E., Meng, F. W., Makowski, C. E. & Murphy, P. J. Genome-wide chromatin accessibility is restricted by ANP32E. *Nat Commun* **11**, 5063, doi:10.1038/s41467-020-18821-x (2020).
- 8 Hsieh, T. S. *et al.* Enhancer-promoter interactions and transcription are largely maintained

upon acute loss of CTCF, cohesin, WAPL or YY1. *Nat Genet* **54**, 1919-1932, doi:10.1038/s41588-022-01223-8 (2022).

Decision Letter, first revision:

31st Oct 2023

Dear Zhe,

Your Article, "Cohesin prevents cross-domain gene co-activation", has now been seen by the 3 original referees. You will see from their comments below that while overall they find your work improved, some important points are raised. We are interested in the possibility of publishing your study in Nature Genetics, but would like to consider your response to these concerns in the form of a revised manuscript before we make a final decision on publication.

Reviewer #1 is overall positive about the revision but has a major concern regarding cis vs. trans effects. This needs to be carefully addressed.

Reviewer #2 is mostly satisfied and only has one lingering concern, which should be easy to address. Reviewer #3 finds this greatly improved and exciting. He has two final/straightforward requests.

We therefore invite you to revise your manuscript taking into account all reviewer comments. Please highlight all changes in the manuscript text file. At this stage we will need you to upload a copy of the manuscript in MS Word .docx or similar editable format.

We are committed to providing a fair and constructive peer-review process. Do not hesitate to contact me if there are specific requests from the reviewers that you believe are technically impossible or unlikely to yield a meaningful outcome.

*2) If you have not done so already please begin to revise your manuscript so that it conforms to our Article format instructions, available here.

*3) Include a revised version of any required Reporting Summary:

Please be aware of our guidelines on digital image standards.

[redacted]

We hope to receive your revised manuscript within 8 weeks. If you cannot send it within this time, please let us know.

Sincerely,

Tiago

Tiago Faial, PhD
Chief Editor
Nature Genetics
<https://orcid.org/0000-0003-0864-1200>

Reviewers' Comments:

Reviewer #1:

Remarks to the Author:

The authors have done a good job of addressing some of my concerns. However, I feel that one of my main concerns has not been addressed. The central thesis of the study is that genes are co-activated (co-bursting) following depletion of the cohesin subunit RAD21. Because cohesin acts in cis via loop

extrusion the authors claim that the co-activation is in cis. In the original manuscript the authors only compare co-activation in cis and concluded that the co-activation was in cis. However, now they have also compared co-activation in trans (genome-wide). The conclusion from these data is that the effect size of co-activation in trans is roughly similar to that in cis. The authors claim the degree of co-activation in cis is higher than in trans, but this is a very tiny increase. Particularly when compared to the amount of co-activation that is observed genome-wide in this analysis. Unfortunately, these analyses more invalidate their central thesis, rather than support it.

The authors perform a seqFISH analysis for a number of gene pairs in trans and show that there is no increased co-bursting. However, here they choose random gene pairs which do not necessarily co-burst. The analysis that the authors should do is to show for a number of supposedly co-bursting gene pairs that the gene pairs only show co-bursting when the genes are on the same chromosome (in cis), but not when these genes are on homologous chromosomes. So when comparing gene X and gene Y they should be co-bursting when they are both on the maternal or paternal allele. When gene X is on the maternal allele and gene Y is on the paternal allele (or vice versa) there should be no co-bursting. This can be achieved by performing these analyses in 129/Cast F1 hybrids as previously suggested, because in these lines the parental alleles can be distinguished. If the genes do show co-activation in trans this effect is likely contributable to some other factor. This might also explain why there is genome-wide correlation in the single cell gene expression data.

Reviewer #2:

Remarks to the Author:

I thank the authors for their extensive response to our first review and appreciate the response from the authors to many of my concerns. This study showcases a new role for cohesin in limiting co-expression of genes located in different domains along the chromosome using state of the art single-cell sequencing and microscopy techniques.

I do have one comment and remaining question in response to their reply.

1. One small concern I have raised is the reduced expression of Med6 in the Rad21-mAID-eGFP Med6-Halo background (Ext Data Figure 7e, note the legend should be corrected to Med6-Halotag(e)) and asked whether reduced Med6 expression affects size of the ACDs. The authors did not address this expression discrepancy and only commented on the Rad21 levels. Med6, being a smaller protein should not present such large differences due to transfer.

Reviewer #3:

Remarks to the Author:

The authors have presented a careful and comprehensive revision, which has substantially improved an exciting manuscript with rich data novel insights into the influences of 3D genome organization on transcriptional control.

There are just 2 of my original concerns that have not been addressed that I should still like to see the authors address. The first involves no additional experiments, just an alternative way of plotting the available data. The second involves no additional experiments, but some changes to the wording used to describe the results.

1) The concern about how segregation errors would artefactually create co-bursting and co-accessibility were apparently not sufficiently clearly explained in my first comments, as the authors have added several experiments and simulations that in fail to address this concern.

- First, can segregation cause co-bursting? The logic for this can be demonstrated with some simple mathematics, without requiring new simulations. I explain here:

- Consider two genes on the same chromosome that are normally bursting in 20% of the time. The probability the two genes burst at the same time in the same cell is expected to be 4% (0.2×0.2) if the genes are independent – i.e. no co-bursting tendency. Anything statistically larger than this could be taken as evidence of co-bursting. Now assume that a segregation error in a given cell (due to cohesin depletion treatment, a known driver of segregation errors) means this cell has 4 copies of the chromosome that contains both these 2 genes, instead of 2. Critically, this segregation error is unknown to experimenter – the analysis still assumes the cells are diploid and there are only 2 copies. However, apparent co-bursting is observed when any 2 of the 4 burst – It is quite rare that 3 of the 4 or 4 of the 4 all burst at the same time, but there are 4-choose-2 (6) different ways two pairs might be bursting at the same time rather than only 1 way, so bursts will be more common (this is a simple stats exercise). This increase will be interpreted as enhanced co-bursting, even though it we see it actually arises from having more independent copies.

- Note if we replace “co-bursting” above with “co-accessibility” and an identical argument explains why copy number variation following the long cohesin depletion could be responsible for the apparent increase in co-accessibility.

- In their response letter the authors present some results from some additional simulations to conclude that extra copies could not give the illusion of co-bursting. The simulations were not described in nearly sufficient detail for me to understand what they are doing, on the take home message was clearly presented, and this message is at odds with the basic logic just described. I suspect the problem lies with a misunderstanding of the concern as originally stated, and I hope my revised presentation of the concern is clearer. The new experimental analysis the authors added of the ATAC-seq data that a lack of understanding of the concern (as I describe next),

- A natural data driven test for the role of segregation errors in driving the “co-accessibility” trend, requested in my first review, is to look at read abundance *per chromosome per cell* in the ATAC data. This could be presented as a heatmap for example of cells x chromosomes with color as read coverage. The authors instead chose plot read abundance per chromosome in the response, -- however, they show only the result averaged over all cells, which defeats the utility of this analysis. It is not sufficient to look at the variance of this average. If you look at each cell and your read-depth per chromosome per cell is decent enough, if cell 1 has 2x as many reads from chr1 compared to some other cell or compared to the average cell, cell1 may have a segregation error giving it an extra chr1. Somewhere in the population is likely another cell missing a copy of chr1, and if you average these read counts per chromosome across all cells these two balance each-other out and you won't see the effect. I request the authors examine the reads per chromosome per cell in the ATAC seq data. If fluctuations in accessibility concentrate at the chromosomal level (e.g. cell 1 has more co-accessibility from chr 1 and more than expected reads from chr 1 and less evidence of co-accessibility from other chromosomes, while cell 2 it is mostly genes on chr 5 that show co-accessibility and chr 5 also shows more reads, -- this would be evidence of a segregation driven error. If instead on a cell-by-cell basis the reads and the co-accessibility are uniformly distributed rather than concentrated by chromosomes, this would go along way to eliminating this concern.

2) I think the conclusions about the mediator clusters “fusing” should be reworded. It is perfectly reasonable for the authors to describe *their interpretation* of this data to suggest a fusion in the

discussion. However, the fact that the 100 largest clusters get larger upon cohesin depletion could represent (1) increased mediator levels (2) decreased solubility of mediator so more of it clusters (3) or fusion of smaller clusters into bigger ones. Given that all 3 options are possible, and the authors decline doing further experiments to eliminate the other options, I think it inappropriate to conclude in the results section when they describe the data that this is "fusion" – such a speculative (however likely) interpretation should be saved for the discussion (or the authors could introduce additional experiments to eliminate the other possibilities, though personally I don't think it necessary). I appreciate the authors frankness about the many challenges involved in being able to detect and measure the soluble / unclustered mediator and the many challenges that arise from the use of (arbitrary) thresholds in this work and prior work by other authors, in order to define clusters in the first place (the text could have been more frank about these limitations as well, maybe with a "limitations" section, even though other authors are have also not been frank about such limitations). But it is not enhancing the paper to over-interpret the data when describing the observation.

- The authors could always add additional controls to try to rule out some of these possibilities, like the suggested western blot to test an effect on global mediator levels. Or they can just describe in the discussion why they think fusion is the most likely – I happen to agree with that. But I do have a problem with saying the clusters are bigger, so they must have gotten bigger by the small clusters fusing.

I commend the authors on an exciting and comprehensive work.

- Alistair Boettiger,
Stanford University

Author Rebuttal, first revision:

Point-by-point response (R2)

Nature Genetics submission: NG-A61918-R

Cohesin prevents cross-domain gene co-activation

Reviewer #1:

Remarks to the Author:

The authors have done a good job of addressing some of my concerns. However, I feel that one of my main concerns has not been addressed. The central thesis of the study is that genes are co-activated (co-bursting) following depletion of the cohesin subunit RAD21. Because cohesin acts in cis via loop extrusion the authors claim that the co-activation is in cis. In the original manuscript the authors only compare co-activation in cis and concluded that the co-activation was in cis. However, now they have also compared co-activation in trans (genome-wide). The conclusion from these data is that the effect size of co-activation in trans is roughly similar to that in cis. The authors claim the degree

of co-activation in *cis* is higher than in *trans*, but this is a very tiny increase. Particularly when compared to the amount of co-activation that is observed genome-wide in this analysis. Unfortunately, these analyses more invalidate their central thesis, rather than support it.

We understand this reviewer's main concern that our single cell genomic assays (SMART-SCRB and scATAC-seq) might not have the ability to distinguish whether gene co-activation (or chromatin co-opening) occurs on the same chromosome (in *cis*) or from homologous chromosomes. This is a good point. However, we would like to point out that, using allele-specific single-cell RNA-sequencing data of hybrid mouse cells (CAST/EiJ x C57BL/6J), a previous study showed that positive long-range gene co-expression correlation only occurs in *cis* but not between homologous chromosomes¹. One hallmark of the *cis* dependence is that the co-expression coefficient decays as a function of the genomic distance. This makes sense as genes residing in separate homologous chromosomes are not chromosomally linked and thus their co-expression correlation would not decay as the function of genomic distance. Both our gene co-activation (SMART-SCRB) and chromatin co-accessibility (scATAC-seq) coefficients decay as the function of the genomic distance (Extended Data Fig. 3a and Fig. 2c). These results are consistent with the previous report, suggesting that gene co-activation and chromatin co-opening that we measured occur *in cis*. We have revised our manuscript to explain the meaning of these observations more clearly (Page 4, Line 90-95). We thank this reviewer's comment to make our manuscript logically sound.

Secondly, since the final steady-state mRNA level is the accumulative effect from multiple regulatory steps, including chromatin accessibility, promoter-enhancer interaction, transcription factor/cofactor binding, transcription initiation, elongation, termination and mRNA decay, it in principle does not reflect direct and immediate effect of cohesin loss on gene regulation, as this reviewer acknowledged in the previous comments. In contrast, scATAC-seq detects immediate early regulatory changes on *cis* elements. scATAC-seq analysis showed a strong increase in chromatin co-opening between ACDs *in cis* with no significant changes *in trans* (Fig. 2d). We hope this reviewer can kindly take into account results from both assays and in context.

The authors perform a seqFISH analysis for a number of gene pairs in *trans* and show that there is no increased co-bursting. However, here they choose random gene pairs

which do not necessarily co-burst. The analysis that the authors should do is to show for a number of supposedly co-bursting gene pairs that the gene pairs only show co-bursting when the genes are on the same chromosome (in *cis*), but not when these genes are on homologous chromosomes. So when comparing gene X and gene Y they should be co-bursting when they are both on the maternal or paternal allele. When gene X is on the maternal allele and gene Y is on the paternal allele (or vice versa) there should be no co-bursting. This can be achieved by performing these analyses in 129/Cast F1 hybrids as previously suggested, because in these lines the parental alleles can be distinguished. If the genes do show co-activation in *trans* this effect is likely contributable to some other factor. This might also explain why there is genome-wide correlation in the single cell gene expression data.

We appreciate this reviewer's comment. However, FISH approaches (e.g. smRNA-FISH, seqFISH) are based on multi-site hybridization which are in practice incapable of detecting single-nucleotide polymorphism (SNP) and thus allele-specific transcriptional bursting in 129/Cast F1 hybrids ES cells.

On the other hand, as homologous chromosomes occupy distinct territories within the nucleus (Fig. 3b), Intron-FISH and seqFISH have the spatial resolution to probe gene co-bursting on the same chromosome. Specifically, we use a proximity cutoff of $1\mu\text{m}$ to selectively detect co-bursting events on the same chromosome in our FISH-based experiments. To demonstrate the effectiveness of this cutoff, we plotted the distance distribution between spots of co-bursting gene pairs randomly selected from 208 active genes in Chr2 (SeqFISH data set) (Fig. RR1a). As shown from the results, there is a specific enrichment of co-bursting spots on the same chromosome with distance $< 1\mu\text{m}$, whereas other distances are roughly uniformly distributed along the scale up to $20\mu\text{m}$ and reflects co-bursting spots in homologous chromosomes due to unconstrained movement. Consistent with this result, distances between co-bursting gene pairs from different chromosomes only showed the uniform distribution within the range between $0\mu\text{m}$ and $20\mu\text{m}$, without the concentrated fraction $< 1\mu\text{m}$ (Fig. RR1b). Therefore, these results demonstrate that $1\mu\text{m}$ is an appropriate cut-off for analyzing gene co-bursting events in *cis*. We elaborated this technical detail in the revised manuscript (Page 6, Line 157-160) and included these results as Extended Data Fig. 4e-f. Taken together, results from both genomic and imaging assays confirm that elevated chromatin co-accessibility and gene co-expression upon cohesin loss occurs on the same chromosome.

Figure RR1. Co-bursting of gene pairs in *cis* predominately occurs within 1 μm distance. **(a)** 10 gene pairs were randomly selected from the pool of 208 active genes in Chr 2. The distances between spots of co-bursting gene pairs were calculated and plotted for normal and cohesin-depleted conditions. **(b)** The distances between co-bursting gene pairs from different chromosomes were calculated and plotted for normal and cohesin-depleted conditions. Each circle represents the distance between one pair of co-bursting spots within a single cell. The dotted line indicates 1 μm cut-off.

Reviewer #2:

Remarks to the Author:

I thank the authors for their extensive response to our first review and appreciate the response from the authors to many of my concerns. This study showcases a new role for cohesin in limiting co-expression of genes located in different domains along the chromosome using state of the art single-cell sequencing and microscopy techniques.

We appreciate this reviewer's comment.

I do have one comment and remaining question in response to their reply.

1. One small concern I have raised is the reduced expression of Med6 in the Rad21-mAID-eGFP Med6-Halo background (Ext Data Figure 7e, note the legend should be corrected to Med6-HaloTag(e)) and asked whether reduced Med6 expression affects size of the ACDs. The authors did not address this expression discrepancy and only commented on the Rad21 levels. Med6, being a smaller protein should not present such large differences due to transfer.

We thank the reviewer for his/her comments. To address the point, we calculated the auto-correlation function (\$g(r)\$ ) from 3D-ATAC-PALM localization data for wild-type mESCs, RAD21-mAID-eGFP+/+ knockin mESCs and RAD21-mAID-eGFP+/+/MED6-HaloTag+/+ double knockin mESCs, respectively. Our results show no significant changes in \$g(r)\$ values across different length scales, suggesting that the sizes of ACDs are not significantly affected by the band intensity reduction of Med6-HaloTag in western blots (Fig. RR2). This result is also consistent with our previous finding that even alpha amanitin induced Pol II inhibition and degradation do not significantly affect \$g(r)\$ of 3D-ATAC-PALM localizations ², suggesting that the structural maintenance of ACDs is likely independent of transcriptional activities.

Figure RR2. There is no significant difference for spatial distribution of accessible chromatin sites among wild-type mESCs, RAD21-mAID-eGFP^{+/+} knockin mESCs and RAD21-mAID-eGFP^{+/+}/MED6-HaloTag^{+/+} double knockin mESCs. n is the number of single cells used for analysis. Error bar indicates standard error of mean (SEM). Two-sided Mann-Whitney U test was applied for comparing data points at $g(0)$ for three pairs and n.s. represents not significant.

Reviewer #3:

Remarks to the Author:

The authors have presented a careful and comprehensive revision, which has substantially improved an exciting manuscript with rich data novel insights into the influences of 3D genome organization on transcriptional control.

We appreciated this reviewer's previous suggestions, which help improve our manuscript significantly.

There are just 2 of my original concerns that have not been addressed that I should still like to see the authors address. The first involves no additional experiments, just an alternative way of plotting the available data. The second involves no additional experiments, but some changes to the wording used to describe the results.

1) The concern about how segregation errors would artefactually create co-bursting and co-accessibility were apparently not sufficiently clearly explained in my first comments, as the authors have added several experiments and simulations that in fail to address this concern.

- First, can segregation cause co-bursting? The logic for this can be demonstrated with some simple mathematics, without requiring new simulations. I explain here: • Consider two genes on the same chromosome that are normally bursting in 20% of the time. The probability the two genes burst at the same time in the same cell is expected to be 4% (0.2×0.2) if the genes are independent – i.e. no co-bursting tendency. Anything statistically larger than this could be taken as evidence of co-bursting. Now assume that a segregation error in a given cell (due to cohesin depletion treatment, a known driver of segregation errors) means this cell has 4 copies of the chromosome that contains both these 2 genes, instead of 2. Critically, this segregation error is unknown to experimenter – the analysis still assumes the cells are diploid and there are only 2 copies. However, apparent co-bursting is observed when any 2 of the 4 burst – It is quite rare that 3 of the 4 or 4 of the 4 all burst at the same time, but there are 4-choose-2 (6) different ways two pairs might be bursting at the same time rather than only 1 way, so bursts will be more common (this is a simple stats exercise). This increase will be interpreted as enhanced co-bursting, even though it we see it actually arises from having more independent copies. - Note if we replace “co-bursting” above with “co-accessibility” and an identical

argument explains why copy number variation following the long cohesin depletion could be responsible for the apparent increase in co-accessibility. - In their response letter the authors present some results from some additional simulations to conclude that extra copies could not give the illusion of co-bursting. The simulations were not described in nearly sufficient detail for me to understand what they are doing, on the take home message was clearly presented, and this message is at odds with the basic logic just described. I suspect the problem lies with a misunderstanding of the concern as originally stated, and I hope my revised presentation of the concern is clearer.

We thank this reviewer for further clarification. We do understand this reviewer's concern that chromosome loss or gain can generate gene expression and chromatin accessibility co-fluctuation at the chromosomal level, which might affect our correlation analysis. Thus, we performed additional chromatin co-accessibility analysis after filtering out cells with potential chromosome mis-segregation. Our results showed that the enhanced intra-chromosomal co-accessibility upon cohesin remains largely unchanged (Fig. RR3), suggesting that chromosome mis-segregation is unlikely a main driver underlying this observation.

The new experimental analysis the authors added of the ATAC-seq data that a lack of understanding of the concern (as I describe next), - A natural data driven test for the role of segregation errors in driving the "co-accessibility" trend, requested in my first review, is to look at read abundance *per chromosome per cell* in the ATAC data. This could be presented as a heatmap for example of cells x chromosomes with color as read coverage. The authors instead chose plot read abundance per chromosome in the response, -- however, they show only the result averaged over all cells, which defeats the utility of this analysis. It is not sufficient to look at the variance of this average. If you look at each cell and your read-depth per chromosome per cell is decent enough, if cell 1 has 2x as many reads from chr1 compared to some other cell or compared to the average cell, cell1 may have a segregation error giving it an extra chr1. Somewhere in the population is likely another cell missing a copy of chr1, and if you average these read counts per chromosome across all cells these two balance each-other out and you won't see the effect. I request the authors examine the reads per chromosome per cell in the ATAC seq data. If fluctuations in accessibility concentrate at the chromosomal level (e.g. cell 1 has more co- accessibility from chr 1 and more than expected reads from chr 1 and less evidence of co- accessibility from other chromosomes, while cell 2 it is mostly genes on

chr 5 that show co-accessibility and chr 5 also shows more reads, -- this would be evidence of a segregation driven error. If instead on a cell-by-cell basis the reads and the co-accessibility are uniformly distributed rather than concentrated by chromosomes, this would go along way to eliminating this concern.

We thank the reviewer for this suggestion. We further tested this point by using our scATAC-seq data. Since the chromatin co-accessibility coefficient is calculated from ATAC read values of ACD pairs between all batch normalized cells, it cannot be directly correlated to chromosomal ATAC counts at the single-cell level. To address this question, we pooled ATAC read counts within the same chromosome in each cell and calculated the normalized counts per chromosome to the average count of that particular cell. The normalized counts per chromosome for 1,000 cells were plotted as heatmaps for normal and RAD21-depleted conditions, respectively (Fig. RR3a). By setting a stringent threshold (50% gain or loss), we filtered out cells with significant deviation of chromosome-wise ATAC counts, which are possibly cells with either chromosome gain or loss. The fractions of cells with chromosome mis-segregation for normal and RAD21-depleted conditions are 15.38% and 18.27%, respectively (Fig. RR3b), which are comparable with previous report³. We then calculated chromatin co-accessibility coefficient based on the remaining dataset in which mis-segregated cells have been eliminated. Our results showed that the enhanced intra-chromosomal co-accessibility remains largely unchanged (Fig. RR3c), suggesting that chromosome mis-segregation is unlikely a main driver underlying the increase in cross-domain chromatin co-accessibility upon cohesin loss.

Figure RR3. The enhanced intra-chromosomal co-accessibility after cohesin loss remains significant after elimination of cells with chromosomal mis-segregation. (a). The ATAC peaks within the same chromosome were pooled together for each individual cell and the counts per chromosome were normalized to the average count of that particular cell. In the left panels, the normalized counts per chromosome for 1,000 cells were plotted as heatmaps for normal and RAD21-depleted conditions. In the right panels, the normalized counts per chromosome for single cells after elimination of those with segregation error were plotted correspondingly. Black arrows indicate candidate cells with chromosome gain and white arrow indicate those with chromosome loss. (b). Estimated fraction of cells with chromosomal mis-segregation in normal and RAD21-depleted conditions. (c). The heatmap shows that enhanced intra-chromosomal co-accessibility after cohesin loss still holds after elimination of cells with chromosomal mis-segregation.

2) I think the conclusions about the mediator clusters “fusing” should be reworded. It is perfectly reasonable for the authors to describe *their interpretation* of this data to suggest a fusion in the *discussion*. However, the fact that the 100 largest clusters get larger upon cohesin depletion could represent (1) increased mediator levels (2) decreased solubility of mediator so more of it clusters (3) or fusion of smaller clusters into bigger ones. Given that all 3 options are possible, and the authors decline doing further experiments to eliminate the other options, I think it inappropriate to conclude in the results section when they describe the data that this is “fusion” – such a speculative (however likely) interpretation should be saved for the discussion (or the authors could introduce additional experiments to eliminate the other possibilities, though personally I don’t think it necessary). I appreciate the authors frankness about the many challenges involved in being able to detect and measure the soluble / unclustered mediator and the many challenges that arise from the use of (arbitrary) thresholds in this work and prior work by other authors, in order to define clusters in the first place (the text could have been more frank about these limitations as well, maybe with a “limitations” section, even though other authors are have also not been frank about such limitations). But it is not enhancing the paper to over-interpret the data when describing the observation. - The authors could always add additional controls to try to rule out some of these possibilities, like the suggested western blot to test an effect on global mediator levels. Or they can just describe in the discussion why they think fusion is the most likely – I happen to agree with that. But I do have a problem with saying the clusters are bigger, so they must have gotten bigger by the small clusters fusing.

We thank this reviewer’s suggestion and appreciation of the technical difficulty associated with analyzing Med6 live-cell imaging data sets. we replaced ‘fusion’ with more neutral

wording 'spatial mixing' in consistence with our previous publication describing accessible chromatin domains ².

I commend the authors on an exciting and comprehensive work.

We appreciate this reviewer's comment.

References:

- 1 Sun, M. Y. & Zhang, J. Z. Chromosome-wide co-fluctuation of stochastic gene expression in mammalian cells. *Plos Genet* **15** (2019).
- 2 Xie, L. *et al.* BRD2 compartmentalizes the accessible genome. *Nat Genet* **54**, 481-491 (2022).
- 3 Georges, R. O. *et al.* Acute deletion of TET enzymes results in aneuploidy in mouse embryonic stem cells through decreased expression of Khdc3. *Nature Communications* **13** (2022).

Decision Letter, second revision:

5th Apr 2024

Dear Dr. Liu,

hope this email finds you well and I sincerely apologize for the delay in providing our decision.

Thank you for submitting your revised manuscript "Cohesin prevents cross-domain gene co-activation" (NG-A61918R1). We went through the point-by-point response to Reviewers and we think your manuscript has improved in revision therefore, we'll be happy in principle to publish it in Nature Genetics, pending minor revisions to comply with our editorial and formatting guidelines.

Thank you again for your interest in Nature Genetics and please do not hesitate to contact me if you have any questions.

Congratulations!

My best wishes,
Chiara

Chiara Anania, PhD
Associate Editor
Nature Genetics
<https://orcid.org/0000-0003-1549-4157>

Final Decision Letter:

27th Jun 2024

Dear Dr. Liu,

I am delighted to say that your manuscript "Cohesin prevents cross-domain gene co-activation" has been accepted for publication in an upcoming issue of Nature Genetics.

Your paper will be published online after we receive your corrections and will appear in print in the next available issue. You can find out your date of online publication by contacting the Nature Press Office (press@nature.com) after sending your e-proof corrections.

Please note that *Nature Genetics* is a Transformative Journal (TJ). Authors may publish their research with us through the traditional subscription access route or make their paper immediately open access

through payment of an article-processing charge (APC). Authors will not be required to make a final decision about access to their article until it has been accepted. Find out more about Transformative Journals

Authors may need to take specific actions to achieve compliance with funder and institutional open access mandates. If your research is supported by a funder that requires immediate open access (e.g. according to Plan S principles) then you should select the gold OA route, and we will direct you to the compliant route where possible. For authors selecting the subscription publication route, the journal's standard licensing terms will need to be accepted, including <https://www.nature.com/nature-portfolio/editorial-policies/self-archiving-and-license-to-publish>. Those licensing terms will supersede any other terms that the author or any third party may assert apply to any version of the manuscript.

If you have not already done so, we strongly recommend that you upload the step-by-step protocols used in this manuscript to protocols.io. protocols.io is an open online resource that allows researchers to share their detailed experimental know-how. All uploaded protocols are made freely available and are assigned DOIs for ease of citation. Protocols can be linked to any publications in which they are used and will be linked to from your article. You can also establish a dedicated workspace to collect all your lab Protocols. By uploading your Protocols to protocols.io, you are enabling researchers to more readily reproduce or adapt the methodology you use, as well as increasing the visibility of your protocols and papers. Upload your Protocols at <https://protocols.io>. Further information can be found at <https://www.protocols.io/help/publish-articles>.

Sincerely,
Chiara

Chiara Anania, PhD
Associate Editor
Nature Genetics
<https://orcid.org/0000-0003-1549-4157>